# The Interplay Between Interpolation and Aggregation in Regression: Optimal Sample Complexity

**Mikael Møller Høgsgaard** [* 1 2]   **Kasper Green Larsen** [* 1]   **Liang-Yu Zou** [* 1]

## Abstract

This work investigates theoretically the interplay between interpolation and aggregation in regression. We establish that the $\gamma$-graph dimension characterizes learnability for a broad class of natural aggregation procedures. Furthermore, we prove that an extremely simple aggregation procedure, combining three interpolating hypotheses via the median, is optimal among all these aggregation procedures, and is strictly more powerful than proper learning. Finally, we show that some hypothesis classes are learnable only by aggregating infinitely many hypotheses or by using non-interpolating aggregation rules (which may predict outside the range of their inputs), and any finite interpolating aggregation fails to achieve even trivial performance.

## 1. Introduction

*Interpolation* of training data (or near-interpolation) is a recurring phenomenon in both classical machine learning and modern deep learning. At the same time, *aggregation* procedures, including bagging, random forests, mixture-of-experts, and voting schemes, are among the most successful practical tools for improving prediction. In this paper, we investigate theoretically the interplay of these two features in the regression setting, addressing the following questions: when the hypothesis class is rich enough to interpolate the training data, can aggregation provably reduce error compared to a single predictor; if so, which complexity measure governs learnability for natural aggregation rules, and how do these procedures compare to general improper learning?

To answer these questions formally, we adopt the realiz-

able regression framework of (Attias et al., 2023). Given a hypothesis class $\mathcal{H} \subseteq [0,1]^{\mathcal{X}}$, this framework considers distributions $\mathcal{D}$ over $\mathcal{X} \times [0,1]$ for which a training sequence $\mathbf{S} \sim \mathcal{D}^n$ can be labeled by a hypothesis from $\mathcal{H}$, modelling settings in which the class can interpolate the training data. In the following we call a learning algorithm an *interpolator* if it always returns a hypothesis in $\mathcal{H}$ that matches all training labels (i.e., a proper learner)[1]. (Attias et al., 2023) introduces and studies performance measured by the *cutoff loss* with parameter $\gamma \in (0,1)$:

$$\mathcal{L}_{\mathcal{D}}^{\gamma}(h) := \mathop{\mathbb{P}}_{(x,y)\sim\mathcal{D}}[|h(x) - y| > \gamma], \qquad (1)$$

which quantifies the probability of making an error larger than $\gamma$. Bounds on the cutoff loss control tail behavior of the predictor's deviation from the true label and therefore imply bounds on the $L_p$ loss by integration:

$$\mathop{\mathbb{E}}_{(x,y)\sim\mathcal{D}}[|h(x) - y|^p] = \int_0^1 \mathcal{L}_{\mathcal{D}}^{\gamma^{1/p}}(h)\, d\gamma.$$

Henceforth, we use the cutoff loss, to measure performance.

Building on the realizable-regression framework of (Attias et al., 2023), we show that the $\gamma$-graph dimension $d_\gamma$ (Definition 3.2) governs the worst-case sample complexity of a broad class of natural aggregation methods, and that aggregation can strictly outperform any single proper learner. At the same time, we prove that these aggregation procedures remain weaker than general learners, whose learnability is characterized by the $\gamma$-OIG dimension (Definition 3.9) in (Attias et al., 2023). Below we briefly state our main results.

**Sample complexity lower bound for interpolator-based aggregation with proper rules.** We first consider interpolator-based aggregation algorithms that partition the training data into possibly overlapping parts, apply an interpolator to each part, and combine the resulting hypotheses via a *proper* aggregation rule, one that always outputs a value equal to one of its inputs (e.g., any order statistic such as the median when the number of inputs is odd). We prove

---

[*]Equal contribution [1]Department of Computer Science, Aarhus University, Aarhus, Denmark [2]Department of Statistics, University of Oxford, Oxford, United Kingdom. Correspondence to: Mikael Møller Høgsgaard <hogsgaard@cs.au.dk>, Kasper Green Larsen <larsen@cs.au.dk>, Liang-Yu Zou <zou@cs.au.dk>.

*Proceedings of the 43rd International Conference on Machine Learning*, Seoul, South Korea. PMLR 306, 2026. Copyright 2026 by the author(s).

---

[1](Attias et al., 2023) call such an algorithm an ERM-algorithm; however, we choose to use the word interpolator to avoid confusion with an algorithm minimizing the empirical cutoff loss Equation (1), which does not necessarily interpolate the data

that for any hypothesis class with $\gamma$-graph dimension $d_\gamma$, there exist an interpolator and a realizable distribution such that any such interpolator-based algorithm requires sample complexity $\Omega(d_\gamma/\varepsilon)$ to achieve expected cutoff loss $\varepsilon$. This strengthens Theorem 1 of (Attias et al., 2023) in the following sense: their lower bound applies to a single interpolator, while ours shows that the same $\Omega(d_\gamma/\varepsilon)$ barrier persists even when the learner may train arbitrarily many interpolators on possibly overlapping subsamples and combine their predictions through any proper aggregation rule.

**Sample complexity lower bound for finite aggregation with interpolating rules.** We then consider a (broader) class: *finite aggregation* algorithms that select finitely many hypotheses from $\mathcal{H}$ (not necessarily via an interpolator) and combine them using an *interpolating* aggregation rule, one whose output lies between the minimum and maximum of its inputs (e.g., the mean or any convex combination). We construct hypothesis classes with $\gamma$-graph dimension $d_\gamma$ for which any such algorithm requires sample complexity $\Omega(d_\gamma/\varepsilon)$, showing that the worst-case sample complexity of these algorithms is also governed by $d_\gamma$. Furthermore, the constructed classes have $\gamma$-OIG dimension 3, and are therefore learnable in $\tilde{O}(1/\varepsilon)$ samples by a one-inclusion-graph-based algorithm of (Attias et al., 2023). This establishes an arbitrarily large gap between finite interpolating aggregation and general learning algorithms sample complexity.

**Finite interpolating aggregation cannot learn some learnable classes.** The previous lower bound shows how the sample complexity explicitly scales with the $\gamma$-graph dimension when finite. The next result exhibit a hypothesis class with infinite $\gamma$-graph dimension and $\gamma$-OIG dimension 3 (hence learnable) for which *any* finite aggregation algorithm with an interpolating aggregation rule cannot achieve cutoff loss better than $1 - \varepsilon$, regardless of sample size, which for $\varepsilon = o(\gamma)$, implies strictly worse performance than random guessing, which achieves cutoff loss $1 - \Theta(\gamma)$. This provides an example of a strict separation between learnability and learnability by finite interpolating aggregation: to achieve even *non trivial* performance on these classes, learning algorithms must either aggregate infinitely many hypotheses or employ non-interpolating aggregation rules.

**Sample complexity upper bound for median-of-three interpolators is optimal.** While the learner of (Attias et al., 2023) is almost information-theoretically optimal, it is not efficiently implementable. However, we show that a remarkably simple algorithm, median-of-three interpolators, taking the pointwise median of just three interpolators trained on independent samples, achieves expected cutoff loss $O(d_\gamma/n)$ with sample size $n$, implying a sample complexity of $O(d_\gamma/\varepsilon)$, matching the above lower bounds up to constants and establishing optimality among such methods.

**Sample complexity lower bound for proper learning requires more samples than aggregation.** We prove that there exist hypothesis classes with $\gamma$-graph dimension $d_\gamma$ for which any *proper* learning algorithm, one that outputs a hypothesis from $\mathcal{H}$, requires sample complexity $\Omega((d_\gamma/\varepsilon)\ln(1/\varepsilon))$ to achieve expected cutoff loss $\varepsilon$. Combined with our $O(d_\gamma/\varepsilon)$ upper bound for median-of-three interpolators, this demonstrates that aggregation, even of just three interpolators, can be strictly more powerful than any proper learner.

## 2. Related Work

Foundational results on regression include (Bartlett et al., 1994; Kearns & Schapire, 1994; Simon, 1997; Alon et al., 1997; Bartlett & Long, 2024). In particular, (Simon, 1997) showed that finiteness of the scaled Natarajan dimension is necessary for PAC learnability in realizable regression. (Bartlett et al., 1994) showed that finite fat-shattering dimension (introduced by (Kearns & Schapire, 1994)) characterizes learnability when labels are corrupted by noise, and (Alon et al., 1997) proved that finiteness of the fat-shattering dimension characterizes agnostic PAC learnability. (Bartlett & Long, 2024) adapted the one-inclusion-graph algorithm (see (Haussler et al., 1994)) to regression and obtained expected error bounds for it.

More recently, (Attias et al., 2023) characterized learnability of realizable regression in both the batch and online model. Concurrently, (Aden-Ali et al., 2023) derived high-probability guarantees using one-inclusion-graph aggregation. (Daskalakis & Golowich, 2022) examined realizable regression in the online setting.

Selected works demonstrating that aggregation can strictly outperform a single predictor include results in binary classification (Hanneke, 2016; Larsen, 2023; Aden-Ali et al., 2023; Hanneke et al., 2024; Høgsgaard, 2025; Asilis et al., 2025a), multiclass classification (Daniely et al., 2011; Daniely & Shalev-Shwartz, 2014; Brukhim et al., 2022; Aden-Ali et al., 2023; Asilis et al., 2025b), and regression in settings different from those studied here (Audibert, 2007; Lecué & Mendelson, 2009; Mourtada et al., 2022).

A large literature studies generalization of interpolating models in the overparameterized regime, motivated by deep learning; representative works include (Belkin et al., 2018; Jacot et al., 2018; Soudry et al., 2018; Belkin et al., 2019a;b; Liang & Rakhlin, 2020; Bartlett et al., 2020; Belkin, 2021; Hastie et al., 2022).

## 3. Summary of Results

In this section, we summarize our main results on aggregation of (interpolating) predictors under the cutoff loss in the

realizable regression setting. Before stating our first result, we define the $\gamma$-graph dimension, a complexity measure introduced by (Attias et al., 2023). As we will see, it governs learnability for the aggregation methods we consider.

**Definition 3.1** ($\gamma$-graph shattering). A sequence $(x_1, \ldots, x_d) \in \mathcal{X}^d$ is $\gamma$-*graph shattered* by a hypothesis class $\mathcal{H} \subseteq [0, 1]^{\mathcal{X}}$ with witness $h \in \mathcal{H}$ if for any $b \in \{0, 1\}^d$ there exists a hypothesis $h_b \in \mathcal{H}$ such that:

- for all $i \in [d]$ with $b_i = 0$: $h_b(x_i) = h(x_i)$, and

- for all $i \in [d]$ with $b_i = 1$: $|h_b(x_i) - h(x_i)| > \gamma$.

**Definition 3.2** ($\gamma$-graph dimension). The $\gamma$-*graph dimension* of a hypothesis class $\mathcal{H} \subseteq [0, 1]^{\mathcal{X}}$, denoted $d_\gamma$, is the largest integer $d$ such that there exists a sequence $(x_1, \ldots, x_d) \in \mathcal{X}^d$ that is $\gamma$-graph shattered by $\mathcal{H}$ with some witness $h \in \mathcal{H}$.

In words, the $\gamma$-graph dimension is the largest number of points on which the class can realize all patterns of either matching the witness or being $\gamma$-far from it.

With the $\gamma$-graph dimension defined, we begin by examining the natural approach of aggregating interpolators via proper aggregation rules. Concretely, we study algorithms that partition a training sequence into subsequences, apply an interpolator to each part, and combine the resulting hypotheses via a proper aggregation rule, i.e., a rule that always outputs one of its inputs. Formally that is,

**Definition 3.3** (Interpolator-Based Aggregation Algorithm). An *interpolator-based aggregation algorithm* $\mathcal{A}'$ is a mapping that takes as input an interpolator $\mathcal{A}$ (see Definition A.1) and a training sequence $S$ to produce a predictor, i.e., $\mathcal{A}' \colon ((\mathcal{X} \times [0, 1])^* \to [0, 1]^{\mathcal{X}}) \times (\mathcal{X} \times [0, 1])^* \to [0, 1]^{\mathcal{X}}$. Specifically, given $S$ and $\mathcal{A}$, the algorithm $\mathcal{A}'(S, \mathcal{A})$ proceeds as follows:

1. Construct sub-training sequences $S_1, \ldots, S_m$ from $S$, where $m$ may depend on $S$ and each $(x, y) \in S_j$ satisfies $(x, y) \in S$. The sub-sequences need not be disjoint.

2. Apply $\mathcal{A}$ to each sub-sequence to obtain hypotheses $h_1, \ldots, h_m$.

3. Return the predictor $x \mapsto r(h_1(x), \ldots, h_m(x), x)$, where $r \colon [0, 1]^* \times \mathcal{X} \to [0, 1]$ is an aggregation rule.

When $\mathcal{A}$ is clear from context, we write $\mathcal{A}'(S) = \mathcal{A}'(S, \mathcal{A})$.

**Definition 3.4** (Proper Aggregation Rule). An aggregation rule $r \colon [0, 1]^* \times \mathcal{X} \to [0, 1]$ is *proper* if for every sequence $z_1, \ldots, z_m \in [0, 1]$ and every $x \in \mathcal{X}$, we have $r(z_1, \ldots, z_m, x) \in \{z_1, \ldots, z_m\}$.

Examples of proper aggregation rules include selecting the minimum, maximum, or any other order statistic. In particular, the median is proper when $m$ is odd.

Our first main result shows that any interpolator-based aggregation algorithm using a proper aggregation rule has worst-case sample complexity (worst case over choices of the interpolator and the realizable distribution) governed by the $\gamma$-graph dimension of the hypothesis class.

**Theorem 3.5.** *For any hypothesis class $\mathcal{H} \subseteq [0, 1]^{\mathcal{X}}$ with $\gamma$-graph dimension $d_\gamma \geq 2$ and any $0 < \varepsilon < 1/4$, there exist an interpolator $\mathcal{A}$ and a realizable distribution $\mathcal{D}$ such that the following holds. Let $\mathcal{A}'$ be any interpolator-based aggregation algorithm using $\mathcal{A}$ and a proper aggregation rule, and let $\mathbf{S} \sim \mathcal{D}^n$ with $n \leq d_\gamma/(32\varepsilon)$. Then,*

$$\mathbb{E}_{\mathbf{S} \sim \mathcal{D}^n}[\mathcal{L}_{\mathcal{D}}^\gamma(\mathcal{A}'(\mathbf{S}))] > \varepsilon. \tag{2}$$

Theorem 3.5 establishes that no interpolator-based aggregation algorithm using a proper aggregation rule can achieve sample complexity better than $\Omega(d_\gamma/\varepsilon)$ in the worst case, where $d_\gamma$ is the $\gamma$-graph dimension of the hypothesis class. We remark that the interpolator-based aggregation algorithm is allowed to aggregate any number of hypotheses. This strengthens Theorem 1 of (Attias et al., 2023), which established a similar lower bound for a single interpolator (i.e., without aggregation).

Proper aggregation of interpolators has also been studied in the realizable multiclass classification setting by (Asilis et al., 2025b). Their Theorem 10 roughly shows that for a *specific* family of multiclass hypothesis classes with graph dimension $d$, any proper aggregation of an interpolator has worst-case sample complexity at least $\Omega(d/\varepsilon)$. Our result is stronger in that it applies to arbitrary hypothesis classes in the regression setting.

We now provide a short overview of the proof of Theorem 3.5, with the full proof given in Section C.1.

**Proof overview Theorem 3.5:** The proof constructs a hard distribution $\mathcal{D}$ concentrated on a $\gamma$-graph-shattered set of size $d_\gamma$ with witness $h$, with one point having mass $1 - \Theta(\varepsilon)$ and the remaining points each having mass $\Theta(\varepsilon/(d_\gamma - 1))$; all points are labeled by $h$. A worst-case interpolator $\mathcal{A}$ is defined such that on a training sequence $S$, $\mathcal{A}$ returns a hypothesis from $\mathcal{H}$ that is $\gamma$-far from $h$ on all points not present in the training sequence $S$. This is possible because the point set is $\gamma$-graph shattered.

For a training sequence of size $n \leq d_\gamma/(32\varepsilon)$, we bound the number of low-probability points observed in $\mathbf{S}$. Using Markov's inequality, we show that with constant probability, $\Theta(d_\gamma)$ such points are missing from $\mathbf{S}$. Since each subsequence $S_i$ contains only points from $\mathbf{S}$, the same

points are also missing from each $S_i$, implying, by construction of the interpolator $\mathcal{A}$, that each corresponding hypothesis $h_i$ is $\gamma$-far from $h$ on those points. Because the aggregation rule is proper, the final predictor must agree with one of the hypotheses produced by $\mathcal{A}$ and therefore is also $\gamma$-far from $h$ on the $d_\gamma$ missing points, yielding loss $\Theta(d_\gamma) \cdot \Theta(\varepsilon/(d_\gamma - 1)) = \Theta(\varepsilon)$, establishing Theorem 3.5.

The distribution construction, with $1 - \Theta(\varepsilon)$ mass on one point and the remaining spread over the shattered set, is standard in learning theory and dates back to (Ehrenfeucht et al., 1989) and is also used in (Attias et al., 2023). The key novelty lies in observing that when aggregated via a proper aggregation rule, the interpolator can be chosen in a worst-case manner so that errors align, even without knowing in advance which points will be missing from the data. The size of the resulting error region is controlled by the $\gamma$-graph dimension, showing that the lower bound scales with this complexity measure.

In the above lower bound, the aggregation rule is required to be proper, which is also a key property for the proof, as it ensures that the final predictor's errors align with those of the individual hypotheses. This raises the question of whether "improper" aggregation rules can circumvent this lower bound.

To address this question, we consider a natural relaxation: *interpolating* aggregation rules, which output a value within the range of the input predictions rather than exactly one of them. We also generalize from interpolator-based aggregation algorithms to *finite aggregation algorithms*, which select a finite number of hypotheses from the hypothesis class (not necessarily interpolators) and combine them using an aggregation rule. Formally that is:

**Definition 3.6** (Finite Aggregation Algorithm). Let $\mathcal{H} \subseteq [0,1]^{\mathcal{X}}$ be a hypothesis class. A *finite aggregation algorithm* $\mathcal{A}' : (\mathcal{X} \times [0,1])^* \mapsto [0,1]^{\mathcal{X}}$ for $\mathcal{H}$ is defined as follows: Given a training sequence $S$, of size $n \in \mathbb{N}$ it selects at most $m \le m(n) < \infty$ hypotheses $h_1, \ldots, h_m$ from a hypothesis class $\mathcal{H}$, and combines them with an aggregation rule $r : [0,1]^* \times \mathcal{X} \to [0,1]$ and returns $x \mapsto r(h_1(x), \ldots, h_m(x), x)$.

**Definition 3.7** (Interpolating Aggregation Rule). An aggregation rule $r : [0,1]^* \times \mathcal{X} \to [0,1]$ is called *interpolating* if for any sequence $z_1, \ldots, z_m \in [0,1]$ and any $x \in \mathcal{X}$, it satisfies $r(z_1, \ldots, z_m, x) \in [\min_i z_i, \max_i z_i]$.

Interpolating aggregation rules capture a broad variety of ways to aggregate outputs, including natural choices such as the mean, the median, or any convex combination of the inputs. We note that every proper aggregation rule is interpolating, but the converse need not hold.

Furthermore, every interpolator-based aggregation algorithm (Definition 3.3), which always combines a finite num-

ber of hypotheses, is a finite aggregation algorithm, since the hypotheses $h_1, \ldots, h_m$ produced by applying an interpolator to subsequences of the training data are elements of $\mathcal{H}$. Thus, finite aggregation algorithms strictly generalize interpolator-based aggregation algorithms.

Perhaps surprisingly, our next result shows that even this more flexible class of algorithms cannot escape the $\gamma$-graph dimension lower bound. Specifically, there exist hypothesis classes for which any finite aggregation algorithm using an interpolating aggregation rule still requires sample complexity governed by the $\gamma$-graph dimension.

**Theorem 3.8.** *For any $\gamma \in (0,1)$ and $d_\gamma \ge 2$, there exists a hypothesis class $\mathcal{H} \subseteq [0,1]^{\mathcal{X}}$ with $\gamma$-graph dimension $d_\gamma$ and $\gamma$-OIG-dimension at most 3, such that for any finite aggregation algorithm $\mathcal{A}'$ using an interpolating aggregation rule $r$ and any $0 < \varepsilon < 1/32$, there exists a realizable distribution $\mathcal{D}$ over $\mathcal{X} \times [0,1]$ such that when $\mathcal{A}'$ is given a training sequence $\mathbf{S} \sim \mathcal{D}^n$ of size $n \le d_\gamma/(128\varepsilon)$,*

$$\mathbb{E}_{\mathbf{S} \sim \mathcal{D}^n}[\mathcal{L}_{\mathcal{D}}^\gamma(\mathcal{A}'(\mathbf{S}))] > \varepsilon. \tag{3}$$

Theorem 3.8 shows that even when allowing for improper aggregation rules and finite aggregation algorithms, there exist hypothesis classes for which the sample complexity is still lower bounded by $\Omega(d_\gamma/\varepsilon)$, so the sample complexity is governed by the $\gamma$-graph dimension. Thus, the lower bound of Theorem 3.5 extends beyond interpolator-based aggregation with proper rules to a significantly broader class of algorithms. However, unlike Theorem 3.5, this lower bound does not hold for all hypothesis classes.

The hypothesis classes constructed in Theorem 3.8 have low $\gamma$-OIG dimension (at most 3), which by Theorem 2 of (Attias et al., 2023) implies that the hypothesis class is learnable with sample complexity $\tilde{O}(1/\varepsilon)$ using an algorithm that is not a finite aggregation algorithm. Since Theorem 3.8 holds for any $d_\gamma \in \mathbb{N}$, this separates the sample complexity of finite aggregation algorithms with interpolating aggregation rules from that of general learning algorithms in terms of the $\gamma$-graph dimension.

To be able to provide the proof overview of Theorem 3.8, we will now define the $\gamma$-OIG dimension, first introduced by (Attias et al., 2023). The definition of the $\gamma$-OIG dimension is clearly motivated by the out-degree of the one-inclusion graph governing the number of errors the one-inclusion graph algorithm makes.

**Definition 3.9** ($\gamma$-OIG dimension). Let $\mathcal{H} \subseteq [0,1]^{\mathcal{X}}$ be a hypothesis class and let $S = \{x_1, \ldots, x_n\} \subseteq \mathcal{X}$ be a finite set. The *one-inclusion graph* induced by $S$ and $\mathcal{H}$ is the hypergraph with vertex set $V_n = \mathcal{H}|_S = \{(h(x_1), \ldots, h(x_n)) : h \in \mathcal{H}\}$ and hyperedge set $E_n$ where a hyperedge $(f, i)$ contains $v \in V$ if $v(x_j) = f(x_j)$ for all $j \in [n] \setminus \{i\}$.

The *$\gamma$-OIG dimension* of $\mathcal{H}$ is the largest integer $k$ such that

there exists a finite set $S \subseteq \mathcal{X}^k$ with $|S| = k$, such that there exists a finite subgraph $(V, E)$ of the one-inclusion graph $(V_n, E_n)$ induced by $S$, for which any orientation $\sigma : E_n \mapsto V_n$, has out-degree $\max_{v \in V} |\{i \in [n] : |\sigma(e_{v,i}) - v_i| > \gamma\}|$ strictly more than $n/3$.

We now provide a brief overview of the proof of Theorem 3.8, with the full proof given in Section C.2.

**Proof overview Theorem 3.8:** The proof constructs a hypothesis class $\mathcal{H}$ with $\gamma$-graph dimension $d_\gamma$ and $\gamma$-OIG dimension at most 3. The hypothesis class consists of hypotheses on the natural numbers, indexed by sets $A \subset \mathbb{N}$ with $|A| = d_\gamma$: each hypothesis $h_A$ satisfies $h_A(x) = 0$ if $x \in A$ and $h_A(x) = \gamma_A \in (\gamma, 1]$ otherwise, where $\gamma_A$ is a unique value associated with each hypothesis.

To see that the $\gamma$-graph dimension is exactly $d_\gamma$, observe first that any set $A$ of size $d_\gamma$ can be $\gamma$-graph shattered by taking $h_A$ as the witness hypothesis (all zeros on $A$) and including or excluding points from $A$ when shattering the set $A$. Conversely, for any set of size $d_\gamma + 1$, any witness hypothesis $h_A$ must output its unique value $\gamma_A$ on at least one point, call it $x'$. This uniqueness prevents the existence of a hypothesis $h'$ that agrees with $h_A$ on the point $x'$ while being $\gamma$-far on another point, as the former condition $\gamma_A = h_A(x') = h'(x')$ implies $h' = h_A$ by uniqueness of $\gamma_A$.

To bound the $\gamma$-OIG dimension, we analyze the structure of edges in the OIG graph. Since a hypothesis in an edge $(f, i)$ must agree with $f$ outside coordinate $i$, we have: a hypothesis with more than two non-zero values appears only in singleton edges, giving it out-degree zero; a hypothesis with one non-zero value appears in at most one non-singleton edge, giving it out-degree at most one. The only hypothesis that could appear in many non-singleton edges is the all-zeros hypothesis. However, by orienting each edge toward the hypothesis with the smallest value in the direction of the edge, the all-zeros hypothesis has out-degree zero. This implies the $\gamma$-OIG dimension is at most 3.

To construct the hard distribution $\mathcal{D}$, we draw a random vector $\mathbf{A} = (\mathbf{A}_1, \dots, \mathbf{A}_{d_\gamma})$ by setting $\mathbf{A}_1 = 1$ and sampling the remaining $d_\gamma - 1$ entries without replacement from $\{2, \dots, k_u\}$, where $k_u$ is chosen sufficiently large. For each realization of $\mathbf{A}$, the distribution $\mathcal{D}_{\mathbf{A}}$ assigns mass $1 - \Theta(\varepsilon)$ to $\mathbf{A}_1$ and mass $\Theta(\varepsilon/(d_\gamma - 1))$ to each of $\mathbf{A}_2, \dots, \mathbf{A}_{d_\gamma}$, with all labels equal to 0; this distribution is realizable by $h_{\{\mathbf{A}\}}$. We use the following equivalent way to generate a sample from $\mathcal{D}_{\mathbf{A}}$: draw an index $\mathbf{t} \in [d_\gamma]$, with $\mathbf{t} = 1$ having probability $1 - \Theta(\varepsilon)$ and every other index having probability $\Theta(\varepsilon/(d_\gamma - 1))$, and output $(\mathbf{A}_\mathbf{t}, 0)$. Thus an $n$-sample can be written as $\mathbf{S}(\mathbf{A}|\vec{\mathbf{t}}) = ((\mathbf{A}_{\mathbf{t}_1}, 0), \dots, (\mathbf{A}_{\mathbf{t}_n}, 0))$, where the index sequence $\vec{\mathbf{t}} = (\mathbf{t}_1, \dots, \mathbf{t}_n)$ is drawn independently of $\mathbf{A}$. In this view, conditioning on the sample reveals only the entries of $\mathbf{A}$ indexed by $\{\vec{\mathbf{t}}\}$; the unrevealed entries of $\mathbf{A}$ remain random over the unused part of $\{2, \dots, k_u\}$. This separation between the randomness of $\mathbf{A}$ and the randomness of the sampled indices is what lets us analyze the missing points from $\mathbf{A}$ and the predictions of the finite aggregation algorithm $\mathcal{A}'$ on them. We use the same coupling between a random support vector and an independent sequence of sampled indices in the proof sketches of Theorem 3.10 and Theorem 3.12.

More formally, by Markov's inequality, with constant probability, $\Theta(d_\gamma)$ points from $\mathbf{A} \setminus \{1\}$ are missing from the training sequence $\mathbf{S}$. Since $k_u$ is chosen sufficiently large, conditioned on $\mathbf{S}$, these missing points are effectively distributed uniformly at random over $\{2, \dots, k_u\}$. Now, since $\mathcal{A}'$ selects at most $\max_{n \le d_\gamma/(128\varepsilon)} m(n)$ hypotheses and each hypothesis outputs 0 on only $d_\gamma$ points, there are at most $d_\gamma \cdot \max_{n \le d_\gamma/(128\varepsilon)} m(n)$ points where any selected hypothesis outputs 0. On all other points, every selected hypothesis outputs a value strictly larger than $\gamma$, and by the interpolating property of $r$, so does the final predictor.

By choosing $k_u$ sufficiently large relative to $d_\gamma \cdot \max_{n \le d_\gamma/(128\varepsilon)} m(n)$, we ensure that with constant probability, almost all of the $\Theta(d_\gamma)$ missing points from $\mathbf{A}$ are misclassified by the final predictor. This yields a loss of at least $\Theta(d_\gamma) \cdot \Theta(\varepsilon/(d_\gamma - 1)) = \Theta(\varepsilon)$.

The hypothesis set construction is inspired by the first Cantor Class construction in (Daniely et al., 2011; Daniely & Shalev-Shwartz, 2014), used to show lower bounds for proper learners in the multiclass setting. However, our construction differs in that the hypotheses "share" the same input space and what corresponds to the zeros in our setting is a unique value in theirs. The latter distinction is key: it allows us to establish lower bounds not just for proper learners (which output a single hypothesis in $\mathcal{H}$), but for the much larger family of finite aggregation algorithms using interpolating aggregation rules.

Theorem 3.8 shows that for any $\gamma$-graph dimension $d_\gamma$, there exist hypothesis classes with $\gamma$-OIG dimension at most 3 for which finite aggregation algorithms with interpolating aggregation rules require sample complexity $\Omega(d_\gamma/\varepsilon)$. This creates an arbitrarily large gap between the sample complexity of such algorithms and that of general improper learning algorithms, which can achieve sample complexity $\tilde{O}(1/\varepsilon)$ for these classes.

Theorem 3.8 shows how the sample complexity of finite aggregation algorithms with interpolating aggregation rule scales with the $\gamma$-graph dimension when finite. This raises the question: what happens as the $\gamma$-graph dimension grows to infinity while the $\gamma$-OIG dimension remains constant? Can finite aggregation algorithms at least achieve some nontrivial loss bound, even if suboptimal?

Our next result answers this question in the negative. We

show that there exists a hypothesis class with $\gamma$-OIG dimension 3 for which any finite aggregation algorithm with an interpolating aggregation rule cannot achieve cutoff loss better than $1 - \varepsilon$, regardless of the sample size. In other words, for such hypothesis classes, finite aggregation algorithms with interpolating aggregation rules fail to achieve even non-trivial expected cutoff loss.

**Theorem 3.10.** *For any $0 < \gamma < 1$ and $0 < \varepsilon < 1$, there exists a hypothesis class $\mathcal{H}$ with $\gamma$-OIG dimension 3 such that for any finite aggregation learning algorithm $\mathcal{A}'$ with an interpolating aggregation rule and any $n' \in \mathbb{N}$, there exists a realizable distribution $\mathcal{D}$ such that if $\mathcal{A}'$ receives $n \leq n'$ i.i.d. samples $\mathbf{S} \sim \mathcal{D}^n$, it holds that*

$$\underset{\mathbf{S} \sim \mathcal{D}^n}{\mathbb{E}}[\mathcal{L}_{\mathcal{D}}^{\gamma}(\mathcal{A}'(\mathbf{S}))] \geq 1 - \varepsilon. \tag{4}$$

Theorem 3.10 establishes that there exist hypothesis classes with constant $\gamma$-OIG dimension, and hence learnable with $\tilde{O}(1/\varepsilon)$ samples by the algorithm of Theorem 2 in (Attias et al., 2023), for which any finite aggregation algorithm with an interpolating aggregation rule fails to achieve non-trivial expected cutoff loss, regardless of sample size. Furthermore for $\varepsilon = o(\gamma)$ this is strictly worse than random guessing (choose a random point in $\{2i\gamma\}_{i=1}^{1/(2\gamma)}$) which is $1 - \Theta(\gamma)$. This demonstrates a fundamental limitation of such algorithms: to achieve even *non trivial* performance on these classes, learning algorithms must either aggregate an infinite number of hypotheses or employ non-interpolating aggregation rules.

Results of a similar flavor have been established in the realizable multiclass classification setting by (Asilis et al., 2025b). Their Theorem 13 uses a construction inspired by the first Cantor class of (Daniely et al., 2011; Daniely & Shalev-Shwartz, 2014). Our proof technique is inspired by theirs but differs in two ways: it is adapted to the regression setting with cutoff loss, and it applies to interpolating (rather than proper) aggregation rules. We note that interpolating aggregation rules are not well-defined in the multiclass setting, while they are natural in regression. Furthermore in their setting, since the label space is infinite, random guessing can yield arbitrarily low error, so their lower bound does not imply failure to achieve non-trivial error.

We now provide a brief overview of the proof of Theorem 3.10, with the full proof given in Section C.3.

**Proof overview Theorem 3.10:** We construct a hypothesis class $\mathcal{H}$ with $\gamma$-OIG dimension 3 over a "split" input space $\mathcal{X} = \cup_{i \in \mathbb{N}}\{(k, x) : k = i^2, x \leq k\}$. For any $k = i^2$ with $i \in \mathbb{N}$ and each $A \subseteq [k]$ with $|A| = \sqrt{k}$, we define the hypothesis $h_{k,A}$ be 0 on points $(k, x)$ with $x \in A$ and a unique value $\gamma_{k,A} \in (\gamma, 1]$ elsewhere. We let the hypothesis class be $\mathcal{H} = \bigcup_{i=1}^{\infty}\{h_{k,A} : k = i^2, A \subseteq [k], |A| = \sqrt{k}\}$.

The $\gamma$-OIG dimension is at most 3 by similar reasoning as in the proof of Theorem 3.8. Given any point set, each hypothesis with two or more non-zero values appears only in singleton edges, giving it out-degree zero. Each hypothesis with one non-zero value appears in at most one non-singleton edge, giving it out-degree at most one. The only hypothesis that could appear in many non-singleton edges is the all-zeros hypothesis. However, by orienting each edge toward the hypothesis with the smallest value in that direction, which is the all-zeros hypothesis when present, its out-degree is zero. This implies the $\gamma$-OIG dimension is at most 3.

We now choose $k_u$, which determines the effective support size for our hard distribution, sufficiently large relative to $n'$ and $\max_{n \leq n'} m(n)$. For each $A \subseteq [k_u]$ with $|A| = \sqrt{k_u}$, we define a realizable distribution $\mathcal{D}_A$ that assigns uniform mass to points in $\{(k_u, i) : i \in A\}$ and zero mass elsewhere, with all these points labeled 0. This distribution is realizable by $h_{k_u, A}$. The hard distribution $\mathcal{D}_{\mathbf{A}}$ is then obtained by drawing $\mathbf{A}$ uniformly at random from all such sets. As in the proof overview of Theorem 3.8, we may equivalently draw a random vector enumerating $\mathbf{A}$ and then draw an independent sequence of indices whose entries select the points observed in the training sample.

By choosing $k_u = \omega(n'^2)$, the training sequence $\mathbf{S}$ observes only $o(\sqrt{k_u})$ points from $\mathbf{A}$. Conditioned on $\mathbf{S}$, the remaining points in $\mathbf{A}$ are effectively uniformly distributed over $[k_u]$. Since the finite aggregation algorithm $\mathcal{A}'$ selects at most $\max_{n \leq n'} m(n)$ hypotheses, and each selected hypothesis outputs 0 on only $\sqrt{k_u}$ points, there are at most $\sqrt{k_u} \cdot \max_{n \leq n'} m(n)$ points among $(k_u, 1), \ldots, (k_u, k_u)$ where any selected hypothesis outputs 0. On all other points, every selected hypothesis outputs a value strictly larger than $\gamma$, and by the interpolating property of $r$, so does the final predictor. Therefore, on the remaining points in $\mathbf{A}$, the error rate is roughly $(k_u - \sqrt{k_u} \max_{n \leq n'} m(n))/k_u$, which for $k_u$ sufficiently large is at least $1 - \varepsilon$.

In light of these lower bounds, and especially Theorem 3.10, it may seem that the right approach to realizable regression is to run the algorithm of Theorem 2 in (Attias et al., 2023). This algorithm achieves sample complexity $\tilde{O}(d_{\gamma,\mathrm{OIG}}/\varepsilon)$ for classes with $\gamma$-OIG dimension $d_{\gamma,\mathrm{OIG}}$, and (Attias et al., 2023) showed that this complexity measure characterizes learnability in realizable regression: a class is learnable if and only if its $\gamma$-OIG dimension is finite. Moreover, they showed that this sample complexity is nearly optimal. However, while the algorithm is information-theoretically close to optimal, it relies on an extension of the one-inclusion graph to regression and is inherently not efficiently implementable, and far from what is done in practice.

In contrast, finite aggregation, for instance, aggregating empirical risk minimizers/interpolators, is a workhorse of

practical machine learning. Theorem 3.5 and Theorem 3.8 establish that the sample complexity of such aggregation techniques, whether through proper or interpolating aggregation of finitely many hypotheses, is governed by the $\gamma$-graph dimension rather than the $\gamma$-OIG dimension. This raises a natural question: does there exist a simple and practical aggregation algorithm that achieves sample complexity $O(d_\gamma/\varepsilon)$, matching the lower bounds of Theorem 3.5 and Theorem 3.8 up to constant factors? Such an algorithm would witness the tightness of the lower bounds and establish optimality among all such aggregation procedures.

Our next result answers this question affirmatively. We show that a remarkably simple algorithm, taking the pointwise median of just three interpolators trained on independent samples, achieves optimality among all such aggregation procedures.

**Theorem 3.11.** *Let $\gamma \in (0,1)$, let $\mathcal{H}$ be a hypothesis class with $\gamma$-graph dimension $d_\gamma$, and let $\mathcal{D}$ be a distribution over $\mathcal{X} \times [0,1]$ realizable by $\mathcal{H}$. Let $\mathcal{A}$ be any interpolator, and let $\mathbf{S}_1, \mathbf{S}_2, \mathbf{S}_3 \sim \mathcal{D}^n$ be three independent samples. Define the median-of-three interpolator aggregation algorithm $M$ as*

$$M(x) = median(\mathcal{A}(\mathbf{S}_1)(x), \mathcal{A}(\mathbf{S}_2)(x), \mathcal{A}(\mathbf{S}_3)(x)).$$

*Then*

$$\mathop{\mathbb{E}}_{\mathbf{S}_1,\mathbf{S}_2,\mathbf{S}_3 \sim \mathcal{D}^n}[\mathcal{L}_\mathcal{D}^\gamma(M)] = O(d_\gamma/n).$$

Theorem 3.11 shows that the simple median-of-three interpolator aggregation algorithm achieves expected cutoff loss scaling as $O(d_\gamma/n)$. Setting this equal to $\varepsilon$ and solving for $n$ yields a sample complexity of $O(d_\gamma/\varepsilon)$ to achieve expected cutoff loss $\varepsilon$. This matches the lower bounds of Theorem 3.5 and Theorem 3.8 up to constant factors, establishing optimality of this algorithm and the tightness of the lower bounds.

A result similar to Theorem 3.11 was first established by (Aden-Ali et al., 2024) in the context of realizable binary classification. They showed that taking the pointwise majority vote of three interpolators achieves expected 0-1 loss $O(d/n)$, where $d$ is the VC dimension of the hypothesis class. This result was later extended to the weak-to-strong learning setting (Høgsgaard et al., 2024; Høgsgaard & Larsen, 2025) and the multiclass classification setting (Asilis et al., 2025b).

The proof of Theorem 3.11 is provided in Section B. We now provide a brief overview of the proof of Theorem 3.11.

**Proof overview Theorem 3.11:** The proof analyzes the error of the median-of-three interpolator aggregation algorithm by considering the event that the median prediction at a random point $\mathbf{x}$ differs from the true label $\mathbf{y}$ by more than

$\gamma$. This can happen only if at least two of the three interpolators make predictions that differ from the true label by more than $\gamma$. By a union bound, the expected $\gamma$-cutoff loss $\mathcal{L}_\mathcal{D}^\gamma(M)$ is at most 3 times the probability that two specific interpolators are both $\gamma$-far from the true label:

$$3 \mathop{\mathbb{E}}_{\mathbf{S}_1,\mathbf{S}_2 \sim \mathcal{D}^n} [\mathop{\mathbb{P}}_{\mathbf{x},\mathbf{y}}[|\mathcal{A}(\mathbf{S}_1)(\mathbf{x}) - \mathbf{y}| > \gamma, |\mathcal{A}(\mathbf{S}_2)(\mathbf{x}) - \mathbf{y}| > \gamma]],$$

where we have used that $\mathbf{S}_1, \mathbf{S}_2$, and $\mathbf{S}_3$ are identically distributed. Using the independence of $\mathbf{S}_1$ and $\mathbf{S}_2$, and exchanging the order of expectation and probability, we can rewrite this as $3 \mathbb{E}_{\mathbf{x},\mathbf{y}}[\mathbb{P}_{\mathbf{S} \sim \mathcal{D}^n}[|\mathcal{A}(\mathbf{S})(\mathbf{x}) - \mathbf{y}| > \gamma]^2]$. Partitioning $\mathcal{X} \times [0,1]$ into sets $R_i$ eqaul to $\{(x,y) : \mathbb{P}_{\mathbf{S} \sim \mathcal{D}^n}[|\mathcal{A}(\mathbf{S})(x) - y| > \gamma] \in (2^{-i}, 2^{-i+1}]\}$ we can upper bound the above expectation by

$$3 \textstyle\sum_{i=1}^\infty \mathbb{P}_{\mathbf{x},\mathbf{y}}[(\mathbf{x},\mathbf{y}) \in R_i] \cdot 2^{-2i+2}.$$

We show that $p_i := \mathbb{P}_{\mathbf{x},\mathbf{y}}[(\mathbf{x},\mathbf{y}) \in R_i]$ is $O(i^2 2^i d_\gamma/n)$, which is sufficiently fast to ensure that the sum converges to $O(d_\gamma/n)$. To bound $p_i$, let $\mathcal{D}_{R_i}$ denote the distribution over $\mathcal{X} \times [0,1]$ conditioned on $(\mathbf{x},\mathbf{y}) \in R_i$. By definition of $R_i$,

$$2^{-i} < \mathop{\mathbb{E}}_{\mathbf{x},\mathbf{y} \sim \mathcal{D}_{R_i}} [\mathop{\mathbb{P}}_{\mathbf{S} \sim \mathcal{D}^n}[|\mathcal{A}(\mathbf{S})(\mathbf{x}) - \mathbf{y}| > \gamma]].$$

Exchanging the order of expectation, this equals

$$\mathop{\mathbb{E}}_{\mathbf{S} \sim \mathcal{D}^n}[\mathcal{L}_{\mathcal{D}_{R_i}}^\gamma(\mathcal{A}(\mathbf{S}))].$$

Using an upper bound on the interpolator cutoff loss in terms of the $\gamma$-graph dimension, we can bound this quantity by $O(d_\gamma \ln((np_i)/d_\gamma)/(np_i))$, yielding the relation

$$2^{-i} = O(d_\gamma \ln((np_i)/d_\gamma)/(np_i)).$$

Solving for $p_i$ gives the upper bound

$$p_i = O(i^2 2^i d_\gamma/n).$$

The main novelty in this proof lies in a new error bound on the cutoff loss for the interpolator in terms of the $\gamma$-graph dimension, which is sharp enough to yield the desired bound on $p_i$. Specifically, we show that: for any hypothesis class with $\gamma$-graph dimension $d_\gamma$, interpolator $\mathcal{A}$, and distribution $\mathcal{D}$, with probability at least $1 - \delta$ over $\mathbf{S} \sim \mathcal{D}^n$,

$$\mathcal{L}_\mathcal{D}^\gamma(\mathcal{A}(\mathbf{S})) = O\big((d_\gamma \ln^2(n/d_\gamma) + \ln(1/\delta))/n\big).$$

This is a strict improvement over the previous best-known bound of Theorem 1 in (Attias et al., 2023), which established the same guarantee but with $\ln^2(n)$ instead of $\ln^2(n/d_\gamma)$. The difference may seem minor, but it is crucial for obtaining the desired bound on $p_i$ in the proof of Theorem 3.11 and obtaining optimality of the algorithm. Had we

used the previous bound from Theorem 1 of (Attias et al., 2023)[2], we would have obtained

$$2^{-i} = O(d_\gamma \ln^2(np_i)/(np_i)),$$

implying the weaker bound $p_i = O(2^i d_\gamma (\ln^2(d_\gamma) + i^2)/n)$, resulting in a superfluous $\ln^2(d_\gamma)$-factor in the bound, so being insufficient to ensure the optimal $O(d_\gamma/n)$ bound on the expected cutoff loss.

Improving the interpolator upper bound to depend on $\ln^2(n/d_\gamma)$ rather than $\ln^2(n)$ is therefore a key technical contribution of this work. This improvement requires refining a key step in the proof of (Attias et al., 2023), which uses a bound on the disambiguation size of a partial concept class from Theorem 12 of (Alon et al., 2021). We observe that an improved disambiguation size bound yields the improved interpolator bound stated above, and we provide such an improvement to Theorem 12 of (Alon et al., 2021).

Theorem 3.11, combined with Theorem 3.5 and Theorem 3.8, shows optimality of the median-of-three interpolator aggregation algorithm up to constant factors among a broad class of aggregation procedure, and tightness of the lower bounds. With the median-of-three interpolator algorithm being "close" to proper, a natural question arises: can proper learning algorithms, those that output a hypothesis from $\mathcal{H}$, also achieve this optimal sample complexity, or are aggregation algorithms (even simple) inherently more powerful?

Our final result answers this question by establishing a lower bound on the sample complexity of any proper learner in terms of the $\gamma$-graph dimension that is strictly larger than both the lower bounds in Theorem 3.5 and Theorem 3.8 and the matching upper bound in Theorem 3.11. Together, these bounds demonstrate that aggregation, even when combining only three interpolating hypotheses, can achieve strictly better sample complexity than any proper learning algorithm in the realizable regression setting.

**Theorem 3.12.** *For any $0 < \gamma < 1$, and $d_\gamma \geq 2$, there exists a hypothesis class $\mathcal{H}$ with $\gamma$-graph dimension $d_\gamma$ such that for any proper learning algorithm $\mathcal{A}$, and for any $0 < \varepsilon < \frac{1}{64e}$, there exists a realizable distribution $\mathcal{D}$ such that if $\mathcal{A}$ receives $n \leq \frac{d_\gamma}{32\varepsilon} \ln\left(\frac{1}{64e\varepsilon}\right)$ i.i.d. samples $\mathbf{S} \sim \mathcal{D}^n$ from $\mathcal{D}$, it holds that*

$$\underset{\mathbf{S}\sim\mathcal{D}^n}{\mathbb{E}}[\mathcal{L}_\mathcal{D}^\gamma(\mathcal{A}(\mathbf{S}))] \geq 4\varepsilon/3. \tag{5}$$

Theorem 3.12 shows that for any $\gamma$-graph dimension $d_\gamma$, there exist hypothesis classes for which any proper learning algorithm requires sample complexity at least

---

[2]Theorem 1 in (Attias et al., 2023) has a small typo in the statement from reading the proof carefully, the error decays as $O((d_\gamma \ln^2(n) + \ln(1/\delta))/n)$, which is what we stated above.

$\Omega(d_\gamma \log(1/\varepsilon)/\varepsilon)$ to achieve expected cutoff loss $\varepsilon$. This lower bound is strictly larger than both the lower bounds in Theorem 3.5 and Theorem 3.8, which are $\Omega(d_\gamma/\varepsilon)$, and the matching upper bound in Theorem 3.11, which is $O(d_\gamma/\varepsilon)$. Thus, aggregation, even when combining only three interpolating hypotheses, can achieve strictly better sample complexity than any proper learning algorithm in the realizable regression setting.

We now provide a brief overview of the proof of Theorem 3.12, with the full proof given in Section C.4.

**Proof overview Theorem 3.12:** The proof constructs a hypothesis class $\mathcal{H}$ with $\gamma$-graph dimension $d_\gamma$, over a "split" input space $\mathcal{X} = \bigcup_{k=d_\gamma-1}^{\infty} \mathcal{X}_k$, where each $\mathcal{X}_k = \{(k,x) : x \leq k\}$ for $k \geq d_\gamma - 1$. For each $k$ and $A \subseteq [k]$ with $|A| = d_\gamma - 1$, the hypothesis $h_{k,A}$ is defined to output 0 on points $(k,x)$, $x \in [k]\backslash A$ and a unique value $\gamma_{k,A} \in (\gamma, 1]$ elsewhere. The overall hypothesis class is $\mathcal{H} = \bigcup_{k=d_\gamma-1}^{\infty}\{h_{k,A} : A \subseteq [k], |A| = d_\gamma - 1\}$. To see that the $\gamma$-graph dimension is exactly $d_\gamma$, we first observe that the point set $\{(2d_\gamma, 1), \ldots, (2d_\gamma, d_\gamma)\}$ with witness hypothesis $h_{2d_\gamma,\{d_\gamma+2,\ldots,2d_\gamma\}}$ (which is all zero on the point set) is $\gamma$-graph shattered. This can be seen by for any subset $B \subseteq \{(2d_\gamma, 1), \ldots, (2d_\gamma, d_\gamma)\}$ of size at most $d_\gamma - 1$ there exists a hypothesis $h_{k,B\cup B'}$, with $B' \subseteq \{(2d_\gamma, d_\gamma + 1), \ldots, (2d_\gamma, 2d_\gamma)\}$ such that $|B\cup B'| = d_\gamma - 1$, such that if $(k,x) \in B$ then $h_{k,B\cup B'}(k,x) = \gamma_{k,B\cup B'} > \gamma$ so strictly more than $\gamma$ away from $h_{2d_\gamma,\{d_\gamma+1,\ldots,2d_\gamma\}}$ and else 0 implying that it is equal to $h_{2d_\gamma,\{d_\gamma+1,\ldots,2d_\gamma\}}$ outside $B$. Furthermore any hypothesis $h_{k,A}$ with $k \neq 2d_\gamma$ would be more than $\gamma$ away from $h_{2d_\gamma,\{d_\gamma+1,\ldots,2d_\gamma\}}$ on the whole point set. Concluding that the point set can be $\gamma$-graph shattered. Conversely, for any set of size $d_\gamma + 1$, any witness hypothesis $h_{k,A}$ outputting a unique value $\gamma_{k,A}$ on one of these points, call it $x'$, is preventing the existence of another hypothesis $h'$ that agrees with $h_{k,A}$ on this point $x'$ while being $\gamma$-far on another point, as the condition $h'(x') = h_{k,A}(x') = \gamma_{k,A}$ implies $h' = h_{k,A}$ by uniqueness of value $\gamma_{k,A}$ for the hypothesis $h_{k,A}$. Thus, the only other possible witness hypotheses $h_{k,A}$ are all zero on the $d_\gamma + 1$ points, which implies that all the points $(k_1, x_1), \ldots, (k_{d_\gamma+1}, x_{d_\gamma+1})$ satisfy $k = k_1 = \ldots = k_{d_\gamma+1}$. However, for this witness it is not possible to find a hypothesis that differs on $d_\gamma$ points and agrees on the last one, since any hypothesis $h_{k',A'}$ with $k' = k$ is strictly larger than $\gamma$ on only $|A'| = d_\gamma - 1$ points in $(k, 1), \ldots, (k, k)$, so it can differ from $h_{k,A}$ on at most $d_\gamma - 1$ points, and $h_{k',A'}$ with $k' \neq k$ is strictly larger than $\gamma$ for all points in $(k, 1), \ldots, (k, k)$. Thus, we conclude that no set of $d_\gamma + 1$ points can be $\gamma$-graph shattered, and hence the $\gamma$-graph dimension is exactly $d_\gamma$ as claimed.

To construct the hard distribution $\mathcal{D}$, we first carefully choose $k_u$, the effective universe size, sufficiently large relative to $\varepsilon$, namely $k_u = \Theta(d_\gamma/\varepsilon)$. For any $A \subseteq [k_u]$

with $|A| = k_u - d_\gamma + 1$, we define a realizable distribution $\mathcal{D}_A$ that assigns uniform mass $\Theta(1/k_u) = \Theta(\varepsilon/d_\gamma)$ to points in $\{(k_u, i) : i \in A\}$ and zero mass elsewhere, with the points being labelled 0. That is, the distribution only reveals zeros to the learner and is realizable by $h_{k_u, A^c}$, where $A^c = [k_u] \backslash A$. The hard distribution $\mathcal{D}_\mathbf{A}$ is then obtained by drawing $\mathbf{A}$ uniformly at random from all such sets. Again, as in the proof overview of Theorem 3.8, we may view the sample as generated from a random vector enumerating $\mathbf{A}$ together with an independent sequence of sampled indices. The first observation is that by a coupon collector argument, after drawing $O(d_\gamma \ln(1/\varepsilon)/\varepsilon)$ samples from the distribution $\mathcal{D}_\mathbf{A}$, with constant probability at least $\Omega(d_\gamma)$ points from $\{(k_u, i) : i \in \mathbf{A}\}$ are missing from the training sequence $\mathbf{S}$. Conditioned on $\mathbf{S}$, these points can "intuitively" be thought of as distributed uniformly at random over the points not in $\mathbf{S}$. Furthermore, given $\mathbf{S}$, the proper learning algorithm $\mathcal{A}$ has selected a hypothesis $h_{k_u, A'}$ for some $A' \subseteq [k_u]$ with $|A'| = d_\gamma - 1$. Since at least $\Omega(d_\gamma)$ points are missing from $\mathbf{S}$, we can show that with constant probability, at least $\Omega(d_\gamma)$ of these missing points lie in $A'$, meaning that the hypothesis $h_{k_u, A'}$ outputs a value strictly larger than $\gamma$ on these points and thus misclassifies them. Since each of these points has mass $\Theta(\varepsilon/d_\gamma)$ under $\mathcal{D}_\mathbf{A}$, this yields a total loss of at least $\Omega(d_\gamma) \cdot \Theta(\varepsilon/d_\gamma) = \Omega(\varepsilon)$ as claimed.

The idea in this proof, using a coupon collector argument to show that proper learners incur an extra $\ln(1/\varepsilon)$ factor in their sample complexity, originates from (Auer & Ortner, 2007) in the context of binary classification. However, their result applied only to a specific worst case proper learning algorithm, whereas our proof establishes the lower bound for any proper learning algorithm. The latter was also achieved for binary classification by (Bousquet et al., 2020).

## Acknowledgements

While this work was carried out, Mikael Møller Høgsgaard, Kasper Green Larsen and Liang-Yu Zou were supported by the European Union (ERC, TUCLA, 101125203). Views and opinions expressed are however those of the author(s) only and do not necessarily reflect those of the European Union or the European Research Council. Neither the European Union nor the granting authority can be held responsible for them. Furthermore, Mikael Møller Høgsgaard was supported by an Internationalisation Fellowship from the Carlsberg Foundation.

## Impact Statement

This paper presents work whose goal is to advance the field of Machine Learning. There are many potential societal consequences of our work, none which we feel must be specifically highlighted here.

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

# A. Definitions

In this section we collect the definitions used throughout the paper.

We define the truncated natural logarithm as $\mathrm{Ln}(x) := \max\{2, \ln(x)\}$.

For any set $\mathcal{X}$ we let $\mathcal{X}^* = \cup_{i=1}^{\infty} \mathcal{X}^i$, denote all finite sequences over $\mathcal{X}$.

For a vector or sequence $x = (x_1, \ldots, x_n)$ we write $\{x\} = \{x_1, \ldots, x_n\}$ for the set of the elements in the vector or sequence $x$.

We say that a training sequence $S = ((x_1, y_1), \ldots (x_n, y_n))$ is *realizable* by a hypothesis class $\mathcal{H}$, if there exists a hypothesis $h \in \mathcal{H}$ such that $h(x_i) = y_i$ for all $i = 1, \ldots, n$.

A distribution $\mathcal{D}$ over $\mathcal{X} \times [0, 1]$ is *realizable* by a hypothesis class $\mathcal{H} \subseteq [0, 1]^{\mathcal{X}}$ if there exists a hypothesis $h^\star \in \mathcal{H}$ such that $h^\star(x) = y$ for all $(x, y)$ in the support of $\mathcal{D}$.

For any $\gamma \in (0, 1)$, distribution $\mathcal{D}$ over $\mathcal{X} \times [0, 1]$, and hypothesis $h : \mathcal{X} \to [0, 1]$, the *cutoff loss* (or $\gamma$-*cutoff loss*) of $h$ with respect to $\mathcal{D}$ is defined as

$$\mathcal{L}_{\mathcal{D}}^{\gamma}(h) = \mathbb{P}_{(x,y)\sim\mathcal{D}}\big[|h(x) - y| > \gamma\big].$$

For three numbers $z_1, z_2, z_3$ we define $median(z_1, z_2, z_3)$ as the middle value among the three, with ties broken arbitrarily.

**Definition A.1** (Interpolator). An interpolator $\mathcal{A} : (\mathcal{X} \times [0, 1])^* \to \mathcal{H}$ for a hypothesis class $\mathcal{H}$ is a learning algorithm that, given a realizable training sequence $\mathbf{S} = ((x_1, y_1), \ldots, (x_n, y_n))$, returns a hypothesis $\mathcal{A}(\mathbf{S}) \in \mathcal{H}$ such that $\mathcal{A}(\mathbf{S})(x_i) = y_i$ for all $(x_i, y_i) \in \mathbf{S}$.

**Definition A.2** (Proper Learning Algorithm). A *proper learning algorithm* $\mathcal{A} : (\mathcal{X} \times [0, 1])^* \to \mathcal{H}$ for a hypothesis class $\mathcal{H} \subseteq [0, 1]^{\mathcal{X}}$ is a learning algorithm that, given a training sequence $\mathbf{S} = ((x_1, y_1), \ldots, (x_n, y_n))$, returns a hypothesis $\mathcal{A}(\mathbf{S}) \in \mathcal{H}$.

**Definition 3.1** ($\gamma$-graph shattering). A sequence $(x_1, \ldots, x_d) \in \mathcal{X}^d$ is $\gamma$-*graph shattered* by a hypothesis class $\mathcal{H} \subseteq [0, 1]^{\mathcal{X}}$ with witness $h \in \mathcal{H}$ if for any $b \in \{0, 1\}^d$ there exists a hypothesis $h_b \in \mathcal{H}$ such that:

- for all $i \in [d]$ with $b_i = 0$: $h_b(x_i) = h(x_i)$, and

- for all $i \in [d]$ with $b_i = 1$: $|h_b(x_i) - h(x_i)| > \gamma$.

**Definition 3.2** ($\gamma$-graph dimension). The $\gamma$-*graph dimension* of a hypothesis class $\mathcal{H} \subseteq [0, 1]^{\mathcal{X}}$, denoted $d_\gamma$, is the largest integer $d$ such that there exists a sequence $(x_1, \ldots, x_d) \in \mathcal{X}^d$ that is $\gamma$-graph shattered by $\mathcal{H}$ with some witness $h \in \mathcal{H}$.

**Definition 3.3** (Interpolator-Based Aggregation Algorithm). An *interpolator-based aggregation algorithm* $\mathcal{A}'$ is a mapping that takes as input an interpolator $\mathcal{A}$ (see Definition A.1) and a training sequence $S$ to produce a predictor, i.e., $\mathcal{A}' : ((\mathcal{X} \times [0, 1])^* \to [0, 1]^{\mathcal{X}}) \times (\mathcal{X} \times [0, 1])^* \to [0, 1]^{\mathcal{X}}$. Specifically, given $S$ and $\mathcal{A}$, the algorithm $\mathcal{A}'(S, \mathcal{A})$ proceeds as follows:

1. Construct sub-training sequences $S_1, \ldots, S_m$ from $S$, where $m$ may depend on $S$ and each $(x, y) \in S_j$ satisfies $(x, y) \in S$. The sub-sequences need not be disjoint.

2. Apply $\mathcal{A}$ to each sub-sequence to obtain hypotheses $h_1, \ldots, h_m$.

3. Return the predictor $x \mapsto r(h_1(x), \ldots, h_m(x), x)$, where $r : [0, 1]^* \times \mathcal{X} \to [0, 1]$ is an aggregation rule.

When $\mathcal{A}$ is clear from context, we write $\mathcal{A}'(S) = \mathcal{A}'(S, \mathcal{A})$.

**Definition 3.4** (Proper Aggregation Rule). An aggregation rule $r : [0, 1]^* \times \mathcal{X} \to [0, 1]$ is *proper* if for every sequence $z_1, \ldots, z_m \in [0, 1]$ and every $x \in \mathcal{X}$, we have $r(z_1, \ldots, z_m, x) \in \{z_1, \ldots, z_m\}$.

**Definition 3.6** (Finite Aggregation Algorithm). Let $\mathcal{H} \subseteq [0, 1]^{\mathcal{X}}$ be a hypothesis class. A *finite aggregation algorithm* $\mathcal{A}' : (\mathcal{X} \times [0, 1])^* \mapsto [0, 1]^{\mathcal{X}}$ for $\mathcal{H}$ is defined as follows: Given a training sequence $S$, of size $n \in \mathbb{N}$ it selects at most $m \leq m(n) < \infty$ hypotheses $h_1, \ldots, h_m$ from a hypothesis class $\mathcal{H}$, and combines them with an aggregation rule $r : [0, 1]^* \times \mathcal{X} \to [0, 1]$ and returns $x \mapsto r(h_1(x), \ldots, h_m(x), x)$.

**Definition 3.7** (Interpolating Aggregation Rule). An aggregation rule $r : [0, 1]^* \times \mathcal{X} \to [0, 1]$ is called *interpolating* if for any sequence $z_1, \ldots, z_m \in [0, 1]$ and any $x \in \mathcal{X}$, it satisfies $r(z_1, \ldots, z_m, x) \in [\min_i z_i, \max_i z_i]$.

**Definition 3.9** ($\gamma$-OIG dimension). Let $\mathcal{H} \subseteq [0,1]^{\mathcal{X}}$ be a hypothesis class and let $S = \{x_1, \ldots, x_n\} \subseteq \mathcal{X}$ be a finite set. The *one-inclusion graph* induced by $S$ and $\mathcal{H}$ is the hypergraph with vertex set $V_n = \mathcal{H}|_S = \{(h(x_1), \ldots, h(x_n)) : h \in \mathcal{H}\}$ and hyperedge set $E_n$ where a hyperedge $(f, i)$ contains $v \in V$ if $v(x_j) = f(x_j)$ for all $j \in [n] \setminus \{i\}$.

The $\gamma$-*OIG dimension* of $\mathcal{H}$ is the largest integer $k$ such that there exists a finite set $S \subseteq \mathcal{X}^k$ with $|S| = k$, such that there exists a finite subgraph $(V, E)$ of the one-inclusion graph $(V_n, E_n)$ induced by $S$, for which any orientation $\sigma : E_n \mapsto V_n$, has out-degree $\max_{v \in V} |\{i \in [n] : |\sigma(e_{v,i}) - v_i| > \gamma\}|$ strictly more than $n/3$.

**Definition A.3** (Partial Concept Class). A *partial concept class* over a domain $\mathcal{X}$ is a set $\mathcal{H} \subseteq \{0, 1, *\}^{\mathcal{X}}$, where $*$ represents an undefined label. Its VC-dimension is defined as the largest integer $d$ such that there exists a set of $d$ points in $\mathcal{X}$ that is shattered by $\mathcal{H}$, where a set $\{x_1, \ldots, x_d\}$ is shattered by a partial concept class if for every $b_1, \ldots, b_d \in \{0, 1\}$, there exists $h \in \mathcal{H}$ such that $h(x_i) = b_i$ for all $i = 1, \ldots, d$.

**Definition A.4** (Disambiguation). Let $\mathcal{H} \subseteq \{0, 1, *\}^{\mathcal{X}}$ be a partial concept class. A total concept class $\bar{\mathcal{H}} \subseteq \{0, 1\}^{\mathcal{X}}$ is called a *disambiguation* of $\mathcal{H}$ if for every $h \in \mathcal{H}$ there exists $\bar{h} \in \bar{\mathcal{H}}$ such that

$$\bar{h}(x) = h(x) \quad \text{for all } x \in \mathcal{X} \text{ with } h(x) \neq *.$$

# B. Upper Bounds

## B.1. Proof of Theorem 3.11

In this section, we present the proof of Theorem 3.11, which provides an upper bound on the sample complexity of the median-of-three algorithm in the expectation regime. The argument adapted ideas to the regression setting from the work of (Aden-Ali et al., 2024), which showed how combining three empirical risk minimizers in realizable binary classification to get optimal sample complexity (in expectation). To the end of using the ideas of (Aden-Ali et al., 2024), it is important, as described in the Section 3, to have a sufficiently strong bound on the cutoff loss of a single interpolator. The previous bounds on the cutoff loss for a single interpolator by (Attias et al., 2023) is not sufficient for our prove, and thus it is also crucial that we in this section improve the upper bound for the cutoff loss for a single interpolator in the realizable regression setting, stated in Lemma B.2. Now we restate Theorem 3.11 for convenience.

**Theorem 3.11.** *Let $\gamma \in (0, 1)$, let $\mathcal{H}$ be a hypothesis class with $\gamma$-graph dimension $d_\gamma$, and let $\mathcal{D}$ be a distribution over $\mathcal{X} \times [0, 1]$ realizable by $\mathcal{H}$. Let $\mathcal{A}$ be any interpolator, and let $\mathbf{S}_1, \mathbf{S}_2, \mathbf{S}_3 \sim \mathcal{D}^n$ be three independent samples. Define the median-of-three interpolator aggregation algorithm $M$ as*

$$M(x) = median(\mathcal{A}(\mathbf{S}_1)(x), \mathcal{A}(\mathbf{S}_2)(x), \mathcal{A}(\mathbf{S}_3)(x)).$$

*Then*

$$\mathop{\mathbb{E}}_{\mathbf{S}_1, \mathbf{S}_2, \mathbf{S}_3 \sim \mathcal{D}^n}[\mathcal{L}_\mathcal{D}^\gamma(M)] = O(d_\gamma / n).$$

To prove Theorem 3.11, we first assume the following lemma, which we later prove.

**Lemma B.1.** *Let $\gamma \in (0, 1)$, let $\mathcal{H}$ be a hypothesis class with $\gamma$-graph dimension $d_\gamma$, and let $\mathcal{D}$ be a distribution over $\mathcal{X} \times [0, 1]$ realizable by $\mathcal{H}$. Let $\mathcal{A}$ be any interpolator, and let $\mathbf{S} \sim \mathcal{D}^n$ be a sample of size $n$. Let $R$ be a subset of $\mathcal{X} \times [0, 1]$ with $\mathbb{P}_{\mathbf{x}, \mathbf{y}}[(\mathbf{x}, \mathbf{y}) \in R] \neq 0$. Let $\mathcal{D}_R$ be the conditional distribution of $\mathcal{D}$ over $R$. Then*

$$\mathop{\mathbb{E}}_{\mathbf{S} \sim \mathcal{D}^n}[\mathcal{L}_{\mathcal{D}_R}^\gamma(\mathcal{A}(\mathbf{S}))] \leq 42 \frac{d_\gamma \mathrm{Ln}^2(en\,\mathbb{P}_{\mathbf{x}, \mathbf{y}}[(\mathbf{x}, \mathbf{y}) \in R]/d_\gamma)}{n\,\mathbb{P}_{\mathbf{x}, \mathbf{y}}[(\mathbf{x}, \mathbf{y}) \in R]},$$

*where $\mathrm{Ln}(x) := \max\{2, \ln(x)\}$.*

We now prove Theorem 3.11 using Lemma B.1.

*Proof of Theorem 3.11.* We observe that the median-of-three aggregator $M$ makes an error on a point $(\mathbf{x}, \mathbf{y})$ only if at least two of the three hypotheses $\mathcal{A}(\mathbf{S}_1)$, $\mathcal{A}(\mathbf{S}_2)$, and $\mathcal{A}(\mathbf{S}_3)$ make an error on $(\mathbf{x}, \mathbf{y})$. Besides, the three samples $\mathbf{S}_1$, $\mathbf{S}_2$, and $\mathbf{S}_3$ are independent and identically distributed, which implies that for any pair $\mathbf{S}_i, \mathbf{S}_j$ with $i < j$, the probability that both hypotheses make an error on $(\mathbf{x}, \mathbf{y})$ is the same. Therefore, we have

$$\mathop{\mathbb{E}}_{\mathbf{S}_1, \mathbf{S}_2, \mathbf{S}_3 \sim \mathcal{D}^n}[\mathcal{L}_\mathcal{D}^\gamma(M)] = \mathop{\mathbb{E}}_{\mathbf{S}_1, \mathbf{S}_2, \mathbf{S}_3 \sim \mathcal{D}^n}\left[\mathop{\mathbb{P}}_{\mathbf{x}, \mathbf{y}}[|M(\mathbf{x}) - \mathbf{y}| > \gamma]\right]$$

$$\leq \sum_{1 \leq i < j \leq 3} \mathop{\mathbb{E}}_{\mathbf{S}_i, \mathbf{S}_j \sim \mathcal{D}^n}\left[\mathop{\mathbb{P}}_{\mathbf{x}, \mathbf{y}}[|\mathcal{A}(\mathbf{S}_i)(\mathbf{x}) - \mathbf{y}| > \gamma, |\mathcal{A}(\mathbf{S}_j)(\mathbf{x}) - \mathbf{y}| > \gamma]\right]$$

$$\leq \binom{3}{2} \mathop{\mathbb{E}}_{\mathbf{S}_1, \mathbf{S}_2 \sim \mathcal{D}^n}\left[\mathop{\mathbb{P}}_{\mathbf{x}, \mathbf{y}}[|\mathcal{A}(\mathbf{S}_1)(\mathbf{x}) - \mathbf{y}| > \gamma, |\mathcal{A}(\mathbf{S}_2)(\mathbf{x}) - \mathbf{y}| > \gamma]\right]$$

Next, we change the order of expectation and use the independence between $\mathbf{S}_1$ and $\mathbf{S}_2$ to obtain

$$\mathop{\mathbb{E}}_{\mathbf{S}_1, \mathbf{S}_2, \mathbf{S}_3 \sim \mathcal{D}^n}[\mathcal{L}_\mathcal{D}^\gamma(M)] \leq 3 \mathop{\mathbb{E}}_{\mathbf{x}, \mathbf{y}}\left[\mathop{\mathbb{P}}_{\mathbf{S}_1, \mathbf{S}_2 \sim \mathcal{D}^n}[|\mathcal{A}(\mathbf{S}_1)(\mathbf{x}) - \mathbf{y}| > \gamma, |\mathcal{A}(\mathbf{S}_2)(\mathbf{x}) - \mathbf{y}| > \gamma]\right]$$

$$= 3 \mathop{\mathbb{E}}_{\mathbf{x}, \mathbf{y}}\left[(\mathop{\mathbb{P}}_{\mathbf{S} \sim \mathcal{D}^n}[|\mathcal{A}(\mathbf{S})(\mathbf{x}) - \mathbf{y}| > \gamma])^2\right]$$

For any $(x, y) \in \mathcal{X} \times [0, 1]$, define $p_{x,y} = \mathbb{P}_{\mathbf{S} \sim \mathcal{D}^n}[|\mathcal{A}(\mathbf{S})(x) - y| > \gamma]$. Then we partition $\mathcal{X} \times [0, 1]$ into sets

$$R_i = \{(x, y) \in \mathcal{X} \times [0, 1] : p_{x,y} \in (2^{-i}, 2^{-i+1}]\},$$

where $i \in \mathbb{N}$. By the law of total probability, we have

$$\mathop{\mathbb{E}}_{\mathbf{S}_1,\mathbf{S}_2,\mathbf{S}_3\sim\mathcal{D}^n}[\mathcal{L}_{\mathcal{D}}^{\gamma}(M)] \le 3 \mathop{\mathbb{E}}_{\mathbf{x},\mathbf{y}}[p_{\mathbf{x},\mathbf{y}}^2]$$

$$= 3\sum_{i=1}^{\infty} \mathop{\mathbb{E}}_{\mathbf{x},\mathbf{y}}\left[p_{\mathbf{x},\mathbf{y}}^2|(\mathbf{x},\mathbf{y})\in R_i\right]\mathop{\mathbb{P}}_{\mathbf{x},\mathbf{y}}[(\mathbf{x},\mathbf{y})\in R_i]$$

$$\le 3\sum_{i=1}^{\infty} 2^{-2i+2}\mathop{\mathbb{P}}_{\mathbf{x},\mathbf{y}}[(\mathbf{x},\mathbf{y})\in R_i]$$

$$= 12\sum_{i=1}^{\infty} 2^{-2i}\mathop{\mathbb{P}}_{\mathbf{x},\mathbf{y}}[(\mathbf{x},\mathbf{y})\in R_i]$$

Suppose for any integer $i \ge 1$, $\mathbb{P}_{\mathbf{x},\mathbf{y}}[(\mathbf{x},\mathbf{y})\in R_i] \le C_1 \cdot \frac{d_{\gamma}i^2 2^i}{n}$, where $C_1 > e^2$ is a universal constant. Then,

$$\mathop{\mathbb{E}}_{\mathbf{S}_1,\mathbf{S}_2,\mathbf{S}_3\sim\mathcal{D}^n}[\mathcal{L}_{\mathcal{D}}^{\gamma}(M)] \le 12\sum_{i=1}^{\infty} 2^{-2i}\cdot C_1\cdot\frac{d_{\gamma}i^2 2^i}{n}$$

$$\le \frac{12C_1 d_{\gamma}}{n}\sum_{i=1}^{\infty}\frac{i^2}{2^i}$$

$$= \frac{72C_1 d_{\gamma}}{n},$$

which would complete the proof. What remains is to prove that there exists a universal constant $C_1 > e^2$ such that for any $i \ge 1$,

$$\mathop{\mathbb{P}}_{\mathbf{x},\mathbf{y}}[(\mathbf{x},\mathbf{y})\in R_i] \le C_1\cdot\frac{d_{\gamma}i^2 2^i}{n}.$$

We prove this by contradiction. Suppose that there exists some $i$ such that $\mathbb{P}_{\mathbf{x},\mathbf{y}}[(\mathbf{x},\mathbf{y})\in R_i] > C_1\cdot\frac{d_{\gamma}i^2 2^i}{n}$. Now, let $\mathcal{D}_{R_i}$ be the conditional distribution of $\mathcal{D}$ over $R_i$. By the definition of $R_i$, we have

$$\mathop{\mathbb{E}}_{(\mathbf{x},\mathbf{y})\sim\mathcal{D}_{R_i}}\left[\mathop{\mathbb{P}}_{\mathbf{S}\sim\mathcal{D}^n}[|\mathcal{A}(\mathbf{S})(\mathbf{x})-\mathbf{y}|>\gamma]\right] = \mathop{\mathbb{E}}_{(\mathbf{x},\mathbf{y})\sim\mathcal{D}_{R_i}}[p_{\mathbf{x},\mathbf{y}}] > 2^{-i}. \tag{6}$$

On the other hand, exchanging the order of the expectation yields

$$\mathop{\mathbb{E}}_{(\mathbf{x},\mathbf{y})\sim\mathcal{D}_{R_i}}\left[\mathop{\mathbb{P}}_{\mathbf{S}\sim\mathcal{D}^n}[|\mathcal{A}(\mathbf{S})(\mathbf{x})-\mathbf{y}|>\gamma]\right] = \mathop{\mathbb{E}}_{\mathbf{S}\sim\mathcal{D}^n}\left[\mathop{\mathbb{P}}_{(\mathbf{x},\mathbf{y})\sim\mathcal{D}_{R_i}}[|\mathcal{A}(\mathbf{S})(\mathbf{x})-\mathbf{y}|>\gamma]\right] = \mathop{\mathbb{E}}_{\mathbf{S}\sim\mathcal{D}^n}\left[\mathcal{L}_{\mathcal{D}_{R_i}}^{\gamma}(\mathcal{A}(\mathbf{S}))\right].$$

Now we can apply Lemma B.1. What's more, using the fact that the function $x \mapsto \mathrm{Ln}^2(ex)/x$ is decreasing for $x > 0$ together with the bound $n\,\mathbb{P}_{\mathbf{x},\mathbf{y}}[(\mathbf{x},\mathbf{y})\in R_i]/d_{\gamma} > C_1 i^2 2^i$, we obtain

$$\mathop{\mathbb{E}}_{\mathbf{S}}\left[\mathcal{L}_{\mathcal{D}_{R_i}}^{\gamma}(\mathcal{A}(\mathbf{S}))\right] \le \frac{42 d_{\gamma}\mathrm{Ln}^2\left(en\,\mathbb{P}_{\mathbf{x},\mathbf{y}}[(\mathbf{x},\mathbf{y})\in R_i]/d_{\gamma}\right)}{n\,\mathbb{P}_{\mathbf{x},\mathbf{y}}[(\mathbf{x},\mathbf{y})\in R_i]} \le \frac{42\mathrm{Ln}^2\left(C_1 ei^2 2^i\right)}{C_1 i^2 2^i}. \tag{7}$$

Combining the lower bound (6) and the upper bound (7), we obtain

$$2^{-i} \le \frac{42\mathrm{Ln}^2\left(C_1 ei^2 2^i\right)}{C_1 i^2 2^i}.$$

Multiplying by $2^i$ on both sides, using that $\mathrm{Ln}(C_1 ei^2 2^i) = \ln(C_1 ei^2 2^i) = \mathrm{Ln}(C_1 ei^2) + i\ln(2)$, as $C_1 \ge e^2$ and $i \ge 1$, and that for $a,b > 0$ we have $(a+b)^2 \le 2(a^2+b^2)$ we get

$$1 \le \frac{42\left(\mathrm{Ln}(C_1 ei^2 2^i)\right)^2}{C_1 i^2} \le 84\cdot\left(\frac{\mathrm{Ln}^2(C_1 ei^2)}{C_1 i^2} + \frac{\ln^2(2)}{C_1}\right).$$

As $x \mapsto \mathrm{Ln}^2(ex)/x$ is decreasing for $x > 0$ and $C_1 i^2 \geq C_1$, we have

$$1 \leq 84 \cdot \left( \frac{\mathrm{Ln}^2(C_1 e)}{C_1} + \frac{\ln^2(2)}{C_1} \right),$$

where the right-hand side is strictly less than 1 for $C_1$ sufficiently large, yielding a contradiction. $\square$

What remains is to prove Lemma B.1. To this end, we provide the following lemma which presents a tighter bound on the cutoff loss for a single interpolator compared to Theorem 1 of (Attias et al., 2023).

**Lemma B.2.** *Let $\gamma \in (0,1)$. Let $\mathcal{H}$ be a hypothesis class with $\gamma$-graph dimension $d_\gamma$. Let $\mathcal{D}$ be a distribution over $\mathcal{X} \times [0,1]$ that is realizable by $\mathcal{H}$. Let $\mathbf{S}$ be a sample of size $n$ drawn i.i.d. from $\mathcal{D}$. Then for any interpolator $\mathcal{A}$ and any $0 < \delta < 1$ it holds with probability at least $1 - \delta$ over $\mathbf{S}$ that*

$$\mathcal{L}_{\mathcal{D}}^\gamma(\mathcal{A}(\mathbf{S})) \leq 8 \frac{d_\gamma \mathrm{Ln}^2(2en/d_\gamma) + \ln(2/\delta)}{n}.$$

We defer the proof of Lemma B.2 to Section B.2 and now prove Lemma B.1. For convenience, we restate Lemma B.1 here.

**Lemma B.1.** *Let $\gamma \in (0,1)$, let $\mathcal{H}$ be a hypothesis class with $\gamma$-graph dimension $d_\gamma$, and let $\mathcal{D}$ be a distribution over $\mathcal{X} \times [0,1]$ realizable by $\mathcal{H}$. Let $\mathcal{A}$ be any interpolator, and let $\mathbf{S} \sim \mathcal{D}^n$ be a sample of size $n$. Let $R$ be a subset of $\mathcal{X} \times [0,1]$ with $\mathbb{P}_{\mathbf{x},\mathbf{y}}[(\mathbf{x},\mathbf{y}) \in R] \neq 0$. Let $\mathcal{D}_R$ be the conditional distribution of $\mathcal{D}$ over $R$. Then*

$$\mathbb{E}_{\mathbf{S} \sim \mathcal{D}^n}[\mathcal{L}_{\mathcal{D}_R}^\gamma(\mathcal{A}(\mathbf{S}))] \leq 42 \frac{d_\gamma \mathrm{Ln}^2(en\,\mathbb{P}_{\mathbf{x},\mathbf{y}}[(\mathbf{x},\mathbf{y}) \in R]/d_\gamma)}{n\,\mathbb{P}_{\mathbf{x},\mathbf{y}}[(\mathbf{x},\mathbf{y}) \in R]},$$

*where* $\mathrm{Ln}(x) := \max\{2, \ln(x)\}$.

*Proof of Lemma B.1.* First, consider the case $n\,\mathbb{P}_{\mathbf{x},\mathbf{y}}[(\mathbf{x},\mathbf{y}) \in R] \leq 4d_\gamma$. In this case, the claim follows immediately since $\mathcal{L}_{\mathcal{D}_R}^\gamma(\mathcal{A}(\mathbf{S})) \leq 1$, and by the definition of $\mathrm{Ln}(\cdot)$,

$$\frac{42 d_\gamma \mathrm{Ln}^2(en\,\mathbb{P}_{\mathbf{x},\mathbf{y}}[(\mathbf{x},\mathbf{y}) \in R]/d_\gamma)}{n\,\mathbb{P}_{\mathbf{x},\mathbf{y}}[(\mathbf{x},\mathbf{y}) \in R]} \geq \frac{168 d_\gamma}{n\,\mathbb{P}_{\mathbf{x},\mathbf{y}}[(\mathbf{x},\mathbf{y}) \in R]} > 1.$$

In the following, we assume that $n\,\mathbb{P}_{\mathbf{x},\mathbf{y}}[(\mathbf{x},\mathbf{y}) \in R] > 4d_\gamma$. For any $m \in \mathbb{N}$, define the event

$$E_m = \{|\{i \in [n] : (\mathbf{x}_i,\mathbf{y}_i) \in R\}| = m\},$$

where $(\mathbf{x}_i,\mathbf{y}_i)$ is the $i$-th point of $\mathbf{S}$. And let

$$E = \bigcup_{m \geq n\,\mathbb{P}_{\mathbf{x},\mathbf{y}}[(\mathbf{x},\mathbf{y})\in R]/2} E_m.$$

Define $X_i = \mathbb{I}\{(\mathbf{x}_i,\mathbf{y}_i) \in R\}$ for $i \in [n]$ and $X = \sum_i X_i$. Then $\mathbb{E}_{\mathbf{S}}[X] = n\,\mathbb{P}_{\mathbf{x},\mathbf{y}}[(\mathbf{x},\mathbf{y}) \in R]$ and $\mathbb{P}_{\mathbf{S}}[E] = \mathbb{P}[X \geq \mathbb{E}[X]/2]$. By a Chernoff bound, we obtain

$$\mathbb{P}_{\mathbf{S}}[\bar{E}] = \mathbb{P}_{\mathbf{S}}[X < \mathbb{E}[X]/2] \leq \exp\left(-\frac{n\,\mathbb{P}_{\mathbf{x},\mathbf{y}}[(\mathbf{x},\mathbf{y}) \in R]}{8}\right) \leq \frac{8}{n\,\mathbb{P}_{\mathbf{x},\mathbf{y}}[(\mathbf{x},\mathbf{y}) \in R]},$$

where the last inequality uses the fact that $e^{-x} \leq \frac{1}{x}$ for any $x > 0$. By the law of total probability,

$$\mathbb{E}_{\mathbf{S}}\left[\mathcal{L}_{\mathcal{D}_R}^\gamma(\mathcal{A}(\mathbf{S}))\right] = \sum_{m \geq n\,\mathbb{P}_{\mathbf{x},\mathbf{y}}[(\mathbf{x},\mathbf{y})\in R]/2} \mathbb{E}_{\mathbf{S}}\left[\mathcal{L}_{\mathcal{D}_R}^\gamma(\mathcal{A}(\mathbf{S})) \mid E_m\right] \cdot \mathbb{P}_{\mathbf{S}}[E_m] + \mathbb{E}_{\mathbf{S}}\left[\mathcal{L}_{\mathcal{D}_R}^\gamma(\mathcal{A}(\mathbf{S})) \mid \bar{E}\right] \cdot \mathbb{P}_{\mathbf{S}}[\bar{E}]$$

$$\leq \sum_{m \geq n\,\mathbb{P}_{\mathbf{x},\mathbf{y}}[(\mathbf{x},\mathbf{y})\in R]/2} \mathbb{E}_{\mathbf{S}}\left[\mathcal{L}_{\mathcal{D}_R}^\gamma(\mathcal{A}(\mathbf{S})) \mid E_m\right] \cdot \mathbb{P}_{\mathbf{S}}[E_m] + \frac{8}{n\,\mathbb{P}_{\mathbf{x},\mathbf{y}}[(\mathbf{x},\mathbf{y}) \in R]}. \tag{8}$$

By the above, it suffices to show that for any $m \geq n\,\mathbb{P}_{\mathbf{x},\mathbf{y}}[(\mathbf{x},\mathbf{y}) \in R]/2$,

$$\mathop{\mathbb{E}}_{\mathbf{S}}\left[\mathcal{L}_{\mathcal{D}_R}^\gamma(\mathcal{A}(\mathbf{S})) \mid E_m\right] \leq \frac{34 d_\gamma \mathrm{Ln}^2\left(en\,\mathbb{P}_{\mathbf{x},\mathbf{y}}[(\mathbf{x},\mathbf{y}) \in R]/d_\gamma\right)}{n\,\mathbb{P}_{\mathbf{x},\mathbf{y}}[(\mathbf{x},\mathbf{y}) \in R]}. \tag{9}$$

We now prove Inequality (9). By the non-negativity of $\mathcal{L}_{\mathcal{D}_R}^\gamma(\mathcal{A}(\mathbf{S}))$,

$$\begin{aligned}
\mathop{\mathbb{E}}_{\mathbf{S}}\left[\mathcal{L}_{\mathcal{D}_R}^\gamma(\mathcal{A}(\mathbf{S})) \mid E_m\right] &= \int_0^\infty \mathop{\mathbb{P}}_{\mathbf{S}}\left[\mathcal{L}_{\mathcal{D}_R}^\gamma(\mathcal{A}(\mathbf{S})) > x \mid E_m\right] dx \\
&\leq \frac{16 d_\gamma \mathrm{Ln}^2(em/d_\gamma)}{m} + \int_{16 d_\gamma \mathrm{Ln}^2(em/d_\gamma)/m}^1 \mathop{\mathbb{P}}_{\mathbf{S}}\left[\mathcal{L}_{\mathcal{D}_R}^\gamma(\mathcal{A}(\mathbf{S})) > x \mid E_m\right] dx.
\end{aligned} \tag{10}$$

Conditioned on $E_m$, if we consider only these $m$ samples, $\mathcal{A}(\mathbf{S})$ can be viewed as the returned hypothesis of an interpolator trained on $m$ i.i.d. samples from $\mathcal{D}_R$. Thus, we can apply Lemma B.2 to control its error with respect to $\mathcal{D}_R$. For simplicity, we use the following weaker bound: with probability at least $1 - \delta$,

$$\mathcal{L}_{\mathcal{D}_R}^\gamma(\mathcal{A}(\mathbf{S})) \leq 16 \max\left\{\frac{d_\gamma \mathrm{Ln}^2(2em/d_\gamma)}{m}, \frac{\mathrm{Ln}(2/\delta)}{m}\right\}.$$

For any $x \geq 16 d_\gamma \mathrm{Ln}^2(2em/d_\gamma)/m$, setting $\delta(x) = 2e^{-mx/16}$, we have $\ln(2/\delta(x)) = mx/16 \geq 2$, which implies that $\frac{16\mathrm{Ln}(2/\delta(x))}{m} = \frac{16\ln(2/\delta(x))}{m} = x$, and thus

$$\begin{aligned}
\mathop{\mathbb{P}}_{\mathbf{S}}\left[\mathcal{L}_{\mathcal{D}_R}^\gamma(\mathcal{A}(\mathbf{S})) > x \mid E_m\right] &= \mathop{\mathbb{P}}_{\mathbf{S}}\left[\mathcal{L}_{\mathcal{D}_R}^\gamma(\mathcal{A}(\mathbf{S})) > \frac{16\mathrm{Ln}(2/\delta(x))}{m} \mid E_m\right] \\
&= \mathop{\mathbb{P}}_{\mathbf{S}}\left[\mathcal{L}_{\mathcal{D}_R}^\gamma(\mathcal{A}(\mathbf{S})) > 16 \max\left\{\frac{d_\gamma \mathrm{Ln}^2(2em/d_\gamma)}{m}, \frac{\mathrm{Ln}(2/\delta(x))}{m}\right\} \mid E_m\right]
\end{aligned}$$

where the last equality uses our choice of $x$. This is bounded by the weaker error bound stated above, and hence the probability is at most $\delta(x)$. Therefore,

$$\begin{aligned}
\int_{16 d_\gamma \mathrm{Ln}^2(2em/d_\gamma)/m}^1 \mathop{\mathbb{P}}_{\mathbf{S}}\left[\mathcal{L}_{\mathcal{D}_R}^\gamma(\mathcal{A}(\mathbf{S})) > x \mid E_m\right] dx &\leq \int_{16 d_\gamma \mathrm{Ln}^2(em/d_\gamma)/m}^1 \delta(x) dx \\
&= \int_{16 d_\gamma \mathrm{Ln}^2(2em/d_\gamma)/m}^1 2e^{-mx/16} dx \\
&\leq \frac{32 e^{-d_\gamma \mathrm{Ln}^2(2em/d_\gamma)}}{m} \\
&\leq \frac{d_\gamma \mathrm{Ln}^2(2em/d_\gamma)}{m}.
\end{aligned}$$

Here, the last inequality uses the fact that $d_\gamma \mathrm{Ln}^2(2em/d_\gamma) \geq 4$. Substituting this into Inequality (10) and using that the function $x \mapsto \mathrm{Ln}^2(ex)/x$ is decreasing for $x > 0$, together with $m \geq n\,\mathbb{P}_{\mathbf{x},\mathbf{y}}[(\mathbf{x},\mathbf{y}) \in R]/2 > 0$, we conclude that

$$\mathop{\mathbb{E}}_{\mathbf{S}}\left[\mathcal{L}_{\mathcal{D}_R}^\gamma(\mathcal{A}(\mathbf{S})) \mid E_m\right] \leq \frac{17 d_\gamma \mathrm{Ln}^2(2em/d_\gamma)}{m} < \frac{34 d_\gamma \mathrm{Ln}^2\left(en\,\mathbb{P}_{\mathbf{x},\mathbf{y}}[(\mathbf{x},\mathbf{y}) \in R]/d_\gamma\right)}{n\,\mathbb{P}_{\mathbf{x},\mathbf{y}}[(\mathbf{x},\mathbf{y}) \in R]},$$

as claimed.

$\square$

### B.2. Proof of Lemma B.2

The best known upper bound of the cutoff loss for single interpolator is given by Theorem 1[3] in (Attias et al., 2023). Informally speaking, it yields that: for any hypothesis class $\mathcal{H}$ with $\gamma$-graph dimension $d_\gamma$ and any distribution $\mathcal{D}$ over

---

[3]Theorem 1 in (Attias et al., 2023) has a small typo in the statement from reading the proof carefully, the error decays as $O((d_\gamma \ln^2(n) + \ln(1/\delta))/n)$, which is what we stated above.

$\mathcal{X} \times [0, 1]$ realized by $\mathcal{H}$, let $\mathbf{S}$ be an i.i.d. sample of size $n$ drawn from $\mathcal{D}$, then with probability at least $1 - \delta$, the cutoff loss of the hypothesis returned by interpolator $\mathcal{A}$ satisfies

$$\mathcal{L}_{\mathcal{D}}^{\gamma}(\mathcal{A}(\mathbf{S})) \leq O\left(\frac{d_{\gamma} \ln^2 n + \ln(1/\delta)}{n}\right).$$

In Lemma B.2, we improve the above bound by replacing the dependence on $\ln^2 n$ with $\text{Ln}^2(n/d_{\gamma})$. For convenience, we restate the lemma here.

**Lemma B.2.** *Let $\gamma \in (0, 1)$. Let $\mathcal{H}$ be a hypothesis class with $\gamma$-graph dimension $d_{\gamma}$. Let $\mathcal{D}$ be a distribution over $\mathcal{X} \times [0, 1]$ that is realizable by $\mathcal{H}$. Let $\mathbf{S}$ be a sample of size $n$ drawn i.i.d. from $\mathcal{D}$. Then for any interpolator $\mathcal{A}$ and any $0 < \delta < 1$ it holds with probability at least $1 - \delta$ over $\mathbf{S}$ that*

$$\mathcal{L}_{\mathcal{D}}^{\gamma}(\mathcal{A}(\mathbf{S})) \leq 8 \frac{d_{\gamma} \text{Ln}^2(2en/d_{\gamma}) + \ln(2/\delta)}{n}.$$

Our proof of Lemma B.2 follows the same structure as that of Theorem 1 in (Attias et al., 2023), which consists of three steps: symmetrization, permutation, and reduction to a finite class. Similar results are also established in the proof of Theorem 4.3 in (Anthony & Bartlett, 2009). The novelty of these steps is due to (Attias et al., 2023) and (Anthony & Bartlett, 2009). Our insight to the improvement comes from showing a tighter bound on the size of the finite hypothesis class reduced to. Specifically, we provide a tighter bound on the size of a disambiguation of a partial concept class (see Definition A.3 and Definition A.4), which controls the size of the finite hypothesis class reduced to. We, now state and prove the lemma regarding the size of a disambiguation.

**Lemma B.3.** *Let $\mathcal{H} \subseteq \{0, 1, *\}^{\mathcal{X}}$ be a partial concept class of VC-dimension $d \geq 1$, where $\mathcal{X}$ is a finite domain with $|\mathcal{X}| = n \geq 1$. Then there exists a disambiguation $\overline{\mathcal{H}}$ of $\mathcal{H}$ such that*

$$\ln |\overline{\mathcal{H}}| \leq 2d \, \text{Ln}^2\left(\frac{en}{d}\right).$$

Lemma B.3 strengthens Theorem 12 in (Alon et al., 2021), which proves that $\ln |\overline{\mathcal{H}}|$ is bounded by $O(d \ln^2 n)$. Our proof follows the original proof technique. However, through a more careful analysis, we use a sharper bound on the binomial sum, which leads to the improved result.

*Proof of Lemma B.3.* In the case that $n < d$, we can simply take the disambiguation $\overline{\mathcal{H}}$ as the set of all total concepts on $\mathcal{X}$, which gives $|\overline{\mathcal{H}}| \leq 2^n$. Since $n < d$, we have $\ln |\overline{\mathcal{H}}| \leq n \ln 2 < d \ln 2 < 2d \text{Ln}^2(en/d)$, concluding the proof in this case.

In the case that $d \leq n < 6d$, we still bound the size of disambiguation trivially as $2^n$. Notice that for $1 \leq x < 6$, it holds that $2 \ln^2(ex) > x \ln 2$. Setting $x = n/d$, we have $\ln |\overline{\mathcal{H}}| \leq n \ln 2 < 2d \ln^2(en/d)$, concluding the proof in this case.

In the following, we assume that $n \geq 6d$. Let $\mathcal{X} = \{x_1, \ldots, x_n\}$. For any partial concept class $\mathcal{H}' \subseteq \mathcal{H}$, define the shattering strength $s(\mathcal{H}')$ as the number of subsets of $\mathcal{X}$ that are shattered[4] by $\mathcal{H}'$, i.e.,

$$s(\mathcal{H}') = |\{S \subseteq \mathcal{X} : S \text{ is shattered by } \mathcal{H}'\}|.$$

Since $\mathcal{H}$ has VC-dimension $d$, the VC-dimension of $\mathcal{H}'$ is at most $d$, yielding $s(\mathcal{H}') \leq \binom{n}{\leq d}$. For $(x, y) \in \mathcal{X} \times \{0, 1\}$, define the restriction of $\mathcal{H}'$ on $(x, y)$ as

$$\mathcal{H}'_{(x,y)} = \{h' \in \mathcal{H}' : h'(x) = y\}.$$

Then, it holds that $s(\mathcal{H}') \geq s(\mathcal{H}'_{(x,0)}) + s(\mathcal{H}'_{(x,1)})$. This is because, for any $S \subseteq \mathcal{X}$, if $S$ is shattered by both $\mathcal{H}'_{(x,0)}$ and $\mathcal{H}'_{(x,1)}$, then $S$ and $S \cup \{x\}$ are shattered by $\mathcal{H}'$ (and $\mathcal{H}'_{(x,0)}$ and $\mathcal{H}'_{(x,1)}$ can not shatter $S \cup \{x\}$ so it is not double counted); if $S$ is shattered by only one of $\mathcal{H}'_{(x,0)}$ and $\mathcal{H}'_{(x,1)}$, then $S$ is also shattered by $\mathcal{H}'$; if $S$ isn't shattered by $\mathcal{H}'_{(x,0)}$ or $\mathcal{H}'_{(x,1)}$, then it is still possible that $\mathcal{H}'$ can shatter $S$.

---

[4]We remark that the empty set is also considered shattered as long as the hypothesis class is not the empty set - this implies that $s(\mathcal{H}) = 1$, means any point in $\mathcal{X}$ is either labelled as something with in $\{0, *\}$ or $\{*, 1\}$ for all hypothesis in $\mathcal{H}$. This further means that given $s(\mathcal{H}) = 1$, no more updates will be made by the following disambiguation algorithm. When the hypothesis class $\mathcal{H} = \emptyset$ is the empty set we define $s(\emptyset) = 0$.

Also, for $\mathcal{H}' \subseteq \mathcal{H}$ and $x \in \mathcal{X}$, let $M_{\mathcal{H}'}(x) \in \{0, 1\}$ be the label that maximizes the shattering strength of the restriction, i.e.,

$$M_{\mathcal{H}'}(x) = \underset{y \in \{0,1\}}{\mathrm{argmax}} \; s(\mathcal{H}'_{(x,y)}),$$

breaking ties always towards 1(or a arbitrarily way).

Now, we construct the disambiguation $\overline{\mathcal{H}}$ of $\mathcal{H}$. For each $h \in \mathcal{H}$, we write the entries of the disambiguation $\overline{h}$ iteratively. We start with $\mathcal{H}' = \mathcal{H}$, for $x$ moving from $x_1$ to $x_n$, if $h(x) \neq *$ and $h(x) = 1 - M_{\mathcal{H}'}(x)$, we set $\overline{h}(x) = h(x)$ and update $\mathcal{H}' = \mathcal{H}'_{(x,h(x))}$; otherwise, we set $\overline{h}(x) = M_{\mathcal{H}'}(x)$ and keep $\mathcal{H}'$ unchanged. We repeat the process until all entries are written.

Since $s(\mathcal{H}'_{(x,M_{\mathcal{H}'}(x))}) \geq s(\mathcal{H}'_{(x,1-M_{\mathcal{H}'}(x))})$ we have that

$$s(\mathcal{H}') \geq s(\mathcal{H}'_{(x,0)}) + s(\mathcal{H}'_{(x,1)}) = s(\mathcal{H}'_{(x,M_{\mathcal{H}'}(x))}) + s(\mathcal{H}'_{(x,1-M_{\mathcal{H}'}(x))}) \geq 2s(\mathcal{H}'_{(x,1-M_{\mathcal{H}'}(x))}),$$

implying that each time we update $\mathcal{H}'$, the shattering strength is at least halved. Moreover, the hypothesis $h$ which we plan to disambiguate is always in $\mathcal{H}'$, which implies $\mathcal{H}'$ is always non-empty. Because the VC dimension of $\mathcal{H}$ is $d$, the initial shattering strength is at most $\binom{n}{\leq d}$. For each $h \in \mathcal{H}$, define $u(h)$ as the number of updates of $\mathcal{H}'$. As the shattering strength was at least halved each time an update is made we have $u(h) \leq \log_2(s(\mathcal{H})) \leq \log_2\left(\binom{n}{\leq d}\right)$. Instead of using the bound $\binom{n}{\leq d} \leq 2n^d$ as in the original proof, we apply the sharper inequality $\binom{n}{\leq d} \leq (\frac{en}{d})^d$, which holds for all integers $n \geq d \geq 1$. Thus, the number of updates $u(h)$ satisfies[5]

$$u(h) \leq \log_2\left(\binom{n}{\leq d}\right) \leq d\log_2\left(\frac{en}{d}\right) \leq 2d\ln\left(\frac{en}{d}\right).$$

By the construction of the disambiguation process, update paths with updates in the same places will give the same disambiguation of a hypothesis. Thus, the number of update paths bounds the size of the disambiguation. The number of ways to place the updates, bounds the number of update paths, and is at most

$$|\overline{\mathcal{H}}| \leq \binom{n}{\leq u(h)} \leq \binom{n}{\leq 2d\ln\left(\frac{en}{d}\right)}. \tag{11}$$

Because $n \geq 6d$, which implies $n \geq 2d\ln(\frac{en}{d})$, we can use the sharper bound on the binomial sum again to obtain

$$|\overline{\mathcal{H}}| \leq \binom{n}{\leq 2d\ln\left(\frac{en}{d}\right)} \leq \left(\frac{en}{2d\ln\left(\frac{en}{d}\right)}\right)^{2d\ln(en/d)},$$

which solves to

$$\ln|\overline{\mathcal{H}}| \leq 2d\ln\left(\frac{en}{d}\right) \cdot \left[\ln\left(\frac{en}{d}\right) - \ln\left(2\ln\left(\frac{en}{d}\right)\right)\right] \leq 2d\ln^2\left(\frac{en}{d}\right).$$

Combining the three cases concludes the proof. $\qquad\square$

With Lemma B.3 at hand, we are now ready to prove Lemma B.2, before doing so we introduce some notation. In the following, for a training sequence $S = ((x_1, y_1), \ldots, (x_n, y_n))$ and a classifier $h \in [0, 1]^{\mathcal{X}}$, let $\mathcal{L}_S^{\neq}(h) = \frac{1}{n}\sum_{i=1}^n \mathbb{1}\{h(x_i) \neq y_i\}$ and $\mathcal{L}_S^{\gamma}(h) = \frac{1}{n}\sum_{i=1}^n \mathbb{1}\{|h(x_i) - y_i| > \gamma\}$.

*Proof of Lemma B.2.* Let $\varepsilon(n, d_\gamma, \delta) = 8\frac{d_\gamma \mathrm{Ln}^2(2en/d_\gamma) + \ln(2/\delta)}{n}$. We want to show that for any interpolator $\mathcal{A}$, $\mathbb{P}_{\mathbf{S}\sim\mathcal{D}^n}[\mathcal{L}_\mathcal{D}^\gamma(\mathcal{A}(\mathbf{S})) > \varepsilon(n, d_\gamma, \delta)] \leq \delta$. If $\varepsilon(n, d_\gamma, \delta) \geq 1$, the statement trivially holds since the cutoff loss is always upper bounded by 1. In the following, we assume that $\varepsilon(n, d_\gamma, \delta) < 1$.

By the definition of interpolator, $\mathcal{A}(\mathbf{S})$ satisfies $\mathcal{L}_{\mathbf{S}}^{\neq}(\mathcal{A}(\mathbf{S})) = 0$. The probability $\mathcal{A}$ returns a bad hypothesis is thus upper bounded by the probability that there exists a hypothesis $h \in \mathcal{H}$ that interpolates the training set $\mathbf{S}$ and has cutoff loss more than $\varepsilon(n, d_\gamma, \delta)$, i.e.,

$$\mathbb{P}_{\mathbf{S}\sim\mathcal{D}^n}[\mathcal{L}_\mathcal{D}^\gamma(\mathcal{A}(\mathbf{S})) > \varepsilon(n, d_\gamma, \delta)] \leq \mathbb{P}_{\mathbf{S}\sim\mathcal{D}^n}[\exists h \in \mathcal{H} : \mathcal{L}_{\mathbf{S}}^{\neq}(h) = 0, \mathcal{L}_\mathcal{D}^\gamma(h) > \varepsilon(n, d_\gamma, \delta)]. \tag{12}$$

---

[5] We take $0\log_2(\frac{en}{0}) = 0\ln(\frac{en}{0}) = 0$ in the following dealing with the degenerated case $d = 0$, which implies that $|\overline{\mathcal{H}}| = 1$, stopping the calculations at Equation (11).

**Symmetrization.** To bound the right-hand side probability, we consider the probability of another event: there exists a hypothesis that interpolates the first sample of size $n$ while performing poorly on an independent ghost sample of the same size, i.e.,

$$\mathop{\mathbb{P}}_{\mathbf{S}_0,\mathbf{S}_1\sim\mathcal{D}^n}[\exists h \in \mathcal{H} : \mathcal{L}^{\neq}_{\mathbf{S}_0}(h) = 0, \mathcal{L}^{\gamma}_{\mathbf{S}_1}(h) > \varepsilon(n, d_\gamma, \delta)/2],$$

where $\mathbf{S}_0$ and $\mathbf{S}_1$ are i.i.d. samples of size $n$ drawn from $\mathcal{D}$. We notice that for any realization $S_0$ of $\mathbf{S}_0$ such that there exists $h' \in \mathcal{H}$ with $\mathcal{L}^{\neq}_{S_0}(h') = 0$ and $\mathcal{L}^{\gamma}_{\mathcal{D}}(h') > \varepsilon(n, d_\gamma, \delta)$, it holds that

$$\mathop{\mathbb{P}}_{\mathbf{S}_1\sim\mathcal{D}^n}[\exists h \in \mathcal{H} : \mathcal{L}^{\neq}_{S_0}(h) = 0, \mathcal{L}^{\gamma}_{\mathbf{S}_1}(h) > \varepsilon(n, d_\gamma, \delta)/2] \geq \mathop{\mathbb{P}}_{\mathbf{S}_1\sim\mathcal{D}^n}[\mathcal{L}^{\gamma}_{\mathbf{S}_1}(h') > \varepsilon(n, d_\gamma, \delta)/2] \tag{13}$$

where the inequality holds since the event on the left-hand side contains the event on the right-hand side. Moreover, by the multiplicative Chernoff bound, we have

$$\mathop{\mathbb{P}}_{\mathbf{S}_1\sim\mathcal{D}^n}[\mathcal{L}^{\gamma}_{\mathbf{S}_1}(h') \leq \varepsilon(n, d_\gamma, \delta)/2] \leq \mathop{\mathbb{P}}_{\mathbf{S}_1\sim\mathcal{D}^n}[\mathcal{L}^{\gamma}_{\mathbf{S}_1}(h') \leq \mathcal{L}^{\gamma}_{\mathcal{D}}(h')/2] \leq \exp\left(-n\,\mathcal{L}^{\gamma}_{\mathcal{D}}(h')/8\right) \leq \exp\left(-n\varepsilon(n, d_\gamma, \delta)/8\right)$$

$$\leq \delta/2 \leq 1/2, \tag{14}$$

where the third inequality holds by the definition of $\varepsilon(n, d_\gamma, \delta)$. Now combining Equations (13) and (14), we have

$$\mathop{\mathbb{P}}_{\mathbf{S}_1\sim\mathcal{D}^n}[\exists h \in \mathcal{H} : \mathcal{L}^{\neq}_{S_0}(h) = 0, \mathcal{L}^{\gamma}_{\mathbf{S}_1}(h) > \varepsilon(n, d_\gamma, \delta)/2] \geq 1/2.$$

for any realization $S_0$ of $\mathbf{S}_0$ such that there exists $h' \in \mathcal{H}$ with $\mathcal{L}_{S_0}(h') = 0$ and $\mathcal{L}^{\gamma}_{\mathcal{D}}(h') > \varepsilon(n, d_\gamma, \delta)$. This implies

$$\mathbb{1}\{\exists h \in \mathcal{H} : \mathcal{L}^{\neq}_{\mathbf{S}_0}(h) = 0, \mathcal{L}^{\gamma}_{\mathcal{D}}(h) > \varepsilon(n, d_\gamma, \delta)\}$$
$$\leq 2 \mathop{\mathbb{P}}_{\mathbf{S}_1\sim\mathcal{D}^n}[\exists h \in \mathcal{H} : \mathcal{L}^{\neq}_{\mathbf{S}_0}(h) = 0, \mathcal{L}^{\gamma}_{\mathbf{S}_1}(h) > \varepsilon(n, d_\gamma, \delta)/2]\mathbb{1}\{\exists h \in \mathcal{H} : \mathcal{L}^{\neq}_{\mathbf{S}_0}(h) = 0, \mathcal{L}^{\gamma}_{\mathcal{D}}(h) > \varepsilon(n, d_\gamma, \delta)\}$$

Now using this relation and combining with Equation (12), we get that

$$\mathop{\mathbb{P}}_{\mathbf{S}\sim\mathcal{D}^n}[\mathcal{L}^{\gamma}_{\mathcal{D}}(\mathcal{A}(\mathbf{S})) > \varepsilon(n, d_\gamma, \delta)]$$
$$\leq \mathop{\mathbb{P}}_{\mathbf{S}_0\sim\mathcal{D}^n}[\exists h \in \mathcal{H} : \mathcal{L}^{\neq}_{\mathbf{S}_0}(h) = 0, \mathcal{L}^{\gamma}_{\mathcal{D}}(h) > \varepsilon(n, d_\gamma, \delta)]$$
$$= \mathop{\mathbb{E}}_{\mathbf{S}_0\sim\mathcal{D}^n}[\mathbb{1}\{\exists h \in \mathcal{H} : \mathcal{L}^{\neq}_{\mathbf{S}_0}(h) = 0, \mathcal{L}^{\gamma}_{\mathcal{D}}(h) > \varepsilon(n, d_\gamma, \delta)\}]$$
$$\leq \mathop{\mathbb{E}}_{\mathbf{S}_0}[2 \mathop{\mathbb{P}}_{\mathbf{S}_1\sim\mathcal{D}^n}[\exists h \in \mathcal{H} : \mathcal{L}^{\neq}_{\mathbf{S}_0}(h) = 0, \mathcal{L}^{\gamma}_{\mathbf{S}_1}(h) > \varepsilon(n, d_\gamma, \delta)/2]\mathbb{1}\{\exists h \in \mathcal{H} : \mathcal{L}^{=}_{\mathbf{S}_0}(h) = 0, \mathcal{L}^{\gamma}_{\mathcal{D}}(h) > \varepsilon(n, d_\gamma, \delta)\}]$$
$$\leq 2 \mathop{\mathbb{P}}_{\mathbf{S}_0,\mathbf{S}_1\sim\mathcal{D}^n}[\exists h \in \mathcal{H} : \mathcal{L}^{\neq}_{\mathbf{S}_0}(h) = 0, \mathcal{L}^{\gamma}_{\mathbf{S}_1}(h) > \varepsilon(n, d_\gamma, \delta)/2].$$

Thus, by this relation it suffices to bound the probability on the right-hand side by $\delta/2$ as we will do in the following.

**Permutation.** For $\mathbf{S}_0, \mathbf{S}_1$ defined above and any permutation $\sigma \in \{0,1\}^n$, let $\mathbf{S}_\sigma = (\mathbf{S}_{\sigma_1,1}, \dots, \mathbf{S}_{\sigma_n,n})$ and $\mathbf{S}_{1-\sigma} = (\mathbf{S}_{1-\sigma_1,1}, \dots, \mathbf{S}_{1-\sigma_n,n})$. Observe that $\mathbf{S}_\sigma, \mathbf{S}_{1-\sigma}$ are distributed as $\mathbf{S}_0, \mathbf{S}_1$. This implies

$$\mathop{\mathbb{P}}_{\mathbf{S}_0,\mathbf{S}_1\sim\mathcal{D}^n}[\exists h \in \mathcal{H} : \mathcal{L}^{\neq}_{\mathbf{S}_0}(h) = 0, \mathcal{L}^{\gamma}_{\mathbf{S}_1}(h) > \varepsilon(n, d_\gamma, \delta)/2] = \mathop{\mathbb{P}}_{\mathbf{S}_0,\mathbf{S}_1\sim\mathcal{D}^n}[\exists h \in \mathcal{H} : \mathcal{L}^{\neq}_{\mathbf{S}_\sigma}(h) = 0, \mathcal{L}^{\gamma}_{\mathbf{S}_{1-\sigma}}(h) > \varepsilon(n, d_\gamma, \delta)/2].$$

Now let $\boldsymbol{\sigma} = (\boldsymbol{\sigma}_1, \dots, \boldsymbol{\sigma}_n)$ be i.i.d. Bernoulli random variables independent of $\mathbf{S}_0, \mathbf{S}_1$, i.e. $\mathbb{P}[\boldsymbol{\sigma}_i = 0] = \mathbb{P}[\boldsymbol{\sigma}_i = 1] = 1/2$ for all $i \in [n]$. We then have that

$$\mathop{\mathbb{P}}_{\mathbf{S}_0,\mathbf{S}_1\sim\mathcal{D}^n}[\exists h \in \mathcal{H} : \mathcal{L}^{\neq}_{\mathbf{S}_0}(h) = 0, \mathcal{L}^{\gamma}_{\mathbf{S}_1}(h) > \varepsilon(n, d_\gamma, \delta)/2]$$
$$= \mathop{\mathbb{E}}_{\boldsymbol{\sigma}}\left[\mathop{\mathbb{P}}_{\mathbf{S}_0,\mathbf{S}_1\sim\mathcal{D}^n}[\exists h \in \mathcal{H} : \mathcal{L}^{\neq}_{\mathbf{S}_\sigma}(h) = 0, \mathcal{L}^{\gamma}_{\mathbf{S}_{1-\sigma}}(h) > \varepsilon(n, d_\gamma, \delta)/2]\right]$$
$$= \mathop{\mathbb{E}}_{\mathbf{S}_0,\mathbf{S}_1\sim\mathcal{D}^n}\left[\mathop{\mathbb{P}}_{\boldsymbol{\sigma}}[\exists h \in \mathcal{H} : \mathcal{L}^{\neq}_{\mathbf{S}_\sigma}(h) = 0, \mathcal{L}^{\gamma}_{\mathbf{S}_{1-\sigma}}(h) > \varepsilon(n, d_\gamma, \delta)/2]\right]$$
$$\leq \sup_{S_0, S_1 \in (\mathcal{X}\times[0,1])^n} \mathop{\mathbb{P}}_{\boldsymbol{\sigma}}\left[\exists h \in \mathcal{H} : \mathcal{L}^{\neq}_{S_\sigma}(h) = 0, \mathcal{L}^{\gamma}_{S_{1-\sigma}}(h) > \varepsilon(n, d_\gamma, \delta)/2\right]$$

where the first equality holds by the distributional equivalence of $(\mathbf{S}_0, \mathbf{S}_1)$ and $(\mathbf{S}_\sigma, \mathbf{S}_{1-\sigma})$ for any $\sigma \in \{0,1\}^n$, the second equality holds by independence of $\boldsymbol{\sigma}$, $\mathbf{S}_0$ and $\mathbf{S}_1$ and shifting order of expectation, and the last inequality holds by the definition of supremum. By the above, it suffices to show that for any fixed $(S_0, S_1) \in (\mathcal{X} \times [0,1])^{2n}$ it holds that

$$\mathbb{P}_{\boldsymbol{\sigma}}\left[\exists h \in \mathcal{H} : \mathcal{L}_{S_\sigma}^{\neq}(h) = 0, \mathcal{L}_{S_{1-\sigma}}^{\gamma}(h) > \varepsilon(n, d_\gamma, \delta)/2\right] \leq \delta/2.$$

**Reduction to a finite class.** Let $S_0 = ((x_{0,1}, y_{0,1}), \ldots, (x_{0,n}, y_{0,n})), S_1 = ((x_{1,1}, y_{1,1}), \ldots, (x_{1,n}, y_{1,n})) \in (\mathcal{X} \times [0,1])^n$ be fixed from now on. And for any $\sigma = (\sigma_1, \ldots, \sigma_n) \in \{0,1\}^n$, let $S_\sigma = ((x_{\sigma_1,1}, y_{\sigma_1,1}), \ldots, (x_{\sigma_n,n}, y_{\sigma_n,n}))$ and $S_{1-\sigma} = ((x_{1-\sigma_1,1}, y_{1-\sigma_1,1}), \ldots, (x_{1-\sigma_n,n}, y_{1-\sigma_n,n}))$.

Now for any $h \in \mathcal{H}$, define a partial concept $h' : S_0 \cup S_1 \to \{0, 1, \star\}$ on $S_0 \cup S_1$ as follows: for any $j \in \{0,1\}$ and $i \in [n]$

$$h'(x_{j,i}, y_{j,i}) = \begin{cases} 0 & \text{if } h(x_{j,i}) = y_{j,i} \\ \star & \text{if } 0 < |h(x_{j,i}) - y_{j,i}| \leq \gamma \\ 1 & \text{if } \gamma < |h(x_{j,i}) - y_{j,i}| \end{cases}$$

and let $\mathcal{H}' = \cup_{h \in \mathcal{H}}\{h'\}$. We notice that by construction the VC-dimension of the partial concept class $\mathcal{H}'$ is at most the $\gamma$-graph dimension of $\mathcal{H}$, i.e., $d_\gamma$. Observing that $\mathcal{H}'$ is over a finite domain $S_0 \cup S_1$ of size $m \leq 2n$, we use Lemma B.3 to construct a disambiguation $\overline{\mathcal{H}'}$ of $\mathcal{H}'$, which satisfies

$$\ln|\overline{\mathcal{H}'}| \leq 2\mathrm{VC}(\mathcal{H}')\mathrm{Ln}^2\left(\frac{em}{\mathrm{VC}(\mathcal{H}')}\right) \leq 2d_\gamma\mathrm{Ln}^2\left(\frac{2en}{d_\gamma}\right)$$

where, we have used the fact that function $2x\mathrm{Ln}^2(ey/x)$ is non-decreasing with respect to both $x \geq 1$ and $y \geq 1$.

We notice that for any $h \in \mathcal{H}$, if $h$ interpolates $S_\sigma$ and is off by more than $\gamma$ on at least $n\varepsilon/2$ of the points in $S_{1-\sigma}$, the corresponding partial concept $h' \in \mathcal{H}'$ is 0 on the points in $S_\sigma$ and is 1 on the same $n\varepsilon/2$ of the points in $S_{1-\sigma}$. Moreover, the corresponding disambiguated concept $\overline{h'} \in \overline{\mathcal{H}'}$ is 0 on the first $n$ points and is 1 on those same $n\varepsilon/2$ of the remaining points. This implies that the event $\{\exists h \in \mathcal{H} : \mathcal{L}_{S_\sigma}^{\neq}(h) = 0, \mathcal{L}_{S_{1-\sigma}}^{\gamma}(h) > \varepsilon(n, d_\gamma, \delta)/2\}$ is contained in the event

$$\left\{\exists \overline{h'} \in \overline{\mathcal{H}'} : \sum_{i=1}^{n} \mathbb{1}\{\overline{h'}(x_{\boldsymbol{\sigma}_i,i}, y_{\boldsymbol{\sigma}_i,i}) = 1\} = 0, \sum_{i=1}^{n} \mathbb{1}\{\overline{h'}(x_{1-\boldsymbol{\sigma}_i,i}, y_{1-\boldsymbol{\sigma}_i,i}) = 1\} \geq n\varepsilon(n, d_\gamma, \delta)/2\right\}.$$

Using this we have that

$$\mathbb{P}_{\boldsymbol{\sigma}}\left[\exists h \in \mathcal{H} : \mathcal{L}_{S_\sigma}^{\neq}(h) = 0, \mathcal{L}_{S_{1-\sigma}}^{\gamma}(h) > \varepsilon(n, d_\gamma, \delta)/2\right]$$

$$\leq \mathbb{P}_{\boldsymbol{\sigma}}\left[\exists \overline{h'} \in \overline{\mathcal{H}'} : \sum_{i=1}^{n} \mathbb{1}\{\overline{h'}(x_{\boldsymbol{\sigma}_i,i}, y_{\boldsymbol{\sigma}_i,i}) = 1\} = 0, \sum_{i=1}^{n} \mathbb{1}\{\overline{h'}(x_{1-\boldsymbol{\sigma}_i,i}, y_{1-\boldsymbol{\sigma}_i,i}) = 1\} \geq n\varepsilon(n, d_\gamma, \delta)/2\right]$$

$$\leq \sum_{\overline{h'} \in \overline{\mathcal{H}'}} \mathbb{P}_{\boldsymbol{\sigma}}\left[\sum_{i=1}^{n} \mathbb{1}\{\overline{h'}(x_{\boldsymbol{\sigma}_i,i}, y_{\boldsymbol{\sigma}_i,i}) = 1\} = 0, \sum_{i=1}^{n} \mathbb{1}\{\overline{h'}(x_{1-\boldsymbol{\sigma}_i,i}, y_{1-\boldsymbol{\sigma}_i,i}) = 1\} \geq n\varepsilon(n, d_\gamma, \delta)/2\right]$$

where the last inequality follows from the union bound. Now, fix any $\overline{h'} \in \overline{\mathcal{H}'}$, we bound the probability inside the summation, i.e.,

$$\mathbb{P}_{\boldsymbol{\sigma}}\left[\sum_{i=1}^{n} \mathbb{1}\{\overline{h'}(x_{\boldsymbol{\sigma}_i,i}, y_{\boldsymbol{\sigma}_i,i}) = 1\} = 0, \sum_{i=1}^{n} \mathbb{1}\{\overline{h'}(x_{1-\boldsymbol{\sigma}_i,i}, y_{1-\boldsymbol{\sigma}_i,i}) = 1\} \geq n\varepsilon(n, d_\gamma, \delta)/2\right].$$

For this probability to be non zero, it must be the case that among the $n$ pairs $((x_{0,i}, y_{0,i}), (x_{1,i}, y_{1,i}))$, there exists at least $n\varepsilon(n, d_\gamma, \delta)/2$ indices $i$ such that one of $\overline{h'}(x_{0,i}, y_{0,i}), \overline{h'}(x_{1,i}, y_{1,i})$ is equal to 0 and the other is equal to 1. Assume this condition holds. Under this assumption, for the event inside the probability to occur, it must be the case that all of these pairs has the 0 entry mapped to $S_\sigma$ and the 1 entry mapped to $S_{1-\sigma}$. Since each pair is swapped with probability $1/2$ independently, we have that the probability of this event is at most $2^{-n\varepsilon(n, d_\gamma, \delta)/2}$.

Combining this with the bound on the size of $\overline{\mathcal{H}'}$, we obtain

$$
\begin{aligned}
\mathop{\mathbb{P}}_{\boldsymbol{\sigma}}\left[\exists h \in \mathcal{H} : \mathcal{L}_{S_{\boldsymbol{\sigma}}}^{\neq}(h) = 0, \mathcal{L}_{S_{1-\boldsymbol{\sigma}}}^{\gamma}(h) > \varepsilon(n, d_{\gamma}, \delta)/2\right] &\leq \sum_{h \in \overline{\mathcal{H}'}} 2^{-n\varepsilon(n,d_{\gamma},\delta)/2} \\
&\leq e^{2d_{\gamma}\mathrm{Ln}^2(2en/d_{\gamma})} \cdot 2^{-n\varepsilon(n,d_{\gamma},\delta)/2} \\
&\leq e^{2d_{\gamma}\mathrm{Ln}^2(2en/d_{\gamma})-n\varepsilon(n,d_{\gamma},\delta)/4} \\
&\leq \frac{\delta}{2},
\end{aligned}
$$

where the second to last inequality follows from $\ln(2) \geq 1/2$ and the last inequality follows from the definition of $\varepsilon(n, d_{\gamma}, \delta) = 8\frac{d_{\gamma}\mathrm{Ln}^2(2en/d_{\gamma})+\ln(2/\delta)}{n}$. This concludes the proof of the lemma.

$\square$

# C. Lower Bounds

## C.1. Proof of Theorem 3.5

In this section we present the proof of Theorem C.1 which contains both a constant probability lower bound and an expectation lower bound, where the statement in Theorem 3.5 follows from the latter. We now state Theorem C.1 and then give its proof.

**Theorem C.1.** *For any hypothesis class $\mathcal{H} \subseteq [0,1]^{\mathcal{X}}$ with $\gamma$-graph dimension $d_\gamma \geq 2$ and any $0 < \varepsilon < 1/4$, there exist an interpolator $\mathcal{A}$ and a realizable distribution $\mathcal{D}$ such that the following holds. Let $\mathcal{A}'$ be any interpolator-based aggregation algorithm using $\mathcal{A}$ and a proper aggregation rule, and let $\mathbf{S} \sim \mathcal{D}^n$ with $n \leq d_\gamma/(32\varepsilon)$. Then, with probability at least $1/2$ over $\mathbf{S}$,*

$$\mathcal{L}_{\mathcal{D}}^{\gamma}(\mathcal{A}'(\mathbf{S})) > 2\varepsilon, \tag{15}$$

*and moreover,*

$$\mathop{\mathbb{E}}_{\mathbf{S} \sim \mathcal{D}^n} \left[ \mathcal{L}_{\mathcal{D}}^{\gamma}(\mathcal{A}'(\mathbf{S})) \right] > \varepsilon. \tag{16}$$

*The distributions $\mathcal{D}$ witnessing Equation* (15) *and Equation* (16) *may differ.*

*Proof of Theorem C.1:* Since $\mathcal{H}$ has $\gamma$-graph dimension $d$, we know that there exists a set of size $d$ that is $\gamma$-graph shattered by $\mathcal{H}$. Let $\mathcal{X}_d = \{x_1, \ldots, x_d\}$, denote such a $\gamma$-graph shattered set of size $d$ and let $h$ be a hypothesis witnessing the graph shattering. We now construct the hard distribution $\mathcal{D}$ as follows: let $\mathcal{D}(x_1, h(x_1)) = 1 - 4\varepsilon$ and $\mathcal{D}(x_i, h(x_i)) = \frac{4\varepsilon}{d-1}$ for $i = 2, \ldots, d$. We notice this distribution is realizable by $h \in \mathcal{H}$. We next construct a worst-case interpolator $\mathcal{A}$ for aggregation. Let $\mathcal{A}$ be any interpolator algorithm that, given a training sequence $S$ consisting of examples from $\{(x_1, h(x_1)), \ldots, (x_d, h(x_d))\}$, besides being an interpolator (see Definition A.1) - returns a hypothesis $\mathcal{A}(S) \in \mathcal{H}$ that is consistent with $S$, satisfies the following: for any $i \in \{1, \ldots, d\}$, if $(x_i, h(x_i)) \notin S$, then $|\mathcal{A}(S)(x_i) - h(x_i)| > \gamma$. Since a hypothesis $h' \in \mathcal{H}$ with this property exists by $\mathcal{H}$ $\gamma$-graph shattering $\mathcal{X}_d$ with $h$ as witness, an interpolator with this property exists.

We are now ready to prove the theorem. Let $\mathbf{S} = ((\mathbf{s}_1, h(\mathbf{s}_1)), \ldots, (\mathbf{s}_n, h(\mathbf{s}_n)))$ be a training sequence of size $n \leq d/(32\varepsilon)$ with the examples being sampled i.i.d from $\mathcal{D}$. Let $h_1, \ldots, h_m$ be the hypotheses produced by $\mathcal{A}'$ when calling $\mathcal{A}$ with sub-training sequences $\mathbf{S}_1, \ldots, \mathbf{S}_m$.

Consider the random variable $\mathbf{X}$ which counts the number of times points in $\{x_2, \ldots, x_d\}$ appear in $\mathbf{S}$, i.e., $\mathbf{X} = \sum_{j=1}^{n} \mathbb{1}\{\mathbf{s}_j \in \{x_2, \ldots, x_d\}\}$. We have that

$$\mathbb{E}[\mathbf{X}] = \sum_{j=1}^{n} \mathbb{P}[\mathbf{s}_j \in \{x_2, \ldots, x_d\}] = n(d-1)\frac{4\varepsilon}{d-1} = 4n\varepsilon \leq 4d/32 = d/8,$$

where the inequality follows from the assumption that $n \leq d/(32\varepsilon)$. By Markov's inequality, we then have that

$$\mathbb{P}[\mathbf{X} \geq d/4] \leq \frac{\mathbb{E}[\mathbf{X}]}{d/4} \leq \frac{d/8}{d/4} = \frac{1}{2}.$$

Let $E$ be the event that $\mathbf{X} < d/4$. By the above, $\mathbb{P}[E] \geq 1/2$. Now consider any realization $S$ of $\mathbf{S}$ where the event $E$ holds. Let $I_S = \{i \in \{2, \ldots, d\} : x_i \in S\}$ be the indices of points observed in $S$. Let $I_{bad} = \{2, \ldots, d\} \setminus I_S$ be the indices of the unobserved points. Under event $E$, the number of observed points from $\{x_2, \ldots, x_d\}$ is less than $d/4$, i.e., $|I_S| < d/4$, so $|I_{bad}| > (d-1) - d/4$.

Furthermore, by the definition of an Itnerpolator-Based aggregation algorithm (see Definition 3.3) $\mathcal{A}'$, each sub-training sequence $S_j$, generated from $S$ contains only points from $S$. Therefore, for any $i \in I_{bad}$, the point $x_i$ is not in any $S_j$. This implies, by the definition of $\mathcal{A}$, that for each $j \in [k]$, and for all $i \in I_{bad}$, $|h_j(x_i) - h(x_i)| > \gamma$. Since the aggregation rule $r$ is proper, for any $x$, $\mathcal{A}'(\mathbf{S})(x) = r(h_1(x), \ldots, h_m(x), x) \in \{h_1(x), \ldots, h_m(x)\}$. Therefore, for all $i \in I_{bad}$, we have $|\mathcal{A}'(\mathbf{S})(x_i) - h(x_i)| > \gamma$.

So we conclude that the aggregation algorithm fails on the set $\{x_i : i \in I_{bad}\}$. By the definition of the distribution $\mathcal{D}$, the loss of $\mathcal{A}'(\mathbf{S})$ is then lower bounded as:

$$\mathcal{L}_{\mathcal{D}}^{\gamma}(\mathcal{A}'(\mathbf{S})) = \underset{(x,y)\sim\mathcal{D}}{\mathbb{P}}[|\mathcal{A}'(\mathbf{S})(x) - y| > \gamma] \geq 4\varepsilon\frac{|I_{bad}|}{d-1} > 4\varepsilon\left(1 - \frac{d}{4(d-1)}\right) \geq 2\varepsilon,$$

where the strict inequality follows from $|I_{bad}| > d - 1 - d/4$ and the last inequality follows from $d \geq 2$. Since $E$ happens with probability at least $1/2$ and implies the above lower bound on the error the above gives that Equation (15) holds with probability $1/2$ as claimed. Furthermore we by the above that

$$\underset{\mathbf{S}\sim\mathcal{D}^n}{\mathbb{E}}[\mathcal{L}_{\mathcal{D}}^{\gamma}(\mathcal{A}'(\mathbf{S}))] \geq \underset{\mathbf{S}\sim\mathcal{D}^n}{\mathbb{P}}[E] \cdot \underset{\mathbf{S}\sim\mathcal{D}^n}{\mathbb{E}}[\mathcal{L}_{\mathcal{D}}^{\gamma}(\mathcal{A}'(\mathbf{S}))|E] > \frac{1}{2}2\varepsilon = \varepsilon,$$

implying Equation (16) completing the proof.

$\square$

### C.2. Proof of Theorem 3.8

In this section we present the proof of Theorem C.2 which contains both a constant probability lower bound and an expectation lower bound, where the statement in Theorem 3.8 follows from the latter. We now state Theorem C.2.

**Theorem C.2.** *For any $\gamma \in (0,1)$ and $d_\gamma \geq 2$, there exists a hypothesis class $\mathcal{H}^{\gamma,d_\gamma} \subseteq [0,1]^{\mathcal{X}}$ with $\gamma$-graph dimension $d_\gamma$ and $\gamma$-OIG-dimension at most 3, such that for any finite aggregation algorithm $\mathcal{A}'$ using an interpolating aggregation rule $r$ and any $0 < \varepsilon < 1/32$, there exists a realizable distribution $\mathcal{D}$ over $\mathcal{X} \times [0,1]$ such that when $\mathcal{A}'$ is given a training sequence $\mathbf{S} \sim \mathcal{D}^n$ of size $n \leq d_\gamma/(128\varepsilon)$, it holds with probability at least $1/16$ that*

$$\mathcal{L}_{\mathcal{D}}^{\gamma}(\mathcal{A}'(\mathbf{S})) > \varepsilon. \tag{17}$$

*and*

$$\underset{\mathbf{S}\sim\mathcal{D}^n}{\mathbb{E}}[\mathcal{L}_{\mathcal{D}}^{\gamma}(\mathcal{A}'(\mathbf{S}))] > 2\varepsilon. \tag{18}$$

*where the distributions $\mathcal{D}$ in the two cases may be different.*

To show Theorem C.2 we need the following anti-concentration inequality.

**Lemma C.3.** *[Reverse Markov's inequality see e.g. Lemma B.1 (Shalev-Shwartz & Ben-David, 2014)] Let $\mathbf{X}$ be a random variable that takes values in $[0,1]$. Then, for any $a \in (0,1)$,*

$$\mathbb{P}[\mathbf{X} > a] \geq \frac{\mathbb{E}[\mathbf{X}] - a}{1 - a} \geq \mathbb{E}[\mathbf{X}] - a.$$

With the above lemma we are now ready to prove the theorem.

*Proof of Theorem C.2:* We first for any $0 < \gamma < 1$ and $d \in \mathbb{N}$ construct a hypothesis class $\mathcal{H}^{\gamma,d}$ with $\gamma$-graph dimension $d$ and $\gamma$-OIG-dimension at most 3. So let $\gamma$ and $d$ be fixed.

For each $A \subseteq \mathbb{N}$ with $|A| = d$ define a unique number $\gamma_A \in (\gamma, 1]$. Since there is a countably infinite number of such subsets (this can be seen by $\{A \subseteq \mathbb{N} : |A| = d\}$ being equal to the countable union of finite sets $\cup_{k\in\mathbb{N}}\{A \subseteq [k] : |A| = d\}$), we can always choose these numbers to be distinct, by for instance setting $\gamma_A = \gamma + (1-\gamma)/l$ where $l$ is number of $A$ in the enumeration of $\{A \subseteq \mathbb{N} : |A| = d\}$. For each such subset $A$, we then define a hypothesis on the input space $\mathcal{X} = \mathbb{N}$ as follows:

$$h_A(i) = \begin{cases} 0 & \text{if } i \in A \\ \gamma_A & \text{else} \end{cases}$$

and let $\mathcal{H}^{\gamma,d} = \cup_{A\in\{A'\subseteq\mathbb{N}:|A'|=d\}}h_A$.

**$\gamma$-graph dimension of $\mathcal{H}^{\gamma,d}$.** We notice that this hypothesis class has $\gamma$-graph dimension $d$, as for any subset $B \subseteq \mathbb{N}$ with $|B| = d$, the hypothesis $h_B$ witnesses the graph shattering, as it is zero on the whole set $B$, and we can find a hypothesis that is 0 on any subset $B'$ of $B$ and strictly larger than $\gamma$ else by choosing $A' = B' \cup C$ where $C \subseteq \mathbb{N}\backslash B$ such that $|B' \cup C| = d$. Furthermore for a subset $S \subseteq \mathbb{N}$ with $|S| = d+1$, there is no hypothesis in $\mathcal{H}^{\gamma,d}$ that can witness the graph shattering on $S$. To see this can not happen assume such an $h_A$ exists. Since we have that $|S| = d + 1$, there exists at least one point $i \in S$ such that $h_A(i) = \gamma_A$, however in this case we can not have a $h_B$ that satisfies $\gamma_A = h_A(i) = h_B(i)$ and $|h_A(j), h_B(j)| > \gamma$ for $j \neq i$, since the first condition, by the uniqueness of $\gamma_A$, implies that $B = A$, so the second condition can not hold.

**$\gamma$-OIG dimension of $\mathcal{H}^{\gamma,d}$.** We now argue that the $\gamma$-OIG-dimension of $\mathcal{H}^{\gamma,d}$ is at most 3. To see this, consider any set of $n \geq 3$ points $S = \{x_1, \ldots x_n\}$, and the $\gamma$ one inclusion graph induced by $S$ and $\mathcal{H}^{\gamma,d}$ which has vertex set $V = \mathcal{H}^{\gamma,d}|_S = \left\{(h(x_1), \ldots, h(x_n))|h \in \mathcal{H}^{\gamma,d}\right\}$, and edge set $E$ where each edge is a hyperedge $(f, i)$, containing $v \in V$ if $v(x_j) = f(x_j)$ for all $j \in [n]\backslash i$. Consider any finite subgraph $(V', E')$ of the one inclusion graph, we now show that we can orient the edges of this subgraph such that each vertex has out-degree at most 1, which by $n \geq 3$ would imply that the $\gamma$-OIG-dimension is at most 3 by Definition 3.9. We claim that the orientation that given an edge $(f, i)$ orients it towards $v \in (f, i)$ such that $v(i) < v'(i)$ for all other $v' \in (f, i)$ gives an out-degree of at most 1 for each vertex in the $\gamma$ - one inclusion graph.

To see this, we first notice that any vertex $v \in V'$ with at least 2 non zero values is always the only element in an edge $(f, i)$ containing $v$, as the hypothesis $v$ has 2 non zero values there exists a $j \in [n]\backslash i$ such that $f(x_j) = v(x_j) \neq 0$ and since the non zero values are uniquely defined by the value $\gamma_A$, there can not be another hypothesis in $\mathcal{H}^{\gamma,d}$ that has the same value on $j$ and a different value on $i$. Thus, such a vertex will always have the edge oriented towards it, and have outdegree at most 0.

Now consider a vertex $v \in V'$ with 1 non zero value, with the non zero value on $x_j$. Similar to above we have that the value $v(x_j)$ is unique for $v$ so for any edge $(f, i)$ with $i \neq j$, and the edge containing $v$, $v \in (f, i)$, we have that $(f, i)$ only contains $v$, and thus the edge is oriented towards $v$. Thus, the only place where the edge can be oriented away from $v$ is when $i = j$, implying an outdegree of at most 1 for $v$.

Finally, if there exists a vertex $v \in V'$ with 0 non zero values, i.e. all zero, it will by the choice of the edge orientation (we are orienting towards the vertex with the smallest value on $i$). Thus, such an vertex will have an outdegree of at most 0.

Thus we have shown that the orientation gives an outdegree of at most 1 for each vertex in the $\gamma$ - one inclusion graph, and thus for $n \geq 3$ the outdegree is at most $1 \leq n/3$ implying that the $\gamma$-OIG-dimension of $\mathcal{H}^{\gamma,d}$ is at most 3.

**Lower bounds.** We now define a family of hard distributions and prove the claimed lower bounds on these distributions. Each distribution is indexed by a vector in the set

$$Q = \left\{ \vec{A} \in \{1\} \times [k_u]^{d-1} \mid \forall i, j \in [d], i \neq j \implies \vec{A}_i \neq \vec{A}_j \right\},$$

where $k_u = \lceil 2d \max_{n' \leq d/(128\varepsilon)} m(n') + d/4 + 1 \rceil$ is the effective universe size for the distributions. Let $\{\vec{A}\} = \{\vec{A}_1, \ldots, \vec{A}_d\}$. For $0 < \varepsilon < 1/32$ and $\vec{A} \in Q$, we define the distribution $\mathcal{D}_{\varepsilon,\vec{A}}$ as follows:

$$\mathcal{D}_{\varepsilon,\vec{A}}(i,0) = \begin{cases} 1 - 16\varepsilon & \text{if } i = 1 \\ \frac{16\varepsilon}{d-1} & \text{if } i \in \{\vec{A}\} \backslash \{1\} \\ 0 & \text{else} \end{cases}$$

We notice that this distribution is realizable by $\mathcal{H}^{\gamma,d}$, by the hypothesis $h_{\{\vec{A}\}}$. Furthermore, we notice that for any $\vec{A} \in Q$ we have that sampling from $\mathcal{D}_{\varepsilon,\vec{A}}$ is equivalent to sampling first a number $\mathbf{t} \in [d]$ with

$$\mathbb{P}_{\mathcal{D}_t}[\mathbf{t} = 1] = 1 - 16\varepsilon, \quad \mathbb{P}_{\mathcal{D}_t}[\mathbf{t} = i] = \frac{16\varepsilon}{d-1} \text{ for } i = 2, \ldots, d,$$

and then returning the sample $(\vec{A}_{\mathbf{t}}, 0)$. For $\vec{A} \in Q$ and $\vec{\mathbf{t}} = (\mathbf{t}_1, \ldots, \mathbf{t}_n) \sim \mathcal{D}_t^n$, let $\vec{A}|\vec{\mathbf{t}} = (\vec{A}_{\mathbf{t}_1}, \ldots, \vec{A}_{\mathbf{t}_n})$. Let $\mathbf{S}(\vec{A}|\vec{\mathbf{t}}) = ((\vec{A}_{\mathbf{t}_1}, 0), \ldots, (\vec{A}_{\mathbf{t}_n}, 0))$ be a training sequence of size $n$ sampled i.i.d from $\mathcal{D}_{\varepsilon,\vec{A}}$ in the equivalent way via the $\mathbf{t}_i$'s i.e. $\mathbf{t}_i$ are i.i.d. from $\mathcal{D}_t$. From now on we will let $n \leq d/(128\varepsilon)$.

Let now $\mathbf{h}_1, \ldots \mathbf{h}_m$ be the hypothesis produced by $\mathcal{A}'$ when given $\mathbf{S}(\vec{A}|\vec{\mathbf{t}})$ and let $\mathcal{A}'(\mathbf{S}(\vec{A}|\vec{\mathbf{t}})) = r(\mathbf{h}_1, \ldots, \mathbf{h}_m, x)$ be the output of the aggregation rule (the number $m$ might be random but $m \leq m(n) \leq \max_{n' \leq d/(128\varepsilon)} m(n') < \infty$ ). We have by Definition 3.6 that each $\mathbf{h}_1, \ldots, \mathbf{h}_m \in \mathcal{H}^{\gamma,d}$, implying that each hypothesis $\mathbf{h}_j$ is 0 on at most $d$ points. Thus if we let $R_0(\vec{A}|\vec{\mathbf{t}}) = \{i \in \mathbb{N} | \exists j \in [k] : \mathbf{h}_j(i) = 0\}$ be the region where at least one hypothesis is zero, we have that $|R_0(\vec{A}|\vec{\mathbf{t}})| \leq md \leq \max_{n' \leq d/(128\varepsilon)} m(n')d$. Furthermore since the aggregation rule $r$ is interpolating, we have that for all points outside this region $i \notin R_0(\vec{A}|\vec{\mathbf{t}})$, it holds that $\mathcal{A}'(S(\vec{A}|\vec{\mathbf{t}}))(i) > \gamma$, as all hypotheses are larger than $\gamma$ on these points by construction of $\mathcal{H}^{\gamma,d}$. Noticing this we have that

$$\mathcal{L}_{\mathcal{D}_{\varepsilon,\vec{A}}}(\mathcal{A}'(S(\vec{A}|\vec{\mathbf{t}}))) = \underset{(\mathbf{x},\mathbf{y}) \sim \mathcal{D}_{\varepsilon,\vec{A}}}{\mathbb{P}} \left[ |\mathcal{A}'(S(\vec{A}|\vec{\mathbf{t}}))(\mathbf{x}) - \mathbf{y}| > \gamma \right] = \underset{\mathbf{t} \sim \mathcal{D}_t}{\mathbb{P}} \left[ \mathcal{A}'(S(\vec{A}|\vec{\mathbf{t}}))(\vec{A}_\mathbf{t}) > \gamma \right]$$
$$\geq \underset{\mathbf{t} \sim \mathcal{D}_t}{\mathbb{P}} \left[ \vec{A}_\mathbf{t} \notin R_0(\vec{A}|\vec{\mathbf{t}}) \right]$$

where we have used in the second equality that $y = 0$ under the distribution $\mathcal{D}_{\varepsilon,\vec{A}}$ and the equivalent way of sampling from $\mathcal{D}_{\varepsilon,\vec{A}}$ via $\mathbf{t}$, and in the last inequality that $\mathcal{A}'(S(\vec{A}|\vec{\mathbf{t}}))(i) > \gamma$ for all $i \notin R_0(\vec{A}|\vec{\mathbf{t}})$. Now let's define the set of indices from from $[d] \setminus \{1\}$ seen and not seen in $\vec{\mathbf{t}}$ as

$$I_{\vec{\mathbf{t}}} = \left\{ i \in [d] \setminus \{1\} : i \in \{\vec{\mathbf{t}}\} \right\} \quad \text{and} \quad I_{\vec{\mathbf{t}}'} = \left\{ i \in [d] \setminus \{1\} : i \notin \{\vec{\mathbf{t}}\} \right\}.$$

We then furthermore have that

$$\mathcal{L}_{\mathcal{D}_{\varepsilon,\vec{A}}}(\mathcal{A}'(S(\vec{A}|\vec{\mathbf{t}}))) \geq \underset{\mathbf{t} \sim \mathcal{D}_t}{\mathbb{P}} \left[ \vec{A}_\mathbf{t} \notin R_0(\vec{A}|\vec{\mathbf{t}}), \mathbf{t} \notin I_{\vec{\mathbf{t}}}, \mathbf{t} \neq 1 \right] \tag{19}$$

where we have used that intersection with events only makes probabilities smaller. We will show that there exists a choice of $\vec{A}$ such that the expectation of the right hand side of Equation (19) over the randomness of the training sequence $\mathbf{S}(\vec{A}|\vec{\mathbf{t}})$ is large,

$$\underset{\mathbf{S} \sim \mathcal{D}_{\varepsilon,\vec{A}}^n}{\mathbb{E}} \left[ \underset{\mathbf{t} \sim \mathcal{D}_t}{\mathbb{P}} \left[ \vec{A}_\mathbf{t} \notin R_0(\vec{A}|\vec{\mathbf{t}}), \mathbf{t} \notin I_{\vec{\mathbf{t}}}, \mathbf{t} \neq 1 \right] \right] \geq 2\varepsilon. \tag{20}$$

We notice that Equation (19) and Equation (20) combined gives the claim about the expected loss over the randomness of the training sequence $\mathbf{S}(\vec{A}|\vec{\mathbf{t}})$ in Equation (18). Furthermore, we have by the definition of $\mathcal{D}_t$ that

$$\underset{\mathbf{t} \sim \mathcal{D}_t}{\mathbb{P}} \left[ \vec{A}_\mathbf{t} \notin R_0(\vec{A}|\vec{\mathbf{t}}), \mathbf{t} \notin I_{\vec{\mathbf{t}}}, \mathbf{t} \neq 1 \right] \leq 16\varepsilon.$$

Now by these two observations and the reverse Markov's inequality Lemma C.3 we have that

$$\underset{\mathbf{S} \sim \mathcal{D}_{\varepsilon,\{\vec{A}\}}^n}{\mathbb{P}} \left[ \underset{\mathbf{t} \sim \mathcal{D}_t}{\mathbb{P}} \left[ \vec{A}_\mathbf{t} \notin R_0(\vec{A}|\vec{\mathbf{t}}), \mathbf{t} \notin I_{\vec{\mathbf{t}}}, \mathbf{t} \neq 1 \right] > \varepsilon \right] \geq \frac{\mathbb{E}_{\mathbf{S} \sim \mathcal{D}_{\varepsilon,\vec{A}}^n} \left[ \mathbb{P}_{\mathbf{t} \sim \mathcal{D}_t} \left[ \vec{A}_\mathbf{t} \notin R_0(\vec{A}|\vec{\mathbf{t}}), \mathbf{t} \notin I_{\vec{\mathbf{t}}}, \mathbf{t} \neq 1 \right] \right] - \varepsilon}{16\varepsilon},$$

where we before applying Lemma C.3 have normalized by $16\varepsilon$ since the random variable inside the probability is upper bounded by this value. This combined with the lower bound in Equation (19) and Equation (20) implies that with probability at least $1/16$, the loss $\mathcal{L}_{\mathcal{D}_{\varepsilon,\vec{A}}}(\mathcal{A}'(S(\vec{A}|\vec{\mathbf{t}}))) > \varepsilon$, giving the claim in Equation (17).

We now prove that there exists $\vec{A} \in Q$ which satisfies Equation (20). We first notice that $R_0(\vec{A}, \vec{\mathbf{t}})$ and $I_{\vec{\mathbf{t}}}$ only depend on the randomness of the training sequence $\mathbf{S}(\vec{A}|\vec{\mathbf{t}})$ via $\vec{\mathbf{t}}$, so we have that

$$\underset{\mathbf{S} \sim \mathcal{D}_{\varepsilon,\vec{A}}^n}{\mathbb{E}} \left[ \underset{\mathbf{t} \sim \mathcal{D}_t}{\mathbb{P}} \left[ \vec{A}_\mathbf{t} \notin R_0(\vec{A}|\vec{\mathbf{t}}), \mathbf{t} \notin I_{\vec{\mathbf{t}}}, \mathbf{t} \neq 1 \right] \right] = \underset{\vec{\mathbf{t}} \sim \mathcal{D}_t^n}{\mathbb{E}} \left[ \underset{\mathbf{t} \sim \mathcal{D}_t}{\mathbb{P}} \left[ \vec{A}_\mathbf{t} \notin R_0(\vec{A}|\vec{\mathbf{t}}), \mathbf{t} \notin I_{\vec{\mathbf{t}}}, \mathbf{t} \neq 1 \right] \right] \tag{21}$$

so it suffices to bound the right hand side. We notice that

$$\underset{\vec{\mathbf{t}} \sim \mathcal{D}_t^n}{\mathbb{E}} [|I_{\vec{\mathbf{t}}}|] \leq \underset{\vec{\mathbf{t}} \sim \mathcal{D}_t^n}{\mathbb{E}} \left[ \sum_{i=1}^n \mathbb{1}\{\mathbf{t}_i \in [d] \setminus \{1\}\} \right] = \sum_{i=1}^n \underset{\mathbf{t}_i \sim \mathcal{D}_t}{\mathbb{P}} [\mathbf{t}_i \in [d] \setminus \{1\}] = 16n\varepsilon \leq d/8.$$

where we in the first inequality have used that $I_{\vec{\mathbf{t}}} = \{i \in [d]\backslash\{1\} : \exists j \in [n] : \mathbf{t}_j = i\}$, so it is at most the number of $\mathbf{t}_j$ that are in $[d]\backslash\{1\}$, and in the last inequality we have used the assumption that $n \leq d/(128\varepsilon)$. By Markov's inequality we then have that

$$\mathbb{P}\left[|I_{\vec{\mathbf{t}}}| \geq d/4\right] \leq \frac{1}{2}.$$

Let $E(\vec{\mathbf{t}})$ denote the event that $|I_{\vec{\mathbf{t}}}| < d/4$, which happens with probability at least $1/2$. Let $\vec{\mathbf{A}}$ be drawn uniformly at random from $Q$ (we will denote this $\vec{\mathbf{A}} \sim Q$). By taking expectation over the right hand side of Equation (21) with respect to $\vec{\mathbf{A}}$ we get that

$$\mathbb{E}_{\vec{\mathbf{A}}\sim Q}\left[\mathbb{E}_{\vec{\mathbf{t}}\sim\mathcal{D}_t^n}\left[\mathbb{P}_{\mathbf{t}\sim\mathcal{D}_t}\left[\vec{\mathbf{A}}_{\mathbf{t}} \notin R_0(\vec{\mathbf{A}}|\vec{\mathbf{t}}), \mathbf{t} \notin I_{\vec{\mathbf{t}}}, \mathbf{t} \neq 1\right]\right]\right]$$

$$= \mathbb{E}_{\vec{\mathbf{t}}\sim\mathcal{D}_t^n}\left[\mathbb{E}_{\mathbf{t}\sim\mathcal{D}_t}\left[\mathbb{E}_{\vec{\mathbf{A}}\sim Q}\left[\mathbb{1}\{\vec{\mathbf{A}}_{\mathbf{t}} \notin R_0(\vec{\mathbf{A}}|\vec{\mathbf{t}})\}\mathbb{1}\{\mathbf{t} \notin I_{\vec{\mathbf{t}}}, \mathbf{t} \neq 1\}\right]\right]\right]$$

$$= \mathbb{E}_{\vec{\mathbf{t}}\sim\mathcal{D}_t^n}\left[\mathbb{E}_{\mathbf{t}\sim\mathcal{D}_t}\left[\mathbb{E}_{\vec{\mathbf{A}}\sim Q}\left[\mathbb{1}\{\vec{\mathbf{A}}_{\mathbf{t}} \notin R_0(\vec{\mathbf{A}}|\vec{\mathbf{t}})\}\right]\mathbb{1}\{\mathbf{t} \notin I_{\vec{\mathbf{t}}}, \mathbf{t} \neq 1\}\right]\right]$$

$$\geq \mathbb{E}_{\vec{\mathbf{t}}\sim\mathcal{D}_t^n}\left[\mathbb{E}_{\mathbf{t}\sim\mathcal{D}_t}\left[\mathbb{E}_{\vec{\mathbf{A}}\sim Q}\left[\mathbb{1}\{\vec{\mathbf{A}}_{\mathbf{t}} \notin R_0(\vec{\mathbf{A}}|\vec{\mathbf{t}})\}\right]\mathbb{1}\{\mathbf{t} \notin I_{\vec{\mathbf{t}}}, \mathbf{t} \neq 1\}\mathbb{1}\{E(\vec{\mathbf{t}})\}\right]\right] \tag{22}$$

where the first equality follows from reordering expectations(which can be done since the random variables are independent) and we have written the probability as an expectation over indicators, in the second equality we have moved indicators not depending on $\vec{\mathbf{A}}$ outside the expectation over $\vec{\mathbf{A}}$, and in the last inequality we have intersected with the event $E(\vec{\mathbf{t}})$, which only makes the expectation smaller. Now consider any realization $\vec{t}$ of $\vec{\mathbf{t}}$ where the event $E(\vec{\mathbf{t}})$ holds, and a realization $t$ of $\mathbf{t}$ such that $\mathbf{t} \notin I_{\vec{\mathbf{t}}}$ and $\mathbf{t} \neq 1$, given such realizations $\vec{t}$ and $t$, we will show that

$$\mathbb{E}_{\vec{\mathbf{A}}\sim Q}\left[\mathbb{1}\{\vec{\mathbf{A}}_t \notin R_0(\vec{\mathbf{A}}|\vec{t})\}\right] = \mathbb{P}_{\vec{\mathbf{A}}\sim Q}\left[\vec{\mathbf{A}}_t \notin R_0(\vec{\mathbf{A}}|\vec{t})\right] \geq \frac{1}{2}$$

which combined with Equation (22) implies that

$$\mathbb{E}_{\vec{\mathbf{A}}\sim Q}\left[\mathbb{E}_{\vec{\mathbf{t}}\sim\mathcal{D}_t^n}\left[\mathbb{P}_{\mathbf{t}\sim\mathcal{D}_t}\left[\vec{A}_{\mathbf{t}} \notin R_0(\vec{A}|\vec{\mathbf{t}}), \mathbf{t} \notin I_{\vec{\mathbf{t}}}, \mathbf{t} \neq 1\right]\right]\right] \geq \frac{1}{2}\mathbb{E}_{\vec{\mathbf{t}}\sim\mathcal{D}_t^n}\left[\mathbb{P}_{\mathbf{t}\sim\mathcal{D}_t}[\mathbf{t} \notin I_{\vec{\mathbf{t}}}, \mathbf{t} \neq 1]\mathbb{1}\{E(\vec{\mathbf{t}})\}\right].$$

By using that for any realization $\vec{t}$ of $\vec{\mathbf{t}}$ such that the event $E(\vec{\mathbf{t}})$ holds, it holds that

$$\mathbb{P}_{\mathbf{t}\sim\mathcal{D}_t}[\mathbf{t} \notin I_{\vec{t}}, \mathbf{t} \neq 1] \geq \sum_{i=2}^{d}\mathbb{P}_{\mathbf{t}\sim\mathcal{D}_t}[\mathbf{t} = i]\mathbb{1}\{i \notin I_{\vec{t}}\} = \frac{16\varepsilon}{d-1}\sum_{i=2}^{d}\mathbb{1}\{i \notin I_{\vec{t}}\}$$

$$= \frac{16\varepsilon}{d-1}(d-1-|I_{\vec{t}}|) > 16\varepsilon\left(1 - \frac{d}{4(d-1)}\right) \geq 8\varepsilon,$$

where the last inequality follows from the event $E(\vec{t})$ holding implies $|I_{\vec{t}}| < d/4$ and the assumption that $d \geq 2$ implying that $d/(d-1) \leq 2$. Combining this with us previously concluding that the event $E(\vec{\mathbf{t}})$ holds with probability at least $1/2$, we then have that

$$\mathbb{E}_{\vec{\mathbf{t}}\sim\mathcal{D}_t^n}\left[\mathbb{E}_{\mathbf{t}\sim\mathcal{D}_t}\left[\mathbb{E}_{\vec{\mathbf{A}}\sim Q}\left[\mathbb{1}\{\vec{\mathbf{A}}_{\mathbf{t}} \notin R_0(\vec{\mathbf{A}}|\vec{\mathbf{t}})\}\mathbb{1}\{\mathbf{t} \notin I_{\vec{\mathbf{t}}}, \mathbf{t} \neq 1\}\right]\right]\right] \geq \frac{1}{2}(8\varepsilon)\frac{1}{2} = 2\varepsilon.$$

which concludes the claim in Equation (20). Thus what remains is to show that for any realization $\vec{t}$ of $\vec{\mathbf{t}}$ where the event $E(\vec{\mathbf{t}})$ holds, and a realization $t$ of $\mathbf{t}$ such that $\mathbf{t} \notin I_{\vec{\mathbf{t}}}$ and $\mathbf{t} \neq 1$, it holds that

$$\mathbb{P}_{\vec{\mathbf{A}}\sim Q}\left[\vec{\mathbf{A}}_t \notin R_0(\vec{\mathbf{A}}|\vec{t})\right] \geq \frac{1}{2}. \tag{23}$$

To see this, let $\vec{\mathbf{A}}_{I_{\vec{t}}} = (\vec{\mathbf{A}}_{(I_{\vec{t}})_1}, \ldots, \vec{\mathbf{A}}_{(I_{\vec{t}})_{|I_{\vec{t}}|}})$ be the subvector of $\vec{\mathbf{A}}$ consisting of indices from $I_{\vec{t}}$ and let $\vec{\mathbf{A}}_{I_{\vec{v}}} = (\vec{\mathbf{A}}_{(I_{\vec{v}})_1}, \ldots, \vec{\mathbf{A}}_{(I_{\vec{v}})_{|I_{\vec{v}}|}})$ be the subvector of $\vec{\mathbf{A}}$ consisting of indices from $I_{\vec{v}}$. Now by the law of total expectation we have that

$$\underset{\vec{\mathbf{A}} \sim Q}{\mathbb{P}} \left[ \vec{\mathbf{A}}_t \notin R_0(\vec{\mathbf{A}}|\vec{t}) \right] = \sum_{x \in [k_u]^{|I_{\vec{t}}|}} \underset{\vec{\mathbf{A}} \sim Q}{\mathbb{P}} \left[ \vec{\mathbf{A}}_t \notin R_0(\vec{\mathbf{A}}|\vec{t})|\vec{\mathbf{A}}_{I_{\vec{t}}} = x \right] \underset{\vec{\mathbf{A}}_{I_{\vec{t}}} \sim Q_{I_{\vec{t}}}}{\mathbb{P}} \left[ \vec{\mathbf{A}}_{I_{\vec{t}}} = x \right], \tag{24}$$

where $Q_{I_{\vec{t}}}$ is the distribution over $\vec{\mathbf{A}}_{I_{\vec{t}}}$ induced by $\vec{\mathbf{A}}$ having the uniform distribution on $Q$. We will show that for any fixed $x \in [k_u]^{|I_{\vec{t}}|}$, it holds that

$$\underset{\vec{\mathbf{A}} \sim Q}{\mathbb{P}} \left[ \vec{\mathbf{A}}_t \notin R_0(\vec{\mathbf{A}}|\vec{t})|\vec{\mathbf{A}}_{I_{\vec{t}}} = x \right] \geq \frac{1}{2} \tag{25}$$

which will give Equation (23), by plugging this back into Equation (24).

Now for any fixed $x \in [k_u]^{|I_{\vec{t}}|}$, we have that $R_0(\vec{\mathbf{A}}|\vec{t})$ is fixed since $R_0(\vec{\mathbf{A}}|\vec{t})$ only depends on the values of $\vec{\mathbf{A}}$ on the indices in $I_{\vec{t}}$ and $\vec{\mathbf{A}}_1$ if $1 \in \vec{t}$. Furthermore, since $t \notin I_{\vec{t}}$ and $t \neq 1$, and that the values of $\vec{\mathbf{A}}_{I_{\vec{v}}}$ conditioned on $\vec{\mathbf{A}}_{I_{\vec{t}}}$ is uniform over $[k_u]\backslash(\{\text{values in } x\} \cup \{1\})$, we have that $\vec{\mathbf{A}}_t$ is drawn uniformly at random from $[k_u]\backslash(\{\text{values in } x\} \cup \{1\})$, which has size $k_u - |I_{\vec{t}}| - 1$. We thus have that the probability that $\vec{\mathbf{A}}_t$ falls outside region $R_0(\vec{\mathbf{A}}|\vec{t})$ is at least

$$\underset{\vec{\mathbf{A}} \sim Q}{\mathbb{P}} \left[ \vec{\mathbf{A}}_t \notin R_0(\vec{\mathbf{A}}|\vec{t})|\vec{\mathbf{A}}_{I_{\vec{t}}} = x \right] \geq \frac{k_u - |I_{\vec{t}}| - 1 - |R_0(\vec{\mathbf{A}}|\vec{t})|}{k_u - |I_{\vec{t}}| - 1} \geq 1 - \frac{\max_{n' \leq d/(128\varepsilon)} m(n')d}{k_u - |I_{\vec{t}}| - 1}$$

$$\geq 1 - \frac{\max_{n' \leq d/(128\varepsilon)} m(n')d}{k_u - d/4 - 1} \geq \frac{1}{2}$$

where we in the second inequality have used that the region $R_0(\vec{\mathbf{A}}|\vec{t})$ has size at most $md \leq \max_{n' \leq d/(128\varepsilon)} m(n')d$, the third inequality follows from that $|I_{\vec{t}}| < d/4$, under the event $E(\vec{t})$, and the last inequality follows from the choice of $k_u = \lceil 2 \max_{n' \leq d/(128\varepsilon)} m(n')d + d/4 + 1 \rceil$, which show Equation (25) and concludes the proof. $\qquad \square$

## C.3. Proof of Theorem 3.10

In this section we present the proof of Theorem C.4 which contains both a constant probability lower bound and an expectation lower bound, where the statement in Theorem 3.10 follows from the latter. We now state Theorem C.4.

**Theorem C.4.** *For any $0 < \gamma < 1$ and $0 < \varepsilon < 1$, there exists a hypothesis class $\mathcal{H}$ with $\gamma$-OIG dimension at most 3 such that for any finite aggregation learning algorithm $\mathcal{A}'$ with an interpolating aggregation rule and any $n' \in \mathbb{N}$, there exists a realizable distribution $\mathcal{D}$ such that if $\mathcal{A}'$ receives $n \leq n'$ i.i.d. samples $\mathbf{S} \sim \mathcal{D}^n$, then with probability at least $1/2$ over $\mathbf{S}$, it holds that*

$$\mathcal{L}_{\mathcal{D}}^{\gamma}(\mathcal{A}'(\mathbf{S})) \geq 1 - \varepsilon, \tag{26}$$

*and*

$$\underset{\mathbf{S} \sim \mathcal{D}^n}{\mathbb{E}} [\mathcal{L}_{\mathcal{D}}^{\gamma}(\mathcal{A}'(\mathbf{S}))] \geq 1 - \varepsilon/2, \tag{27}$$

*where the distributions $\mathcal{D}$ in the two cases may be different.*

*Proof of Theorem C.4:* We will consider the universe $\mathcal{X} = \cup_{i \in \mathbb{N}}\{(k, x) : k = i^2, x \leq k\}$. We now construct the hypothesis class $\mathcal{H}$ as follows. To this end, we first notice that the set $\cup_{i \in \mathbb{N}}\{(k, A)|k = i^2, A \subseteq [k], |A| = \sqrt{k}\}$ is countable, implying that for each pair $(k, A)$ we can assign a unique real number $\gamma_{k,A} \in (\gamma, 1]$. Now for each such pair $(k, A)$ we define the hypothesis $h_{k,A} : \mathcal{X} \to \{0, 1\}$ as follows:

$$h_{k,A}(x, y) = \begin{cases} 0 & \text{if } x = k \text{ and } y \in A, \\ \gamma_{k,A} & \text{otherwise.} \end{cases}$$

We define the hypothesis class $\mathcal{H} = \cup_{i \in \mathbb{N}}\{h_{k,A} : k = i^2, A \subseteq [k], |A| = \sqrt{k}\}$.

$\gamma$**-OIG dimension of** $\mathcal{H}$**.** We now show that $\mathcal{H}$ has $\gamma$-OIG dimension at most 3. To see this, consider any set of $n \geq 3$ points $S = \{(x_1, y_1), \ldots (x_n, y_n)\}$, and the $\gamma$ one inclusion graph induced by $S$ and $\mathcal{H}$ which has vertex set $V = \mathcal{H}|_S = \{(h(x_1, y_1), \ldots, h(x_n, y_n)) | h \in \mathcal{H}\}$, and edge set $E$ where an edge is a hyperedge $(f, i)$, containing $v \in V$ if $v((x_j, y_j)) = f((x_j, y_j))$ for all $j \in [n] \backslash i$. Consider any finite subgraph $(V', E')$ of the one inclusion graph. We now show that we can orient the edges of this subgraph such that each vertex has out-degree at most 1, which by $n \geq 3$ would imply that the $\gamma$-OIG-dimension is at most 3 by Definition 3.9. We claim that the orientation where, given an edge $(f, i)$, we orient it towards $v \in (f, i)$ such that $v((x_i, y_i)) < v'((x_i, y_i))$ for all other $v' \in (f, i)$, gives an out-degree of at most 1 for each vertex.

To see this, first consider any vertex $v \in V'$ with at least 2 non-zero values on $S$. Let $v$ correspond to hypothesis $h_{k,A}$. Since $v$ has at least 2 non-zero values, there exist distinct indices $j, l \in [n]$ such that $v(x_j, y_j) = \gamma_{k,A}$ and $v(x_l, y_l) = \gamma_{k,A}$. Consider any edge $(f, i)$ containing $v$. If $i \neq j$, then any neighbor $u \in (f, i)$ must satisfy $u(x_j, y_j) = v(x_j, y_j) = \gamma_{k,A}$. Since $\gamma_{k,A}$ is unique to the pair $(k, A)$, $u$ must also be generated by $h_{k,A}$, implying $u = v$. If $i = j$, then since $n \geq 3$, there is another index $l \neq i$ where $v(x_l, y_l) = \gamma_{k,A}$. Similarly, any neighbor must agree on $l$, forcing the neighbor to have value $\gamma_{k,A}$ at $l$, and thus be $h_{k,A}$. Thus, a vertex with at least 2 non-zero values forms a singleton edge (self-loop) in any direction and has out-degree 0.

Now consider a vertex $v \in V'$ with exactly 1 non-zero value, say at index $j$. So $v(x_j, y_j) = \gamma_{k,A}$ and 0 elsewhere. For any edge $(f, i)$ with $i \neq j$, any neighbor must agree on $j$, so must equal $\gamma_{k,A}$, implying the neighbor is $v$ itself. The only direction where an edge can contain other vertices is $i = j$. In this edge, $v$ has value $\gamma_{k,A} > 0$. If there exists a neighbor $u$ with value smaller than $\gamma_{k,A}$ at $i$, the edge is oriented towards $u$. If all neighbors have values larger than $\gamma_{k,A}$, it is oriented towards $v$. Thus, $v$ has out-degree at most 1.

Finally, consider a vertex $v \in V'$ with 0 non-zero values (i.e., the all-zero hypothesis). Since the edge orientation directs edges towards the vertex with the smallest value on index $i$, and 0 is the minimum possible value, any edge $(f, i)$ containing $v$ must be oriented towards $v$. Consequently, such a vertex has an out-degree of 0.

Thus we have shown that the orientation gives an out-degree of at most 1 for each vertex in the $\gamma$ - one inclusion graph, and thus for $n \geq 3$ the out-degree is at most $1 \leq n/3$ implying that the $\gamma$-OIG-dimension of $\mathcal{H}$ is at most 3.

**Lower bounds.** We now show that for any finite aggregation learning algorithm $\mathcal{A}'$ there exists a realizable distribution $\mathcal{D}$ such that if $\mathcal{A}'$ receives $\mathbf{S} \sim \mathcal{D}^n$, $n \leq n'$, then

$$\underset{\mathbf{S} \sim \mathcal{D}^n}{\mathbb{E}} [\mathcal{L}_{\mathcal{D}}^{\gamma}(\mathcal{A}'(\mathbf{S}))] \geq 1 - \varepsilon/2 \tag{28}$$

which implies Equation (27). By the reverse Markov inequality (Lemma C.3), this further implies

$$\mathbb{P}[\mathcal{L}_{\mathcal{D}}^{\gamma}(\mathcal{A}'(\mathbf{S})) > 1 - \varepsilon] \geq \frac{\mathbb{E}_{\mathbf{S} \sim \mathcal{D}^n}[\mathcal{L}_{\mathcal{D}}^{\gamma}(\mathcal{A}'(\mathbf{S}))] - (1 - \varepsilon)}{1 - (1 - \varepsilon)} \geq \frac{1}{2},$$

which implies Equation (26). Thus it suffices to prove the expected loss lower bound in Equation (28). To this end, let $\mathcal{A}'$ be any finite aggregation learning algorithm given $n \leq n'$ training examples. We construct a family of realizable distributions. We choose $k_u$, which will be the effective universe size for the distributions we will construct, as $k_u = i^2$ for $i \in \mathbb{N}$, with $i$ large enough such that

$$\left(1 - \frac{n'}{\sqrt{k_u}}\right) \left(1 - \frac{\sqrt{k_u} \max_{n \in [n']} m(n)}{k_u}\right) \geq 1 - \varepsilon/2. \tag{29}$$

Each distribution is indexed by a vector in the set:

$$Q = \left\{ \vec{A} \in [k_u]^{\sqrt{k_u}} \mid \forall i, j \in [\sqrt{k_u}], i \neq j \implies \vec{A}_i \neq \vec{A}_j \right\}.$$

The distribution $\mathcal{D}_{\vec{A}}$ for each $\vec{A} \in Q$ is defined as:

$$\mathcal{D}_{\vec{A}}(x, y, z) = \begin{cases} \frac{1}{\sqrt{k_u}} & \text{if } x = k_u, y = \vec{A}_i, z = 0, \text{ for some } i \in [\sqrt{k_u}], \\ 0 & \text{otherwise.} \end{cases}$$

We notice that each distribution $\mathcal{D}_{\vec{A}}$ is realizable by the hypothesis $h_{k_u, \{\vec{A}\}} \in \mathcal{H}$, where $\{\vec{A}\} = \{\vec{A}_1, \ldots, \vec{A}_{\sqrt{k_u}}\}$. We notice that sampling from $\mathcal{D}_{\vec{A}}$ can be seen as first sampling $\mathbf{t} \sim \mathcal{D}_t$, where $\mathcal{D}_t$ is the uniform distribution over $[\sqrt{k_u}]$, and then returning $(k_u, \vec{A}_\mathbf{t}, 0)$. Furthermore using this perspective the distribution of training sequence $\mathbf{S} \sim \mathcal{D}_{\vec{A}}^n$ is the same as $\mathbf{S}(\vec{A}, \mathbf{t}) = ((k_u, \vec{A}_{\mathbf{t}_1}, 0), \ldots, (k_u, \vec{A}_{\mathbf{t}_n}, 0))$ where $\mathbf{t}_i$ are i.i.d. from $\mathcal{D}_t$. Using this view of the sampling process we have that

$$\mathcal{L}_{\mathcal{D}_{\vec{A}}}^\gamma(\mathcal{A}'(\mathbf{S})) = \mathop{\mathbb{P}}_{(x,y,z) \sim \mathcal{D}_{\vec{A}}} [|\mathcal{A}'(\mathbf{S})(x,y) - z| > \gamma] = \mathop{\mathbb{P}}_{\mathbf{t} \sim \mathcal{D}_t} \left[ |\mathcal{A}'(\mathbf{S})(k_u, \vec{A}_\mathbf{t}) - 0| > \gamma \right]$$

$$= \frac{|\{i \in \{\vec{A}\} : \mathcal{A}'(\mathbf{S})(k_u, i) > \gamma\}|}{\sqrt{k_u}}.$$

Taking expectation over $\mathbf{S}$:

$$\mathop{\mathbb{E}}_{\mathbf{S} \sim \mathcal{D}_{\vec{A}}^n}[\mathcal{L}_{\mathcal{D}_{\vec{A}}}^\gamma(\mathcal{A}'(\mathbf{S}))] = \mathop{\mathbb{E}}_{\vec{\mathbf{t}} \sim \mathcal{D}_t^n} \left[ \frac{|\{i \in \{\vec{A}\} : \mathcal{A}'(\mathbf{S}(\vec{A}, \vec{\mathbf{t}}))(k_u, i) > \gamma\}|}{\sqrt{k_u}} \right].$$

Furthermore letting $\vec{\mathbf{A}} \sim Q$ denote a random vector sampled uniformly from $Q$, we have that

$$\mathop{\mathbb{E}}_{\vec{\mathbf{A}} \sim Q} \left[ \mathop{\mathbb{E}}_{\mathbf{S} \sim \mathcal{D}_{\vec{\mathbf{A}}}^n}[\mathcal{L}_{\mathcal{D}_{\vec{\mathbf{A}}}}^\gamma(\mathcal{A}'(\mathbf{S}))] \right] = \mathop{\mathbb{E}}_{\vec{\mathbf{A}} \sim Q} \left[ \mathop{\mathbb{E}}_{\vec{\mathbf{t}} \sim \mathcal{D}_t^n} \left[ \frac{|\{i \in \{\vec{\mathbf{A}}\} : \mathcal{A}'(\mathbf{S}(\vec{\mathbf{A}}, \vec{\mathbf{t}}))(k_u, i) > \gamma\}|}{\sqrt{k_u}} \right] \right]$$

$$= \frac{1}{\sqrt{k_u}} \mathop{\mathbb{E}}_{\vec{\mathbf{t}} \sim \mathcal{D}_t^n} \left[ \mathop{\mathbb{E}}_{\vec{\mathbf{A}} \sim Q} \left[ |\{i \in \{\vec{\mathbf{A}}\} : \mathcal{A}'(\mathbf{S}(\vec{\mathbf{A}}, \vec{\mathbf{t}}))(k_u, i) > \gamma\}| \right] \right]$$

and thus if we can lower bound the last expectation by $\sqrt{k_u}(1 - \varepsilon/2)$, then the above is lower bounded by $1 - \varepsilon/2$. Hence, there exists a vector $\vec{A} \in Q$ such that when $\mathbf{S} \sim \mathcal{D}_{\vec{A}}^n$, the corresponding loss is also lower bounded by $1 - \varepsilon/2$, implying Equation (28). We will show the claimed lower bound by showing that for any realization $\vec{t}$ of $\vec{\mathbf{t}}$, it holds that

$$\mathop{\mathbb{E}}_{\vec{\mathbf{A}} \sim Q} \left[ |\{i \in \{\vec{\mathbf{A}}\} : \mathcal{A}'(\mathbf{S}(\vec{\mathbf{A}}, \vec{t}))(k_u, i) > \gamma\}| \right] \geq \sqrt{k_u}(1 - \varepsilon/2). \tag{30}$$

To this end, consider any realization $\vec{t}$ of $\vec{\mathbf{t}}$. Let's define the set of indices seen and not seen in $\vec{t}$ as

$$I_{\vec{t}} = \left\{ j \in [\sqrt{k_u}] : j \in \{\vec{t}\} \right\} \quad \text{and} \quad I_{\vec{t}'} = \left\{ j \in [\sqrt{k_u}] : j \notin \{\vec{t}\} \right\}.$$

and let $\vec{\mathbf{A}}_{I_{\vec{t}'}} = (\vec{\mathbf{A}}_{(I_{\vec{t}'})_1}, \ldots, \vec{\mathbf{A}}_{(I_{\vec{t}'})_{|I_{\vec{t}'}|}})$, be the subvector of $\vec{\mathbf{A}}$ consisting of indices from $I_{\vec{t}'}$ and let $\vec{\mathbf{A}}_{I_{\vec{t}}} = (\vec{\mathbf{A}}_{(I_{\vec{t}})_1}, \ldots, \vec{\mathbf{A}}_{(I_{\vec{t}})_{|I_{\vec{t}}|}})$ be the subvector of $\vec{\mathbf{A}}$ consisting of indices from $I_{\vec{t}}$. By the law of total probability we have that

$$\mathop{\mathbb{E}}_{\vec{\mathbf{A}} \sim Q} \left[ |\{i \in \{\vec{\mathbf{A}}\} : \mathcal{A}'(\mathbf{S}(\vec{\mathbf{A}}, \vec{t}))(k_u, i) > \gamma\}| \right]$$

$$= \sum_{x \in [k_u]^{|I_{\vec{t}}|}} \mathop{\mathbb{E}}_{\vec{\mathbf{A}} \sim Q} \left[ |\{i \in \{\vec{\mathbf{A}}\} : \mathcal{A}'(\mathbf{S}(\vec{\mathbf{A}}, \vec{t}))(k_u, i) > \gamma\}| \Big| \vec{\mathbf{A}}_{I_{\vec{t}}} = x \right] \mathop{\mathbb{P}}_{\vec{\mathbf{A}}_{I_{\vec{t}}} \sim Q_{I_{\vec{t}}}} \left[ \vec{\mathbf{A}}_{I_{\vec{t}}} = x \right],$$

where $Q_{I_{\vec{t}}}$ is the distribution over $\vec{\mathbf{A}}_{I_{\vec{t}}}$ induced by $\vec{\mathbf{A}}$ having the uniform distribution on $Q$.

We will now show that for any fixed $x$ with distinct entries (the only ones that has non zero mass in the above sum) the conditional expectation is lower bounded by $\sqrt{k_u}(1 - \varepsilon/2)$, which implies Equation (30). To this end, consider any fixed $x \in [k_u]^{|I_{\vec{t}}|}$ with distinct entries. We notice that for any fixed $\vec{\mathbf{A}}_{I_{\vec{t}}} = x$, the vector $\mathbf{S}(\vec{\mathbf{A}}, \vec{t}) = r((k_u, \vec{\mathbf{A}}_{t_1}, 0), \ldots, (k_u, \vec{\mathbf{A}}_{t_n}, 0), x)$ is fixed as it only depends on the entries of $\vec{\mathbf{A}}$ indexed by $I_{\vec{t}}$, which are fixed to $\vec{\mathbf{A}}_{I_{\vec{t}}} = x$. This furthermore implies that $\mathcal{A}'(\mathbf{S}(\vec{\mathbf{A}}, \vec{t}))$ is fixed as well.

By the definition of finite aggregation learning algorithm using a interpolating aggregation rule, we have that there exists a finite set of hypotheses $\{h_{k_1, A_1}, \ldots, h_{k_m, A_m}\}$ in $\mathcal{H}$ such that $\mathcal{A}'(\mathbf{S}(\vec{\mathbf{A}}, \vec{t})) = r(h_{k_1, A_1}, \ldots, h_{k_m, A_m}, x)$, where

$m \leq m(n) \leq \max_{n \in [n']} m(n) < \infty$. Furthermore, by construction of $\mathcal{H}$, each hypothesis $h_{k_i, A_i}$ can output a zero value at most $\sqrt{k_u}$ times on $[k_u]$ otherwise its output is strictly larger than $\gamma$. This implies combined with the aggregation rule being interpolating so outputting a value between the minimum and the maximum of the aggregated values implies that $R_{\gamma} = \{i \in [k_u] : \mathcal{A}'(\mathbf{S}(\vec{\mathbf{A}}, \vec{t}))(k_u, i) > \gamma\} = \{i \in [k_u] : r(h_{k_1, A_1}(k_u, i), \dots, h_{k_m, A_m}(k_u, i), x) > \gamma\}$ is at least of size $k_u - \sqrt{k_u} \max_{n \in [n']} m(n)$. Using this we have that

$$|\{i \in \{\vec{\mathbf{A}}\} : \mathcal{A}'(\mathbf{S}(\vec{\mathbf{A}}, \vec{t}))(k_u, i) > \gamma\}| = |\{\vec{\mathbf{A}}\} \cap R_{\gamma}| \geq |\{\vec{\mathbf{A}}_{I_{\vec{v}}}\} \cap R_{\gamma}|$$

We furthermore notice that conditioned on $\vec{\mathbf{A}}_{I_{\vec{t}}} = x$, the vector $\vec{\mathbf{A}}_{I_{\vec{v}}}$ is uniformly distributed over all vectors in $([k_u] \backslash \{x\})^{|I_{\vec{v}}|}$ with entries being distinct. This implies that

$$\mathbb{E}_{\vec{\mathbf{A}} \sim Q} \left[ |\{i \in \{\vec{\mathbf{A}}\} : \mathcal{A}'(\mathbf{S}(\vec{\mathbf{A}}, \vec{t}))(k_u, i) > \gamma\}| \Big| \vec{\mathbf{A}}_{I_{\vec{t}}} = x \right] \geq \mathbb{E}_{\vec{\mathbf{A}} \sim Q} \left[ |\{\vec{\mathbf{A}}_{I_{\vec{v}}}\} \cap R_{\gamma}| \Big| \vec{\mathbf{A}}_{I_{\vec{t}}} = x \right]$$

$$= \sum_{i=1}^{|I_{\vec{v}}|} \mathbb{P}_{\vec{\mathbf{A}} \sim Q} \left[ \vec{\mathbf{A}}_{(I_{\vec{v}})_i} \in R_{\gamma} \Big| \vec{\mathbf{A}}_{I_{\vec{t}}} = x \right] = \sum_{i=1}^{|I_{\vec{v}}|} \frac{|R_{\gamma}|}{|[k_u] \backslash \{x\}|} \geq |I_{\vec{v}}| \frac{k_u - \sqrt{k_u} \max_{n \in [n']} m(n)}{k_u - |I_{\vec{t}}|}.$$

where we in the last inequality used that $|[k_u] \backslash \{x\}| \geq |[k_u]| - |\{x\}| = k_u - |I_{\vec{t}}|$ and that $|R_{\gamma}| \geq k_u - \sqrt{k_u} \max_{n \in [n']} m(n)$. Since $|I_{\vec{t}}| \leq \sqrt{k_u}$ and $|I_{\vec{v}}| = \sqrt{k_u} - |I_{\vec{t}}| \geq \sqrt{k_u} - n'$, we have that

$$\mathbb{E}_{\vec{\mathbf{A}} \sim Q} \left[ |\{i \in \{\vec{\mathbf{A}}\} : \mathcal{A}'(\mathbf{S}(\vec{\mathbf{A}}, \vec{t}))(k_u, i) > \gamma\}| \Big| \vec{\mathbf{A}}_{I_{\vec{t}}} = x \right] \geq (\sqrt{k_u} - n') \frac{k_u - \sqrt{k_u} \max_{n \in [n']} m(n)}{k_u}$$

$$\geq \sqrt{k_u} \left( 1 - \frac{n'}{\sqrt{k_u}} \right) \left( 1 - \frac{\sqrt{k_u} \max_{n \in [n']} m(n)}{k_u} \right).$$

which by the choice of $k_u$ in Equation (29) is at least $\sqrt{k_u}(1 - \varepsilon/2)$, and as mentioned implies Equation (30) completes the proof.

$\square$

### C.4. Proof of Theorem 3.12

In this section we present the proof of Theorem C.5 which contains both a constant probability lower bound and expectation lower bound, where the statement in Theorem 3.12 follows from the latter. We now state Theorem C.5 and then give its proof.

**Theorem C.5.** *For any $0 < \gamma < 1$, and $d_{\gamma} \geq 2$, there exists a hypothesis class $\mathcal{H}$ with $\gamma$-graph dimension $d_{\gamma}$ such that for any proper learning algorithm $\mathcal{A}$, and for any $0 < \varepsilon < \frac{1}{64e}$, there exists a realizable distribution $\mathcal{D}$ such that if $\mathcal{A}$ receives $n \leq \frac{d_{\gamma}}{32\varepsilon} \ln \left( \frac{1}{64e\varepsilon} \right)$ i.i.d. samples $\mathbf{S} \sim \mathcal{D}^n$ from $\mathcal{D}$, then with probability at least $1/48$ over $\mathbf{S}$, it holds that*

$$\mathcal{L}_{\mathcal{D}}^{\gamma}(\mathcal{A}(\mathbf{S})) > \varepsilon, \tag{31}$$

*and*

$$\mathbb{E}_{\mathbf{S} \sim \mathcal{D}^n} [\mathcal{L}_{\mathcal{D}}^{\gamma}(\mathcal{A}(\mathbf{S}))] \geq 4\varepsilon/3. \tag{32}$$

*where the distributions $\mathcal{D}$ in the two cases may be different.*

*Proof of Theorem C.5.* Let $0 < \gamma < 1$, $d \in \mathbb{N}$, and let the input space be $\mathcal{X} = \bigcup_{k=d-1}^{\infty} \{(k, x) : x \leq k\}$. We start constructing the hypothesis class $\mathcal{H}^{\gamma, d}$. To this end, we first notice that $\bigcup_{k=d-1}^{\infty} \{(k, A) : A \subseteq [k], |A| = d - 1\}$ is a countably infinite set. Thus, for each $(k, A)$ in this set, we can define a unique $\gamma_{k, A} \in (\gamma, 1]$, which we will do in the following. We will now map each of these tuples to a hypothesis, where the union of these hypotheses will constitute our hypothesis class. Now for integers $k \geq d - 1$ and $A \subseteq [k]$, where $|A| = d - 1$, we define the hypothesis $h_{k, A} : \mathcal{X} \to [0, 1]$ as

$$h_{k, A}(x, y) = \begin{cases} 0 & \text{if } x = k, y \in [k] \backslash A, \\ \gamma_{k, A} & \text{else}. \end{cases}$$

We then define the hypothesis class $\mathcal{H}^{\gamma, d} = \bigcup_{k=d-1}^{\infty} \{h_{k, A} : A \subseteq [k], |A| = d - 1\}$.

$\gamma$-**graph dimension of** $\mathcal{H}^{\gamma,d}$. We now notice that the hypothesis class $\mathcal{H}^{\gamma,d}$ has $\gamma$-graph dimension at least $d$, since the set $\{(2d,y) : 1 \le y \le d\}$ is $\gamma$-graph shattered by $\mathcal{H}^{\gamma,d}$, with the witness hypothesis $h_{2d,\{d+2,\ldots,2d\}}$. To see why, we first notice that $h_{2d,\{d+2,\ldots,2d\}}$ is all 0 on $\{(2d,y) : 1 \le y \le d\}$, so it suffices to show that for any $B \subseteq [d]$ there is a hypothesis $h \in \mathcal{H}^{\gamma,d}$, such that $h((2d,y)) > \gamma$ for $y \in B$ and $h((2d,y)) = 0$ for $y \in [d]\backslash B$. Thus if $B = [d]$, then $h_{2d+1,\{1,\ldots,d-1\}}$ satisfies that $h_{2d+1,\{1,\ldots,d-1\}}((2d,y)) = \gamma_{2d+1,\{1,\ldots,d-1\}} > \gamma$, for $y \in B$ as wanted. Otherwise, if $|B| \le d-1$, the hypothesis $h_{2d,B\cup B'}$, where $B' \subseteq \{d+1,\ldots,2d\}$ with $|B'| = d-1-|B|$, satisfies that for any $y \in B$, $h_{2d,B\cup B'}(2d,y) = \gamma_{2d,B\cup B'} > \gamma$, and for any $y \in [d]\backslash B$, $h_{2d,B\cup B'}(2d,y) = 0$, as wanted. Thus $\mathcal{H}^{\gamma,d}$ has $\gamma$-graph dimension at least $d$.

We now show that the hypothesis class $\mathcal{H}^{\gamma,d}$ has $\gamma$-graph dimension at most $d$. Let $S = \{(x_1,y_1),\ldots,(x_{d+1},y_{d+1})\} \subseteq \mathcal{X}$ be any set of size $d+1$ and assume for the sake of contradiction that $\mathcal{H}^{\gamma,d}$ shatters $S$ with $h_{k,A} \in \mathcal{H}^{\gamma,d}$ as the witness hypothesis. We first observe that if there exists an $i \in [d+1]$ with $h_{k,A}((x_i,y_i)) = \gamma_{k,A} > 0$, then by uniqueness of $\gamma_{k,A}$, it must hold that if $h_{k',A'}((x_i,y_i)) = h_{k,A}((x_i,y_i))$, then $(k',A') = (k,A)$, implying that for $j \in [d+1]$ with $j \ne i$ we can not have $h_{k',A'}((x_i,y_i)) = h_{k,A}((x_i,y_i))$ and $|h_{k',A'}((x_j,y_j)) - h_{k,A}((x_j,y_j))| > \gamma$ simultaneously, contradicting that $h_{k,A}$ is a witness hypothesis of the $\gamma$-shattering. Now for the case that $h_{k,A}((x_i,y_i)) = 0$ for all $i \in [d+1]$, it must be the case that $x = x_1 = \ldots = x_{d+1}$, furthermore implying that $x \ge d+1$, since otherwise the points in $S$ could not be distinct, which is required for them to be shattered. We now claim that for $B = \{1,\ldots,d\}$, there is no hypothesis $h_{k',A'} \in \mathcal{H}^{\gamma,d}$, such that for any $i \in B$, $h_{k',A'}((x,y_i)) > \gamma$, and $h_{k',A'}((x,y_{d+1})) = 0$. To see this we first notice that any $h_{k',A'}$ with $k' \ne x$ would fail to comply with $h_{k',A'}((x,y_{d+1})) = 0$, since in this case $h_{k',A'}((x,y_{d+1})) = \gamma_{k',A'} > 0$. Furthermore in the case that $k' = x$, since $|A'| = d-1$, there exists at least one $i \in B$ such that $y_i \notin A'$, implying that $h_{k',A'}((x,y_i)) = 0$, contradicting that $h_{k',A'}((x,y_i)) > \gamma$. Thus we have shown that $\mathcal{H}^{\gamma,d}$ has $\gamma$-graph dimension exactly $d$.

**Lower bounds.** We now define a family of distributions wherein we for any proper learning algorithm $\mathcal{A}$ will find a hard distribution. To this end let

$$k_u = \left\lceil \frac{d}{16\varepsilon} \right\rceil$$

be the choice of effective universe size (the place where our distributions will have support inside). We remark that this choice of $k_u$ implies the following inequalities, useful for the proof:

$$\varepsilon(k_u - d + 1) \le \varepsilon\left(\frac{d}{16\varepsilon} - d + 2\right) \le \frac{d}{16} \tag{33}$$

$$k_u \ge \frac{d}{16\varepsilon} > 2d + 1 \tag{34}$$

where in the first line we have used that $\lceil x \rceil \le x + 1$ and $d \ge 2$ and in the second line we have used that $\varepsilon < 1/(64e)$ and $d \ge 2$. Furthermore let

$$Q = \left\{\vec{A} \in [k_u]^{k_u - d + 1} : \forall i,j \in [k_u - d + 1], i \ne j, \vec{A}_i \ne \vec{A}_j\right\}$$

where each of these vectors will be used to define a distribution $\mathcal{D}_{\vec{A}}$ as follows. For any $\vec{A} \in Q$, we define the distribution $\mathcal{D}_{\vec{A}}$ over $\mathcal{X} \times [0,1]$ as

$$\mathcal{D}_{\vec{A}}(((x,y),0)) = \begin{cases} \frac{1}{k_u - d + 1} & \text{if } x = k_u, y \in \vec{A}, \\ 0 & \text{else .} \end{cases}$$

We notice that $\mathcal{D}_{\vec{A}}$ is realizable by $\mathcal{H}^{\gamma,d}$, since $h_{k_u,\{\vec{A}^c\}}$, where $\{\vec{A}^c\} = \left\{i \in [k_u] \mid \forall j \in [k_u - d + 1] : i \ne \vec{A}_j\right\}$ is constant zero on the support of $\mathcal{D}_{\vec{A}}$. We notice that sampling from $\mathcal{D}_{\vec{A}}$ can also be seen as sampling $\mathbf{t} \sim \mathcal{D}_t$ with $\mathbb{P}_{\mathcal{D}}[\mathbf{t} = i] = \frac{1}{k_u - d + 1}$ for $i \in [k_u - d + 1]$, and then returning the sample $((k_u, \vec{A}_\mathbf{t}), 0)$. From now on we will identify samples from $\mathcal{D}_{\vec{A}}$ with samples from $\mathcal{D}_t$ via this mechanism. Thus we will let $\mathbf{S}(\vec{A}|\vec{\mathbf{t}}) = (((k_u, \vec{A}_{\mathbf{t}_1}), 0), \ldots, ((k_u, \vec{A}_{\mathbf{t}_n}), 0))$ be the training sequence of size $n$ sampled i.i.d from $\mathcal{D}_{\vec{A}}$ using the random vector $\vec{\mathbf{t}} = (\mathbf{t}_1, \ldots, \mathbf{t}_n)$. This perspective also implies that for a learning algorithm $\mathcal{A}$,

$$\mathcal{L}^\gamma_{\mathcal{D}_{\vec{A}}}(\mathcal{A}(\mathbf{S}(\vec{A}|\vec{\mathbf{t}}))) = \mathbb{P}_{((\mathbf{x},\mathbf{y}),\mathbf{z}) \sim \mathcal{D}_{\vec{A}}}[|\mathcal{A}(\mathbf{S}(\vec{A}|\vec{\mathbf{t}}))(\mathbf{x},\mathbf{y}) - \mathbf{z}| > \gamma] = \mathbb{P}_{\mathbf{t} \sim \mathcal{D}_t}[\mathcal{A}(\mathbf{S}(\vec{A}|\vec{\mathbf{t}}))(k_u, \vec{A}_\mathbf{t}) > \gamma]$$

where the second equality follows from $\mathbf{z} = 0$ given the definition of $\mathcal{D}_{\vec{A}}$. We will start by showing Equation (31) and then show Equation (32). Let from now on $\mathcal{A}$ be any proper learning algorithm. Now using the above observation about how to rewrite the loss, and using the alternative sampling procedure of $\mathbf{S}(\vec{A}, \vec{\mathbf{t}})$, we have that for any $\vec{A} \in Q$,

$$\mathop{\mathbb{P}}_{\mathbf{S} \sim \mathcal{D}_{\vec{A}}^n} \left[ \mathcal{L}_{\mathcal{D}_{\vec{A}}}^\gamma (\mathcal{A}(\mathbf{S})) > \varepsilon \right] = \mathop{\mathbb{P}}_{\vec{\mathbf{t}} \sim \mathcal{D}_t^n} \left[ \mathop{\mathbb{P}}_{\mathbf{t} \sim \mathcal{D}_t} [\mathcal{A}(\mathbf{S}(\vec{A}|\vec{\mathbf{t}}))(k_u, \vec{A}_{\mathbf{t}}) > \gamma] > \varepsilon \right].$$

Using this and letting $\vec{\mathbf{A}}$ being drawn uniformly from $Q$ (denoted as $\vec{\mathbf{A}} \sim Q$ ), we have that

$$\mathop{\mathbb{P}}_{\vec{\mathbf{A}} \sim Q} \left[ \mathop{\mathbb{P}}_{\mathbf{S} \sim \mathcal{D}_{\vec{\mathbf{A}}}^n} \left[ \mathcal{L}_{\mathcal{D}_{\vec{\mathbf{A}}}}^\gamma (\mathcal{A}(\mathbf{S})) > \varepsilon \right] \right] = \mathop{\mathbb{P}}_{\vec{\mathbf{A}} \sim Q} \left[ \mathop{\mathbb{P}}_{\vec{\mathbf{t}} \sim \mathcal{D}_t^n} \left[ \mathop{\mathbb{P}}_{\mathbf{t} \sim \mathcal{D}_t} [\mathcal{A}(\mathbf{S}(\vec{\mathbf{A}}|\vec{\mathbf{t}}))(k_u, \vec{\mathbf{A}}_{\mathbf{t}}) > \gamma] > \varepsilon \right] \right]$$

$$= \mathop{\mathbb{P}}_{\vec{\mathbf{t}} \sim \mathcal{D}_t^n} \left[ \mathop{\mathbb{P}}_{\vec{\mathbf{A}} \sim Q} \left[ \mathop{\mathbb{P}}_{\mathbf{t} \sim \mathcal{D}_t} [\mathcal{A}(\mathbf{S}(\vec{\mathbf{A}}|\vec{\mathbf{t}}))(k_u, \vec{\mathbf{A}}_{\mathbf{t}}) > \gamma] > \varepsilon \right] \right]. \tag{35}$$

We will show that the last term in the above is lower bounded by $1/48$, i.e.

$$\mathop{\mathbb{P}}_{\vec{\mathbf{t}} \sim \mathcal{D}_t^n} \left[ \mathop{\mathbb{P}}_{\vec{\mathbf{A}} \sim Q} \left[ \mathop{\mathbb{P}}_{\mathbf{t} \sim \mathcal{D}_t} [\mathcal{A}(\mathbf{S}(\vec{\mathbf{A}}|\vec{\mathbf{t}}))(k_u, \vec{A}_{\mathbf{t}}) > \gamma] > \varepsilon \right] \right] \geq \frac{1}{48}, \tag{36}$$

implying that there exists a $\vec{A} \in Q$ such that

$$\mathop{\mathbb{P}}_{\mathbf{S} \sim \mathcal{D}_{\vec{A}}^n} \left[ \mathcal{L}_{\mathcal{D}_{\vec{A}}}^\gamma (\mathcal{A}(\mathbf{S})) > \varepsilon \right] \geq \frac{1}{48},$$

further implying the statement in Equation (31). Thus we now lower bound Equation (36). To this end, let's define the set of indices seen and not seen in $\vec{\mathbf{t}}$

$$I_{\vec{\mathbf{t}}} = \left\{ j \in [k_u - d + 1] : j \notin \{\vec{\mathbf{t}}\} \right\} \quad \text{and} \quad I_{\vec{\mathbf{t}}} = \left\{ j \in [k_u - d + 1] : j \in \{\vec{\mathbf{t}}\} \right\}.$$

We then have that

$$\mathop{\mathbb{P}}_{\vec{\mathbf{t}} \sim \mathcal{D}_t^n} \left[ \mathop{\mathbb{P}}_{\vec{\mathbf{A}} \sim Q} \left[ \mathop{\mathbb{P}}_{\mathbf{t} \sim \mathcal{D}_t} [\mathcal{A}(\mathbf{S}(\vec{\mathbf{A}}|\vec{\mathbf{t}}))(k_u, \vec{A}_{\mathbf{t}}) > \gamma] > \varepsilon \right] \right] \geq \mathop{\mathbb{P}}_{\vec{\mathbf{t}} \sim \mathcal{D}_t^n} \left[ \mathop{\mathbb{P}}_{\vec{\mathbf{A}} \sim Q} \left[ \mathop{\mathbb{P}}_{\mathbf{t} \sim \mathcal{D}_t} [\mathcal{A}(\mathbf{S}(\vec{\mathbf{A}}|\vec{\mathbf{t}}))(k_u, \vec{A}_{\mathbf{t}}) > \gamma] > \varepsilon \right] \mathbb{1}\{|I_{\vec{\mathbf{t}}}| \geq d\} \right].$$

We will show that for any realization $\vec{t}$ of $\vec{\mathbf{t}}$ with $|I_{\vec{t}}| \geq d$, it holds that

$$\mathop{\mathbb{P}}_{\vec{\mathbf{A}} \sim Q} \left[ \mathop{\mathbb{P}}_{\mathbf{t} \sim \mathcal{D}_t} [\mathcal{A}(\mathbf{S}(\vec{\mathbf{A}}|\vec{t}))(k_u, \vec{A}_{\mathbf{t}}) > \gamma] > \varepsilon \right] \geq \frac{1}{24}, \tag{37}$$

implying that

$$\mathop{\mathbb{P}}_{\vec{\mathbf{t}} \sim \mathcal{D}_t^n} \left[ \mathop{\mathbb{P}}_{\vec{\mathbf{A}} \sim Q} \left[ \mathop{\mathbb{P}}_{\mathbf{t} \sim \mathcal{D}_t} [\mathcal{A}(\mathbf{S}(\vec{\mathbf{A}}|\vec{\mathbf{t}}))(k_u, \vec{A}_{\mathbf{t}}) > \gamma] > \varepsilon \right] \right] \geq \frac{1}{24} \mathop{\mathbb{P}}_{\vec{\mathbf{t}} \sim \mathcal{D}_t^n} \left[ |I_{\vec{\mathbf{t}}}| \geq d \right]. \tag{38}$$

Lastly we will show that

$$\mathop{\mathbb{P}}_{\vec{\mathbf{t}} \sim \mathcal{D}_t^n} \left[ |I_{\vec{\mathbf{t}}}| \geq d \right] \geq 1/2, \tag{39}$$

implying Equation (36) by combining the above two inequalities in Equation (38) and Equation (39). We first show Equation (37) and then Equation (39). To this end let $\vec{t}$ be any realization of $\vec{\mathbf{t}}$ with $|I_{\vec{t}}| \geq d$. We have that

$$\mathop{\mathbb{P}}_{\vec{\mathbf{A}} \sim Q} \left[ \mathop{\mathbb{P}}_{\mathbf{t} \sim \mathcal{D}_t} [\mathcal{A}(\mathbf{S}(\vec{\mathbf{A}}|\vec{t}))(k_u, \vec{A}_{\mathbf{t}}) > \gamma] > \varepsilon \right] = \mathop{\mathbb{P}}_{\vec{\mathbf{A}} \sim Q} \left[ \sum_{i \in \{\vec{\mathbf{A}}\}} \frac{\mathbb{1}\{\mathcal{A}(\mathbf{S}(\vec{\mathbf{A}}|\vec{t}))(k_u, i) > \gamma\}}{k_u - d + 1} > \varepsilon \right] \tag{40}$$

$$= \mathop{\mathbb{P}}_{\vec{\mathbf{A}} \sim Q} \left[ |\{i \in \{\vec{\mathbf{A}}\} : \mathcal{A}(\mathbf{S}(\vec{\mathbf{A}}|\vec{t}))(k_u, i) > \gamma\}| > \varepsilon (k_u - d + 1) \right] \geq \mathop{\mathbb{P}}_{\vec{\mathbf{A}} \sim Q} \left[ |\{i \in \{\vec{\mathbf{A}}\} : \mathcal{A}(\mathbf{S}(\vec{\mathbf{A}}|\vec{t}))(k_u, i) > \gamma\}| > \frac{d}{8} \right]$$

where the last inequality follows from Equation (33). Let $\vec{\mathbf{A}}_{I_{\vec{v}}} = (\mathbf{A}_{(I_{\vec{v}})_1}, \ldots, \mathbf{A}_{(I_{\vec{v}})_{|I_{\vec{v}}|}})$, be the subvector of $\vec{\mathbf{A}}$ consisting of indices from $I_{\vec{v}}$ and let $\vec{\mathbf{A}}_{I_{\vec{t}}} = (\mathbf{A}_{(I_{\vec{t}})_1}, \ldots, \mathbf{A}_{(I_{\vec{t}})_{|I_{\vec{t}}|}})$ be the subvector of $\vec{\mathbf{A}}$ consisting of indices from $I_{\vec{t}}$. By the law of total probability we have that

$$\mathbb{P}_{\vec{\mathbf{A}} \sim Q}\left[|\{i \in \{\vec{\mathbf{A}}\} : \mathcal{A}(\mathbf{S}(\vec{\mathbf{A}}|\vec{t}))(k_u, i) > \gamma\}| > \frac{d}{8}\right]$$

$$= \sum_{x \in [k_u]^{k_u - d + 1 - |I_{\vec{v}}|}} \mathbb{P}_{\vec{\mathbf{A}} \sim Q}\left[|\{i \in \{\vec{\mathbf{A}}\} : \mathcal{A}(\mathbf{S}(\vec{\mathbf{A}}|\vec{t}))(k_u, i) > \gamma\}| > \frac{d}{8}\,\middle|\, \vec{\mathbf{A}}_{I_{\vec{t}}} = x\right] \mathbb{P}_{\vec{\mathbf{A}}_{I_{\vec{t}}} \sim Q_{I_{\vec{t}}}}\left[\vec{\mathbf{A}}_{I_{\vec{t}}} = x\right]$$

where $Q_{I_{\vec{t}}}$ is the distribution over $\vec{\mathbf{A}}_{I_{\vec{t}}}$ induced by $\vec{\mathbf{A}}$ having the uniform distribution on $Q$. We will show that

$$\mathbb{P}_{\vec{\mathbf{A}} \sim Q}\left[|\{i \in \{\vec{\mathbf{A}}\} : \mathcal{A}(\mathbf{S}(\vec{\mathbf{A}}|\vec{t}))(k_u, i) > \gamma\}| > \frac{d}{8}\,\middle|\, \vec{\mathbf{A}}_{I_{\vec{t}}} = x\right] \geq \frac{1}{24} \tag{41}$$

for any fixed $x \in [k_u]^{k_u - d + 1 - |I_{\vec{v}}|}$, which by the above implies Equation (37). Now for any fixed $x \in [k_u]^{k_u - d + 1 - |I_{\vec{v}}|}$, we have that $\vec{\mathbf{A}}_{I_{\vec{v}}} \in [k_u]^{|I_{\vec{v}}|}$ is uniformly distributed on the vectors of length $|I_{\vec{v}}|$ with distinct entries from $[k_u] \backslash \{x\}$. Furthermore for a fixed $x$ we have since $\mathbf{S}(\vec{\mathbf{A}}, \vec{t}) = ((((k_u, \vec{\mathbf{A}}_{t_1}), 0), \ldots, ((k_u, \vec{\mathbf{A}}_{t_n}), 0)))$ only depends on $\vec{\mathbf{A}}_{I_{\vec{t}}} = x$, that $\mathcal{A}(\mathbf{S}(\vec{\mathbf{A}}|\vec{t}))$ is fixed. Since $\mathcal{A}$ is a proper learning algorithm, the latter further implies that $\mathcal{A}(\mathbf{S}(\vec{\mathbf{A}}|\vec{t})) = h_{k', A'} \in \mathcal{H}^{\gamma, d}$, meaning that

$$\{i \in \{\vec{\mathbf{A}}\} : \mathcal{A}(\mathbf{S}(\vec{\mathbf{A}}|\vec{t}))(k_u, i) > \gamma\} = \{i \in \{\vec{\mathbf{A}}\} : h_{k', A'}(k_u, i) > \gamma\} = \begin{cases} \{\vec{\mathbf{A}}\} & \text{if } k' \neq k_u, \\ \{\vec{\mathbf{A}}\} \cap A' & \text{if } k' = k_u. \end{cases}$$

where we in the last equality have used that by definition of $h_{k', A'}$, it holds that $h_{k', A'}(k_u, t) = \gamma_{k', A'} > \gamma$ if $k' \neq k_u$, and $h_{k', A'}(k_u, t) = 0$ if $k' = k_u$ and $t \notin A'$, and $h_{k', A'}(k_u, t) = \gamma_{k', A'} > \gamma$ if $k' = k_u$ and $t \in A'$. For the first case in the above, $k' \neq k_u$, the size of the set inside the probability of Equation (41) is $|\{\vec{A}\}| = k_u - d + 1 \geq d + 2$ by Equation (34), and the lower bound in Equation (41) follows. In the second case $k' = k_u$ and $|\{\vec{\mathbf{A}}_{I_{\vec{t}}}\} \cap A'| = |\{x\} \cap A'| \geq d/2$, the lower bound in Equation (41) also follows. In the last case $k' = k_u$ and $|\{\vec{\mathbf{A}}_{I_{\vec{t}}}\} \cap A'| = |\{x\} \cap A'| < d/2$, we have that $|([k_u] \backslash \{x\}) \cap A'| = |([k_u] \cap A') \backslash (\{x\} \cap A')| \geq |([k_u] \cap A')| - |(\{x\} \cap A')| \geq d - 1 - (d/2 - 1/2) \geq (d - 1)/2$, where we have used in the last step that $|\{x\} \cap A'| < d/2$ implies $|\{x\} \cap A'| \leq d/2 - 1/2$. Furthermore we have that $|k_u \backslash \{x\}| = k_u - |\{x\}| = d - 1 + |I_{\vec{v}}|$. Using this we have that

$$\mathbb{E}_{\vec{\mathbf{A}} \sim Q}\left[|\{i \in \{\vec{\mathbf{A}}\} : \mathcal{A}(\mathbf{S}(\vec{\mathbf{A}}|\vec{t}))(k_u, i) > \gamma\}|\,\middle|\, \vec{\mathbf{A}}_{I_{\vec{t}}} = x\right] = \mathbb{E}_{\vec{\mathbf{A}} \sim Q}\left[|\{\vec{\mathbf{A}}\} \cap A'|\,\middle|\, \vec{\mathbf{A}}_{I_{\vec{t}}} = x\right]$$

$$\geq \mathbb{E}_{\vec{\mathbf{A}} \sim Q}\left[|\{\vec{\mathbf{A}}_{I_{\vec{v}}}\} \cap A'|\,\middle|\, \vec{\mathbf{A}}_{I_{\vec{t}}} = x\right] = \mathbb{E}_{\vec{\mathbf{A}} \sim Q}\left[\sum_{i=1}^{|I_{\vec{v}}|} \mathbb{1}\{(\vec{\mathbf{A}}_{I_{\vec{v}}})_i \in A'\}\,\middle|\, \vec{\mathbf{A}}_{I_{\vec{t}}} = x\right] = |I_{\vec{v}}| \frac{|([k_u] \backslash \{x\}) \cap A'|}{|[k_u] \backslash \{x\}|}$$

$$\geq |I_{\vec{v}}| \frac{(d-1)/2}{d - 1 + |I_{\vec{v}}|} = \frac{d-1}{2} \frac{|I_{\vec{v}}|}{d - 1 + |I_{\vec{v}}|} \geq \frac{d-1}{2} \frac{d}{d - 1 + d} = \frac{d-1}{2} \frac{d}{2d - 1} \geq \frac{d}{6}$$

where the second equality follows from the entries of $\vec{\mathbf{A}}_{I_{\vec{v}}}$ being distinct, the third equality from $\vec{\mathbf{A}}_{I_{\vec{v}}} \in [k_u]^{|I_{\vec{v}}|}$ being uniformly distributed on the vectors of length $|I_{\vec{v}}|$ with distinct entries from $[k_u] \backslash \{x\}$, the second to last inequality follows from $|I_{\vec{v}}| \geq d$ and $\frac{a}{b+a} \geq \frac{c}{b+c}$ for $a \geq c > 0$ and $b \geq 0$. Using this lower bound on the expectation and that $|\{\vec{A}\} \cap A'| \leq |A'| = d - 1 \leq d$ we have by Lemma C.3 that

$$\mathbb{P}_{\vec{\mathbf{A}} \sim Q}\left[|\{i \in \{\vec{\mathbf{A}}\} : \mathcal{A}(\mathbf{S}(\vec{\mathbf{A}}|\vec{t}))(k_u, i) > \gamma\}| > d/8\,\middle|\, \vec{\mathbf{A}}_{I_{\vec{t}}} = x\right]$$

$$= \mathbb{P}_{\vec{\mathbf{A}} \sim Q}\left[|\{i \in \{\vec{\mathbf{A}}\} : \mathcal{A}(\mathbf{S}(\vec{\mathbf{A}}|\vec{t}))(k_u, i) > \gamma\}|/d > 1/8\,\middle|\, \vec{\mathbf{A}}_{I_{\vec{t}}} = x\right] \geq \frac{1}{6} - \frac{1}{8} = \frac{1}{24}$$

which shows Equation (41).

We now show Equation (39). To this end let $\mathbf{X}_i$ be the random variable counting number of draws from $\mathcal{D}_t$ before seeing a new element in $[k_u - d + 1]$ after having seen $i - 1$ different elements in $[k_u - d + 1]$, i.e. $\mathbf{X}_i$ is geometrically distributed with parameter $p_i = \frac{k_u - d + 2 - i}{k_u - d + 1}$. We then have that $\mathbf{X} = \sum_{i=1}^{k_u - 2d} \mathbf{X}_i$, is the number of draws needed to have seen $k_u - 2d$ different elements in $[k_u - d + 1]$ meaning there are $k_u - d + 1 - (k_u - 2d) = d + 1$ unseen elements, i.e. $|I_{\vec{v}}| \geq d$. Thus if we can show that $\mathbf{X} > n$ with probability at least $1/2$ we have that Equation (39) follows, thus we show that. We have that

$$\mathbb{E}[\mathbf{X}] = \sum_{i=1}^{k_u - 2d} \frac{k_u - d + 1}{k_u - d + 2 - i} = (k_u - d + 1) \sum_{j=d+2}^{k_u - d + 1} \frac{1}{j} \geq (k_u - d + 1) \ln\left(\frac{k_u - d + 2}{d + 2}\right) \tag{42}$$

$$\geq (k_u - d + 1) \ln\left(\frac{k_u - d}{d + 2}\right) \tag{43}$$

where the first equality follows from the linearity of expectation and from the expectation of a geometric random variable with parameter $p$ being $1/p$, in the second equality we have shifted the index of the sum, and in the inequality we have used that $\sum_{j=m}^{n} \frac{1}{j} \geq \int_m^{n+1} \frac{1}{x} dx = \ln\left(\frac{n+1}{m}\right)$. Furthermore we have that

$$\mathbb{V}\mathrm{ar}\left[\mathbf{X}\right] = \sum_{i=1}^{k_u - 2d} \frac{1 - \frac{k_u - d + 2 - i}{k_u - d + 1}}{\left(\frac{k_u - d + 2 - i}{k_u - d + 1}\right)^2} \leq (k_u - d + 1)^2 \sum_{j=d+2}^{k_u - d + 1} \frac{1}{j^2} \leq (k_u - d + 1)^2 \int_{d+1}^{\infty} \frac{1}{x^2} dx \leq \frac{(k_u - d + 1)^2}{2}. \tag{44}$$

where the first equality follows from the independence of the $\mathbf{X}_i$s and from the variance of a geometric random variable with parameter $p$ being $\frac{1-p}{p^2}$, in the first inequality we have used that $1 - p_i \leq 1$, in the second inequality we have used that $\sum_{j=m}^{n} \frac{1}{j^2} \leq \int_{m-1}^{\infty} \frac{1}{x^2} dx$, and in the last inequality we have used that $\int_a^{\infty} \frac{1}{x^2} dx = \frac{1}{a}$ for $a > 0$. Using Chebyshev's inequality we have that

$$\mathbb{P}\left[|\mathbf{X} - \mathbb{E}[\mathbf{X}]| \geq \sqrt{2 \mathbb{V}\mathrm{ar}\left[\mathbf{X}\right]}\right] \leq \frac{1}{2},$$

implying that with probability at least $1/2$, it holds that

$$\mathbf{X} \geq \mathbb{E}[\mathbf{X}] - \sqrt{2 \mathbb{V}\mathrm{ar}\left[\mathbf{X}\right]} \geq (k_u - d + 1) \ln\left(\frac{k_u - d}{d + 2}\right) - (k_u - d + 1) = (k_u - d + 1) \ln\left(\frac{k_u - d}{(d + 2)e}\right). \tag{45}$$

where the second to last inequality follows from the bounds on $\mathbb{E}[X]$ and $\mathbb{V}\mathrm{ar}[X]$ derived in Equation (42) and Equation (44). Now using that

$$k_u = \left\lceil \frac{d}{16\varepsilon} \right\rceil \geq \frac{d}{16\varepsilon} \geq \frac{3d}{64\varepsilon} + \frac{d}{64\varepsilon} \geq \frac{3d}{64\varepsilon} + d$$

where the last inequality follows from $\varepsilon \leq 1/64e$. Now, we bound each term in the product of the last expression in Equation (45) by respectively

$$\frac{k_u - d}{(d + 2)e} \geq \frac{k_u - d}{2de} \geq \frac{3}{128e\varepsilon}$$

where we have used in the first inequality that $d \geq 2$, and

$$k_u - d + 1 \geq \frac{3d}{64\varepsilon}$$

which implies that with probability at least $1/2$, it holds that

$$\mathbf{X} \geq \frac{3d}{64\varepsilon} \ln\left(\frac{3}{128e\varepsilon}\right) > n,$$

which we concluded earlier would imply Equation (39).

Now for the expected loss bound in Equation (32), we notice that

$$\mathop{\mathbb{E}}_{\vec{\mathbf{A}}\sim Q}\left[\mathop{\mathbb{E}}_{\mathbf{S}\sim\mathcal{D}_{\vec{\mathbf{A}}}^n}\left[\mathcal{L}_{\mathcal{D}_{\vec{\mathbf{A}}}}^\gamma(\mathcal{A}(\mathbf{S}))\right]\right] = \mathop{\mathbb{E}}_{\vec{\mathbf{t}}\sim\mathcal{D}_t^n}\left[\mathop{\mathbb{E}}_{\vec{\mathbf{A}}\sim Q}\left[\mathop{\mathbb{P}}_{\mathbf{t}\sim\mathcal{D}_t}[\mathcal{A}(\mathbf{S}(\vec{\mathbf{A}}|\vec{\mathbf{t}}))(k_u,\vec{A}_{\mathbf{t}}) > \gamma]\right]\right] \tag{46}$$

$$\geq \mathop{\mathbb{E}}_{\vec{\mathbf{t}}\sim\mathcal{D}_t^n}\left[\mathop{\mathbb{E}}_{\vec{\mathbf{A}}\sim Q}\left[\mathop{\mathbb{P}}_{\mathbf{t}\sim\mathcal{D}_t}[\mathcal{A}(\mathbf{S}(\vec{\mathbf{A}}|\vec{\mathbf{t}}))(k_u,\vec{A}_{\mathbf{t}}) > \gamma]\right]\mathbb{1}\{|I_{\vec{\mathbf{t}}}| \geq d\}\right]$$

$$\geq \frac{1}{k_u - d + 1}\mathop{\mathbb{E}}_{\vec{\mathbf{t}}\sim\mathcal{D}_t^n}\left[\mathop{\mathbb{E}}_{\vec{\mathbf{A}}\sim Q}\left[|\{i \in \{\vec{\mathbf{A}}\} : \mathcal{A}(\mathbf{S}(\vec{\mathbf{A}}|\vec{t}))(k_u,i) > \gamma\}|\right]\mathbb{1}\{|I_{\vec{\mathbf{t}}}| \geq d\}\right]$$

where the first equality follows from similar arguments as in Equation (35), the second inequality from similar arguments as in Equation (40). Now letting $\vec{t}$ be any realization of $\vec{\mathbf{t}}$ with $|I_{\vec{t}}| \geq d$, we have that

$$\mathop{\mathbb{E}}_{\vec{\mathbf{A}}\sim Q}\left[|\{i \in \{\vec{\mathbf{A}}\} : \mathcal{A}(\mathbf{S}(\vec{\mathbf{A}}|\vec{t}))(k_u,i) > \gamma\}|\right]$$

$$= \sum_{x\in[k_u]^{k_u-d+1-|I_{\vec{t}}|}}\mathop{\mathbb{E}}_{\vec{\mathbf{A}}\sim Q}\left[|\{i \in \{\vec{\mathbf{A}}\} : \mathcal{A}(\mathbf{S}(\vec{\mathbf{A}}|\vec{t}))(k_u,i) > \gamma\}|\Big|\vec{\mathbf{A}}_{I_{\vec{t}}} = x\right]\mathop{\mathbb{P}}_{\vec{\mathbf{A}}_{I_{\vec{t}}}\sim Q_{I_{\vec{t}}}}\left[\vec{\mathbf{A}}_{I_{\vec{t}}} = x\right].$$

In the proof of Equation (41) we showed that for any fixed $x \in [k_u]^{k_u-d+1-|I_{\vec{t}}|}$, it holds that

$$\mathop{\mathbb{E}}_{\vec{\mathbf{A}}\sim Q}\left[|\{i \in \{\vec{\mathbf{A}}\} : \mathcal{A}(\mathbf{S}(\vec{\mathbf{A}}|\vec{t}))(k_u,i) > \gamma\}|\Big|\vec{\mathbf{A}}_{I_{\vec{t}}} = x\right] \geq \frac{d}{6},$$

which implies that

$$\mathop{\mathbb{E}}_{\vec{\mathbf{A}}\sim Q}\left[|\{i \in \{\vec{\mathbf{A}}\} : \mathcal{A}(\mathbf{S}(\vec{\mathbf{A}}|\vec{t}))(k_u,i) > \gamma\}|\right] \geq \frac{d}{6}.$$

Plugging this back into the expected loss bound, Equation (46), we get that

$$\mathop{\mathbb{E}}_{\vec{\mathbf{A}}\sim Q}\left[\mathop{\mathbb{E}}_{\mathbf{S}\sim\mathcal{D}_{\vec{\mathbf{A}}}^n}\left[\mathcal{L}_{\mathcal{D}_{\vec{\mathbf{A}}}}^\gamma(\mathcal{A}(\mathbf{S}))\right]\right] \geq \frac{1}{k_u-d+1}\frac{d}{6}\mathop{\mathbb{E}}_{\vec{\mathbf{t}}\sim\mathcal{D}_t^n}\left[\mathbb{1}\{|I_{\vec{\mathbf{t}}}| \geq d\}\right] \geq \frac{d}{12(k_u-d+1)} \geq 4\varepsilon/3$$

where the second to last inequality follows from Equation (39) saying $\mathbb{P}_{\vec{\mathbf{t}}\sim\mathcal{D}_t^n}\left[|I_{\vec{\mathbf{t}}}| \geq d\right] \geq 1/2$, and the last inequality follows from $k_u \leq d/(16\varepsilon) + 1$ and $d \geq 2$ implying that $k_u - d + 1 \leq d/(16\varepsilon)$, showing the existence of a $\vec{A} \in Q$ with the same expected loss bound over $\mathbf{S} \sim \mathcal{D}_{\vec{A}}^n$ and completing the proof of Equation (32) in the theorem. $\qquad\square$

