# OpenReview forum: "The Interplay Between Interpolation and Aggregation in Regression: Optimal Sample Complexity"
_ICML.cc/2026/Conference — ICML 2026 regular_

### Official Review · Reviewer_hUBV · 2026-03-09

**Soundness:** 4
**Presentation:** 3
**Significance:** 3
**Originality:** 3
**Overall Recommendation:** 5
**Confidence:** 4

**Summary:**

This paper studies the interpolation between interpolation and aggregation in regression defining two classes of natural aggregation rules: i) proper aggregation of interpolating hypotheses and ii) a more general aggregation rule over any finite set of hypothesis. The authors characterize the PAC sample complexity upto constant factors. The upper bound is attained by a simple median-based aggregation of three interpolating hypotheses, and $\gamma$-graph dimension characterizes learnability. This stands in contrast to $\gamma$-OIG dimension which characterizes learnability for PAC regression [1]. In addition, the authors improve the sample complexity lower bound of proper PAC learning by $\log \left(1/\epsilon\right)$, which is characterized by $\gamma$-graph dimension [1]. The authors conclude that the sample complexity of proper learning is strictly worse than the broad classes of aggregation rules they consider.


[1] Attias, I., Hanneke, S., Kalavasis, A., Karbasi, A., and Velegkas, G. Optimal learners for realizable regression: PAC learning and online learning. In Oh, A., Naumann, T., Globerson, A., Saenko, K., Hardt, M., and Levine, S. (eds.), Advances in Neural
Information Processing Systems 36

**Compliance With Llm Reviewing Policy:**

Affirmed.

**Final Justification:**

This is an interesting work and addresses an interesting question in the field of learning theory. I feel this paper has significant theoretical contributions and deserves a place in the conference.

**Key Questions For Authors:**

**Questions**

1. **Definition 3.3** When the authors define an interpolation-based aggregation algorithm, it might probably be worth mentioning that the aggregation $r$ could potentially be a function of the test-input point $x$. While a careful reader can observe this from lower bounds in Theorems 3.8 and 3.10, it might still be worth clarifying.

2. Although I verified the correctness of proof of theorem 3.8 by going over the appendix C.2, I got slightly confused by the proof sketch in lines 254-260 . In particular, it was not clear to me whether the randomness of the set $\mathbb A$ was fixed for all the training samples, or whether the set $\mathbb A$ is independently sampled for all the training samples. A similar clarification might be needed for some of the other lower bound proofs as well. The appendix makes it super-clear though.

3. The notation {x} when x is a vector is not defined in the proof of Theorem 3.12 (lines 1840-1847). While a reader can guess it to be the set consisting of the components of $x$, a small notation section may help. This same thing is true for $A$ as well.

4. (Minor) In the proof of Theorem 3.12, the proof sketch and the full proof in Appendix C.4 define the vector $A$ differently. While both are internally consistent, it would be good to have consistency in the notations in the main proof and the proof sketch. For example, in the proof sketch, $A$ is defined as a set with cardinality $d_\gamma - 1$; the proof sketch actually uses $A^c$ to denote the same thing. While technically correct, it would be easier for the reader to have consistency between both.

**Limitations:**

yes

**Strengths And Weaknesses:**

**Strengths**: The paper solves an important problem a natural problem of aggregation in PAC realizable regression. This builds on similar existing works [1] on PAC binary classification where a majority of three was shown to be optimal. While proving the desired results, this paper also strengthens the prior analysis of PAC realizable regression for proper learners. While it improves the lower bound sample complexity by a logarithmic factor, it also improves the realizable PAC regression error for proper learners by introducing a $\frac{1}{d_\gamma}$ factor inside the logarithmic term, where $d_\gamma$ denotes the $\gamma$-graph dimension (see line 352 and appendix lemma B.2). The paper is well written and I appreciate the fact that authors provide proof sketches in main paper instead of deferring all proofs to the appendix.

I verified most of the proofs in the appendix and did not find any issues.

**Weakness**: The paper is technically dense and may require some background to fully follow, which is largely unavoidable given the nature of the problem. The authors provide helpful references to prior work that guide readers through the technical developments.

---

> ### Author Rebuttal · Authors · 2026-03-30
>
> Dear reviewer hUBV,
>
> Thank you for reading and assessing our paper. We will now address the questions one by one.
>
> **Question 1:**
> >"Definition 3.3 When the authors define an interpolation-based aggregation algorithm, it might probably be worth mentioning that the aggregation could potentially be a function of the test-input point. While a careful reader can observe this from lower bounds in Theorems 3.8 and 3.10, it might still be worth clarifying."
>
> **Answer 1:**
>
> We thank the reviewer for pointing this out to us, and we agree that it would be good to add a comment mentioning that the aggregation rule $r$ is used pointwise.
>
> **Question 2:**
> >"Although I verified the correctness of proof of theorem 3.8 by going over the appendix C.2, I got slightly confused by the proof sketch in lines 254-260 . In particular, it was not clear to me whether the randomness of the set was fixed for all the training samples, or whether the set is independently sampled for all the training samples. A similar clarification might be needed for some of the other lower bound proofs as well. The appendix makes it super-clear though."
>
> **Answer 2:**
>
> We agree with the reviewer that it would improve the proof sketch to add a comment that the set $A$ is, on a high level, drawn independently of the sample, since the sample in the proof can be simulated as being generated by a random enumeration independent of the set $A$.
>
> **Question 3:**
> >"The notation $\{x\}$ when x is a vector is not defined in the proof of Theorem 3.12 (lines 1840-1847). While a reader can guess it to be the set consisting of the components of, a small notation section may help. This same thing is true for as well."
>
> **Answer 3:**
>
> We thank the reviewer for pointing this out to us, and we agree that this notation should be defined the first time it is used. We will correct this in the next version of the paper.
>
> **Question 4:**
> >"(Minor) In the proof of Theorem 3.12, the proof sketch and the full proof in Appendix C.4 define the vector differently. While both are internally consistent, it would be good to have consistency in the notations in the main proof and the proof sketch. For example, in the proof sketch, is defined as a set with cardinality; the proof sketch actually uses to denote the same thing. While technically correct, it would be easier for the reader to have consistency between both."
>
> **Answer 4:**
>
> We agree with the reviewer that it would improve the readability of the paper if we kept the notation in the two sections consistent. We will change that in the next version of the paper.

---

> > ### Author Rebuttal · Reviewer_hUBV · 2026-04-01
> >
> > All my concerns have been addressed, and I maintain my acceptance recommendation. While the paper is solid, I see its primary impact within the PAC learning and learning theory community, and thus do not increase the score further.

---

### Official Review · Reviewer_9cCn · 2026-03-09

**Soundness:** 3
**Presentation:** 3
**Significance:** 3
**Originality:** 3
**Overall Recommendation:** 5
**Confidence:** 3

**Summary:**

In this paper, the authors investigate the ability of proper learners, the learners generated by aggregating proper learners, and the general improper learners in the realizable regression setting. They discover that the sample complexity of the learners generated by aggregating finite proper learners is also governed by the $\gamma$-graph dimension instead of the $\gamma$-OIG dimension, which governs the sample complexity of general improper learners. They also show that there exists an arbitrarily large gap between the $\gamma$-graph dimension and $\gamma$-OIG dimension. Further, they also show that there is a log factor gap between the sample complexity of the proper learner and the learner generated by aggregating finite proper learners. They also show that the median-of-three is actually optimal among all learners generated by the aggregation of finite proper learners.

**Compliance With Llm Reviewing Policy:**

Affirmed.

**Ethical Review Concerns:**

This is an interesting paper, and my concerns have been fully resolved. Therefore, I keep my positive assessment.

**Final Justification:**

My concerns have been fully resolved. I will keep my positive assessment.

**Key Questions For Authors:**

I have the following questions:
1. Is it possible to get the high probability bound for the median of three algorithm?
2. What about the multiclass setting? Will there also be such a separation between the voting algorithm and a general improper learner?
3. Is it possible to add some discussion on the similarity between the majority of three in the binary classification case and the median of three in the regression setting?

**Limitations:**

Yes.

**Strengths And Weaknesses:**

This paper futher investigate the realizable regression model and characterizes the sample complexity of the learner generated by aggregating finite proper learners. They show the optimal sample complexity of a proper learner and the learner generated by aggregating finite proper learners. They show that the sample complexities for both types are governed by $\gamma$-graph dimension. However, as Attias et al. 2023 showed, the sample complexity of a general improper learner is governed by the $\gamma$-OIG dimension. This paper shows that there is an arbitrary gap between these two dimensions. This result is very interesting and reveals some inherent differences between the realizable regression and realizable binary classification.

The weakness, in my opinion, is that most of the results are not that surprising. After showing that the sample complexity of the learners generated by aggregating finite proper learners is governed by $\gamma$-graph dimension, the results are not that surprising. The median-of-three algorithm is similar to the majority-of-three algorithm for binary classification. The upper bound is also just an in-expectation bound instead of a high probability bound, which slightly weakens the results.

The presentation of the paper is pretty good and clean. Though I did not carefully check the details of the proof, I believe the proofs are sound.

---

> ### Author Rebuttal · Authors · 2026-03-30
>
> Dear reviewer 9cCn,
>
> Thank you for reading and assessing our paper. We will now adress the questions one by one.
>
> **Question 1:**
> >"Is it possible to get the high probability bound for the median of three algorithm?"
>
> **Answer 1:**
>
> The current high-probability bound that we believe we are able to show (not included in the paper), we could not prove is optimal. Furthermore, the bound does not provide any new insights compared to the in-expectation bound presented. Thus, we have chosen not to include a high-probability statement in the paper, to keep the story clean, as the paper already contains several statements. Determining whether the median of three algorithm is optimal with high probability is an interesting future line of research, and resolving it would most likely also have implications for other learning settings where a xxx of three algorithm makes sense.
>
>
>
> **Question 2:**
> >"What about the multiclass setting? Will there also be such a separation between the voting algorithm and a general improper learner?"
>
> **Answer 2:**
>
> The role of aggregation, especially voting algorithms, has been studied in [1], and the conclusion of their work is, on a high level, that the learnability of these methods is governed by a different dimension than the one that governs general improper learning, which the work of [2] shows is governed by the DS-dimension.
>
> **Question 3:**
> >"Is it possible to add some discussion on the similarity between the majority of three in the binary classification case and the median of three in the regression setting?"
>
> **Answer 3:**
> The main technical contribution of our analysis of the median of three in regression, comes from improving the upper bound for a single ERM, such that we can use the proof technique of [3] to get an optimal in-expectation bound for the median of three instead of the majority of three as in the binary classification setting. Furthermore, we remark that we do not know how to carry out the proof for the other alternative, the mean of three, which would also have been a natural candidate for extending the positive result of the majority of three from the binary classification setting.
>
> [1] Understanding Aggregations of Proper Learners in Multiclass Classification Julian Asilis, Mikael Møller Høgsgaard, Grigoris Velegkas
>
> [2] A Characterization of Multiclass Learnability Nataly Brukhim, Daniel Carmon, Irit Dinur, Shay Moran, Amir Yehudayoff
>
> [3] Majority-of-Three: The Simplest Optimal Learner? Ishaq Aden-Ali, Mikael Møller Høandgsgaard, Kasper Green Larsen, Nikita Zhivotovskiy

---

> > ### Author Rebuttal · Reviewer_9cCn · 2026-04-01
> >
> > I thank the authors for their answer to my questions. My questions have been fully resolved.

---

### Official Review · Reviewer_UgZX · 2026-03-12

**Soundness:** 4
**Presentation:** 2
**Significance:** 4
**Originality:** 3
**Overall Recommendation:** 5
**Confidence:** 4

**Summary:**

This submission studied the effects of interpolators' aggression on realizable regression. The authors systematically explored the complexity of different aggregation procedures in learning under $\gamma$-cutoff loss. An important concept examined by the authors is the $\gamma$- dimension, which is shown to be the key measure for aggregation procedures.  The core contributions of the paper are as follows: 1) establishing a lower bound of $\Omega(d_γ/\epsilon)$ related to the $\gamma$-graph dimension for aggregation algorithms based on interpolators (using proper rules); 2) extending this lower bound to a broader class of finite aggregation algorithms employing interpolating aggregation rules; 3) demonstrating that an extremely simple algorithm, the median-of-three interpolators, achieves a sample complexity of $O(d_γ/\epsiolon$), thereby matching the lower bound and proving its optimality; 4) revealing the strict advantage of aggregation algorithms over proper algorithms by proving a lower bound of $\Omega((d_γ/\epsilon)\log(1/\epsilon))$ for proper learning algorithms; and 5) constructing hypothesis classes with small $\gamma$-OIG dimension but infinite $\gamma$-graph dimension, highlighting a fundamental limitation of finite interpolating aggregation algorithms, they cannot achieve nontrivial performance.

**Compliance With Llm Reviewing Policy:**

Affirmed.

**Key Questions For Authors:**

Can the interpolator algorithm be made computationally efficient for general classes? Do the results extend to the agnostic setting? Theorem 3.10 shows that interpolating rules have fundamental limitations. A natural question is: Can one define a measure of how "far" an aggregation rule is from being interpolating.

**Limitations:**

The behavior of these aggregation algorithms under agnostic noise remains an open and crucial question. The authors stated while the learner of (Attias et al., 2023) is almost information-theoretically optimal, it is not efficiently implementable. In this submission, the "simple" median-of-three algorithm may be computationally infeasible in practice, limiting its practical impact. From Theorems 3.8, 3.12, the proposed algorithms cannot guarantee good performance on all classes.

**Strengths And Weaknesses:**

Soundness: The paper contains deep and comprehensive theoretical results, including complexity, lower and upper bounds, and identification bounds of algorithms. The weakness is notations are very heavy, not readers friendly.

Presentation: Clear logic, but not easy to read.

Significance: The paper addresses a fundamental question in learning theory: the power and limitations of aggregation, a technique central to modern machine learning.

Originality: While the paper heavily cites and builds upon Attias et al. 2023, it is far more than "just an extension." It is more accurately described as a significant deepening and broadening of that foundational work.

---

> ### Author Rebuttal · Authors · 2026-03-30
>
> Dear reviewer UgZX,
>
> Thank you for reading and assessing the paper. We will now address the questions.
>
> **Question 1:**
> >"Can the interpolator algorithm be made computationally efficient for general classes?"
> and
> >"Limitations:
> The behavior of these aggregation algorithms under agnostic noise remains an open and crucial question. The authors stated while the learner of (Attias et al., 2023) is almost information-theoretically optimal, it is not efficiently implementable. In this submission, the "simple" median-of-three algorithm may be computationally infeasible in practice, limiting its practical impact. From Theorems 3.8, 3.12, the proposed algorithms cannot guarantee good performance on all classes."
>
> **Answer 1:**
> Our learning algorithm hinges on empirical risk minimization, which in the simple setting of realizable binary classification(which is captured by realizable regression when the hypothesis class is binary valued and $ \gamma <1/2  $) cannot be guaranteed to be computationally tractable, even in simple settings such as 3-Term DNF [1, page 107], intersection of halfspaces [1, page 111 exercise 3.2], and even small neural networks [1,page 276].
> Regardless, empirical risk minimization is still widely used heuristically in practice via gradient methods(not guaranteed to find a global minimum), whereas the one-inclusion graph does not have a similar heuristic and is not implemented in practice.
>
>
> **Question 2:**
> >"Do the results extend to the agnostic setting?"
>
> **Answer 2:**
>
> Regarding learnability in the agnostic setting, there is some nuance in terms of the scale parameter. The work of [1] investigates this relationship and shows necessary and sufficient conditions for learnability in terms of the fat-shattering dimension and related notions. Here there is a gap between the necessary and sufficient conditions, since they are at different scales of the dimension. Thus, to extend our result to the agnostic setting, we would most likely have to consider the relation of our methods to the notion of fat-shattering or related notions, and this would be non-trivial.
>
> **Question 3:**
> >"Theorem 3.10 shows that interpolating rules have fundamental limitations. A natural question is: Can one define a measure of how "far" an aggregation rule is from being interpolating."
>
> **Answer 3:**
> We agree that it would be interesting to consider more aggregation rules, and a natural direction would be to define some measure of how far an aggregation rule is from being interpolating. We currently do not know how to formalize this, but we think it would be a great idea for future work.
>
> [1] Understanding Machine Learning: From Theory to Algorithms Shai Shalev-Shwartz and Shai Ben-David
>
> [2] Prediction, Learning, Uniform Convergence, and Scale-sensitive Dimensions Peter L. Bartlett and Philip M. Long.(arXiv version)

---

> > ### Author Rebuttal · Reviewer_UgZX · 2026-04-03
> >
> > would like to see some future work.

---

### Official Review · Reviewer_AV9C · 2026-03-12

**Soundness:** 2
**Presentation:** 2
**Significance:** 2
**Originality:** 2
**Overall Recommendation:** 4
**Confidence:** 1

**Summary:**

the paper study interpolating solutions, and their aggregation

**Compliance With Llm Reviewing Policy:**

Affirmed.

**Final Justification:**

Thank you

**Key Questions For Authors:**

none

**Strengths And Weaknesses:**

looks an interesting topic, on the other hand perhaps outdated and impractical. I wish I had more time to read it

---

> ### Author Rebuttal · Authors · 2026-03-30
>
> Dear reviewer AV9C,
>
> Thank you for reading and assessing the paper.

---

> > ### Author Rebuttal · Reviewer_AV9C · 2026-04-01
> >
> > OK

---

### Decision · Program_Chairs · 2026-04-30

**Decision:**

Accept (regular)

**Comment:**

This paper considers aggregation of interpolators i.e. models that perfectly fit training data. It presents both lower and upper bounds for the aggregated interpolator. Its main result is an $\Omega(d/\epsilon)$ sample complexity for the aggregated interpolator to achieve expected cutoff loss $\epsilon$, where d is the graph dimension. It shows this bound holds for two types of aggregation rules (proper and interpolating), and a matching upper bound is achievable by the simple median-of-three interpolator. It also shows this bound is better than the lower bound for proper learning, suggesting aggregation can be more powerful than proper learning.

The work is interesting and solid. All reviewers are on the positive side. Two reviewers raise a concern on its readability to the audience without sufficient background, and authors are encouraged to enhance the readability (and incorporate other feedback) in the final version.

I would add one minor point: in Introduction, it is stated the $\Omega(d/\epsilon)$ complexity "strengthens Theorem 1 of (Attias et al., 2023), which established a similar bound for a single interpolator”. But there is no further elaboration. It would be helpful to clarify how exactly this bound strengthens the prior theorem, so readers can immediately grasp the theoretical gain.